# Learning to Localize Leakage
# of Cryptographic Sensitive Variables

**Jimmy Gammell      Anand Raghunathan      Abolfazl Hashemi      Kaushik Roy**
*{jgammell, araghu, abolfazl, kaushik}@purdue.edu*
*Elmore Family School of Electrical and Computer Engineering, Purdue University*

**Reviewed on OpenReview:** *https://openreview.net/forum?id=9qxCSU8nDO&*

## Abstract

While cryptographic algorithms such as the ubiquitous Advanced Encryption Standard (AES) are secure, *physical implementations* of these algorithms in hardware inevitably 'leak' sensitive data such as cryptographic keys. A particularly insidious form of leakage arises from the fact that hardware consumes power and emits radiation in a manner that is statistically associated with the data it processes and the instructions it executes. Supervised deep learning has emerged as a state-of-the-art tool for carrying out *side-channel attacks*, which exploit this leakage by learning to map power/radiation measurements throughout encryption to the sensitive data operated on during that encryption. In this work we develop a principled deep learning framework for determining the relative leakage due to measurements recorded at different points in time, in order to inform *defense* against such attacks. This information is invaluable to cryptographic hardware designers for understanding *why* their hardware leaks and how they can mitigate it (e.g. by indicating the particular sections of code or electronic components which are responsible). Our framework is based on an adversarial game between a classifier trained to estimate the conditional distributions of sensitive data given subsets of measurements, and a budget-constrained noise distribution which probabilistically erases individual measurements to maximize the loss of this classifier. We demonstrate our method's efficacy and ability to overcome limitations of prior work through extensive experimental comparison on 6 publicly-available power/EM trace datasets from AES, ECC and RSA implementations. Our PyTorch code is available here.

## 1 Introduction

The Advanced Encryption Standard (AES)[1] (Daemen & Rijmen, 1999; 2013) is widely used and trusted for protecting sensitive data. For example, it is approved by the United States National Security Agency for protecting top secret information (Committee on National Security Systems, 2003), it is a major component of the Transport Layer Security (TLS) protocol (Rescorla, 2000) which underlies the security of HTTPS (Rescorla, 2000), and it is used in payment card readers to secure card information before transmission to financial institutions (Bluefin Payment Systems, 2023).

AES aims to keep data secret when it is transmitted over insecure channels that are accessible to unknown and untrusted parties (e.g. via wireless transmissions which may be intercepted, or storage on hard drives accessible to untrusted individuals). Prior to transmission, the data is first encoded and partitioned into a sequence of fixed-length bitstrings called *plaintexts*. Each plaintext is then *encrypted* into a *ciphertext* by applying an invertible function from a family of functions indexed by an integer called a *cryptographic key*. This family of functions is designed so that if the key is sampled uniformly at random, then the plaintext and ciphertext are marginally independent. The key is known to the sender and intended recipients of the transmission, and is kept secret from potential eavesdroppers. Thus, the intended recipients can use the key

---

[1]For clarity of exposition we discuss AES here, but our technique also applies to other algorithms.

to *decrypt* the ciphertext back into the original plaintext, while eavesdroppers who possess the ciphertext but not the key learn nothing about the plaintext.

Clearly, such an algorithm is effective only if the cryptographic key remains outside of the hands of attackers. AES is believed to be 'algorithmically secure' in the sense that it is infeasible to determine the cryptographic key by exploiting its intended inputs and outputs: plaintexts and ciphertexts (Mouha, 2021; Tao & Wu, 2015). Despite this, *physical implementations* of AES in hardware 'leak' information about their cryptographic keys. This phenomenon, called *side-channel leakage*, occurs because hardware inevitably emits measurable physical signals that are statistically associated with the data it processes and the instructions it executes (Mangard et al., 2007). There are many diverse physical side-channels which can leak, such as program/operation execution time (Kocher, 1996; Lipp et al., 2018; Kocher et al., 2019), temperature (Hutter & Schmidt, 2014), and sound due to vibration of electronic components (Genkin et al., 2014). In this work the side-channels we consider are *power and electromagnetic (EM) radiation over time* (Kocher et al., 1999; Mangard et al., 2007), which are major security vulnerabilities for AES implementations (Genkin et al., 2016; Bronchain & Standaert, 2020).

*Side-channel attacks* exploit this leakage to break cryptographic implementations by revealing secret internal variables such as cryptographic keys. In this work we consider *profiling* side-channel attacks (Chari et al., 2003; Mangard et al., 2007), which assume a worst-case scenario where the attacker possesses a clone of the target device and can repeatedly measure its power/radiation over time while encrypting arbitrary plaintexts using arbitrary keys. These sequences of power/radiation measurements are recorded as real vectors called *traces*, where each element encodes a measurement at a fixed point in time relative to the start of encryption. The attacker can thereby gather data from the clone device to model the conditional distribution of the secret variable given a trace. They can then defeat the target device by measuring its power/radiation traces, feeding them to the model, and revealing its secret internal variables.

Supervised deep learning has emerged as a state-of-the-art technique for this modeling task, achieving comparable or superior performance to prior approaches while requiring far less data preprocessing and feature selection (Maghrebi et al., 2016; Benadjila et al., 2020; Zaid et al., 2020; Wouters et al., 2020; Lu et al., 2021; Bursztein et al., 2023). Older non-deep learning attacks were mostly based on parametric statistics and had major limitations such as restrictive modeling assumptions (Messerges, 2000; Chari et al., 2003; Agrawal et al., 2005; Schindler et al., 2005; Hospodar et al., 2011) and inability to scale to long traces (Chari et al., 2003; Archambeau et al., 2006). Deep learning overcomes these limitations and consequently poses a major and growing threat to a wide assortment of security measures and evaluations that were designed with the limitations of older attacks in mind.

In this work, we seek to leverage deep learning to *defend* against side-channel attacks by identifying specific points in time at which power/radiation measurements leak sensitive data. Our aim is to aid the designers of implementations in understanding *why* their implementations leak (e.g. by indicating the particular sections of code or electronic components which are responsible) and thereby enable targeted mitigation of the leaks. Our key contributions are:

- We propose a principled information theoretic quantity which measures the 'leakiness' of individual power/EM radiation measurements. Unlike prior approaches, ours is sensitive to arbitrary statistical associations between a chosen secret variable and subsets of measurements. Our quantity is implicitly defined through a constrained optimization problem.

- We propose a novel deep learning algorithm called Adversarial Leakage Localization (ALL) which approximately solves this optimization problem. ALL is based on an adversarial game played between a neural net 'attacker' trained to classify secret data using power/radiation traces, and a budget-constrained noise distribution trained to 'defend' against the attack by introducing noise to individual measurements in the traces. Due to the budget constraint, noise cannot be added everywhere and must be rationed for the leakiest measurements. After training we can thereby surmise the leakiness of each measurement from its noisiness.

- We compare ALL with 11 baseline methods on 6 publicly-available power and EM radiation side-channel leakage datasets from implementations of the AES, ECC, and RSA cryptographic standards.

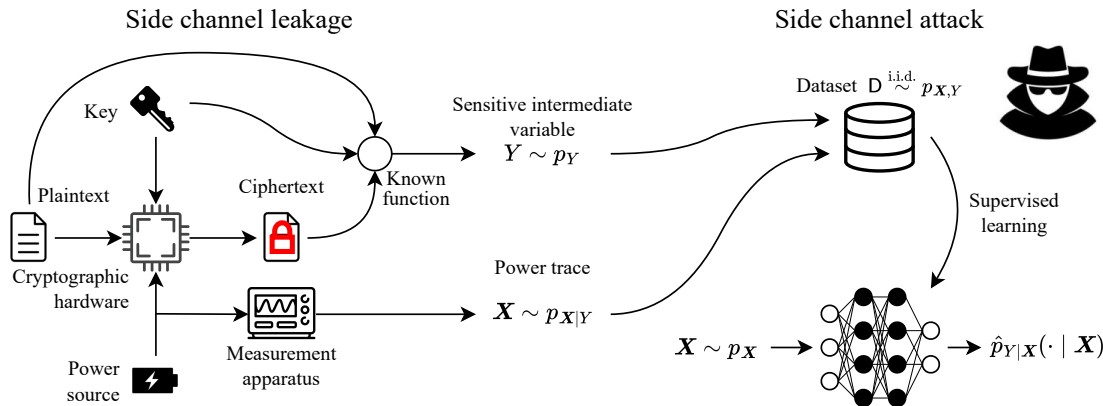

Figure 1: Diagram illustrating our probabilistic framing of side-channel leakage in the special case of power side-channel leakage from a symmetric-key (e.g. AES) cryptographic implementation. A cryptographic device consumes power over time while encrypting data. The power consumption leaks the secret key because it is statistically associated with key-dependent internal variables. Our work considers profiling side-channel attacks, a worst-case scenario where attackers can freely model the relationship between an implementation's secret data and its power consumption over time, then use this model to attack the same device.

> To our knowledge this is by far the most comprehensive and quantitative comparison of leakage localization algorithms which has been done, and we release our code and procedure in the hope of facilitating reproducibility and benchmarking of future work in this area.

## 2 Background and Setting

See Table 1 for a summary of the important notation we will use throughout the rest of the paper. Our setting is standard in the context of profiling power/EM side-channel analysis (Chari et al., 2003; Mangard et al., 2007) and is illustrated in Fig. 1. We assume to have a cryptographic device which encrypts data in a manner dependent on secret variable $y \in \mathsf{Y}$, where $\mathsf{Y}$ is a finite set (e.g. consisting of bitstrings encoding all possible values of the variable). We assume to have some measurement apparatus that allows us to measure the power consumption or EM radiation throughout encryption. We view the resulting measurement sequences as vectors $\boldsymbol{x} \in \mathbb{R}^T$, called *traces*, where $T \in \mathbb{Z}_{++}$ denotes the number of measurements per trace. We view the secret variable as a random variable $Y \sim p_Y$ where $p_Y$ is a simple (e.g. uniform) distribution under our control. The resulting trace $\boldsymbol{X} \mid Y \sim p_{\boldsymbol{X}|Y}$ then comes from a complicated and *a priori* unknown distribution dictated by factors such as the hardware, environment, and measurement setup. Here each element $X_t$ of the random vector $\boldsymbol{X} \equiv (X_1, \ldots, X_T)$ represents a power/radiation measurement at a fixed point in time relative to the start of encryption. In this work we assume that the distributions $p_{\boldsymbol{X},Y}(\cdot \mid y)$ exist and have full support for all $y \in \mathsf{Y}$, which is reasonable because empirically power consumption usually has a 'random' component which is well-described by additive Gaussian noise (Mangard et al., 2007). Most profiling side-channel attacks amount to collecting a dataset $\mathsf{D} \overset{\text{i.i.d.}}{\sim} p_{\boldsymbol{X},Y}$ and using supervised (deep or otherwise) learning to model $p_{Y|\boldsymbol{X}}$.

Given these jointly-distributed $\boldsymbol{X}, Y$, we seek to define for each $X_t$ a scalar quantifying its 'leakiness,' i.e. the extent to which it can be exploited by attackers to learn $Y$ from $\boldsymbol{X}$. Towards this end it is useful to consider the Shannon conditional mutual information (Shannon, 1948)

$$\mathbb{I}[Y; X_t \mid \mathsf{S}] := \mathbb{E}\left[\log p_{Y|X_t,\mathsf{S}}(Y \mid X_t, \mathsf{S}) - \log p_{Y|\mathsf{S}}(Y \mid \mathsf{S})\right], \quad \mathsf{S} \subseteq \{X_1, \ldots, X_T\} \setminus \{X_t\}. \tag{1}$$

Intuitively, $\mathbb{I}[Y; X_t \mid \mathsf{S}]$ tells us the extent to which our uncertainty about $Y$ is reduced upon observing $X_t$, provided we have already observed the elements of $\mathsf{S}$.

Each individual quantity $\mathbb{I}[Y; X_t \mid \mathsf{S}]$ tells us something about the leakiness of $X_t$. However, for each $t$ there are $2^{T-1}$ such quantities, and it is not obvious how they should be combined into a single scalar. Clearly,

Table 1: A summary of the important quantities and variables used throughout this work.

| Notation | Explanation |
|---|---|
| $p_A$, $p_{A|B}$, etc. | Probability distribution of random variable $A$, conditional distribution of $A$ given $B$, etc. |
| $\mathbb{E} f(A, B)$, $\mathbb{E}_A f(A, B)$ | Expected value of $f(A, B)$ (left: over all randomness, right: over randomness of $A$) |
| $\mathbb{I}[A; B \mid C] \in \mathbb{R}_+$ | Conditional mutual information between $A$ and $B$ given $C$ |
| $T \in \mathbb{Z}_{++}$ | Number of measurements/timesteps in power trace |
| $\boldsymbol{X} \equiv (X_1, \ldots, X_T) \in \mathbb{R}^T$ | Power trace (random variable) consisting of $T$ measurements |
| $Y \in \mathsf{Y}$ | Secret data (random variable) from finite set $\mathsf{Y}$ (e.g. $\mathsf{Y} = \{0, 1, \ldots, 255\}$ for $\mathtt{uint8}$ $Y$) |
| $\boldsymbol{\gamma} \equiv (\gamma_1, \ldots, \gamma_T) \in [0, 1]^T$ | Occlusion probabilities, where $\gamma_t$ denotes probability of occluding $X_t$ |
| $C \in \mathbb{R}_{++}$ | Budget for occlusion probability vector, which incurs cost $\sum_{t=1}^{T} c(\gamma_t)$ |
| $\overline{\gamma} = C/(C + T) \in (0, 1)$ | Re-parameterized version of $C$ which is less-sensitive to dataset |
| $\boldsymbol{\mathcal{A}}_{\boldsymbol{\gamma}} \equiv (\mathcal{A}_{\gamma,1}, \ldots, \mathcal{A}_{\gamma,T}) \in \{0, 1\}^T$ | Occlusion mask (random variable), where $\mathrm{Prob}(\mathcal{A}_{\gamma,t} = 0) = \gamma_t$ |
| $\boldsymbol{X} \odot \boldsymbol{\mathcal{A}}_{\boldsymbol{\gamma}} \in \mathbb{R}^T$ | Occluded trace (elementwise product of trace and occlusion mask) |
| $\Phi_{\boldsymbol{\theta}}(\cdot \mid \boldsymbol{X} \odot \boldsymbol{\mathcal{A}}_{\boldsymbol{\gamma}}, \boldsymbol{\mathcal{A}}_{\boldsymbol{\gamma}}) \in [0, 1]$ | Predicted distribution of secret data given occluded trace and mask, by classifier w/ weights $\boldsymbol{\theta}$ |

$X_t$ should be considered leaky if $\mathbb{I}[Y; X_t] > 0$ and non-leaky if $\mathbb{I}[Y; X_t \mid \mathsf{S}] = 0$ $\forall \mathsf{S}$. More-subtly, there are many practical scenarios in which we have *second-order leakage*, where $\mathbb{I}[Y; X_t] = 0$ but $\mathbb{I}[Y; X_t \mid X_{t'}] > 0$ for some $t' \neq t$ (i.e. $X_t$ *alone* reveals nothing about $Y$, but reveals something useful about $Y$ *when combined* with $X_{t'}$). For example, many cryptographic implementations use a Boolean masking countermeasure (Chari et al., 1999; Benadjila et al., 2020) whereby a sensitive variable is decomposed into a pair of independent 'random shares' which are operated on at distant points in time. Hence, the naïve choice of $\mathbb{I}[Y; X_t]$ as our definition of the leakiness of $X_t$ would not work.

Another naive choice would be to define the leakiness of $X_t$ as $\mathbb{I}[Y; X_t \mid \{X_1, \ldots, X_T\} \setminus \{X_t\}]$. This addresses the insensitivity of $\mathbb{I}[Y; X_t]$ to second-order leakage. However, it introduces a new shortcoming: when there are many leaky measurements with 'redundant' information, we may have $\mathbb{I}[Y; X_t \mid \{X_1, \ldots, X_T\} \setminus \{X_t\}] \approx 0$ even if $\mathbb{I}[Y; X_t] \gg 0$. In other words, $X_t$ has little *new* information about $Y$ which is not already provided by the other measurements. As we will show, this phenomenon creates issues for many prior deep learning-based leakage localization algorithms.

We will subsequently propose a natural notion of leakiness which is sensitive to all the quantities described by Eqn. 1. Before we do so, let us consider relevant prior work and its limitations.

## 3 Existing work and its limitations

We consider prior work in the side-channel analysis literature which may be leveraged for leakage localization. One prominent category of such work is *parametric statistics-based methods* which use non-deep learning techniques to look for pairwise associations between the measurements $X_t$ and $Y$. The other is *neural net attribution-based methods*, where 1) a profiling side-channel attack is carried out with supervised deep learning, and 2) the neural net is 'interpreted' to determine the relative importance of its input features. Refer to Appendix C for further details.

### 3.1 First-order parametric statistics-based methods

First-order parametric statistics-based methods (Mangard et al., 2007; Chari et al., 2003; Brier et al., 2004) are widely used for understanding leakage due to their simplicity, interpretability, and low computational cost. However, these cannot detect leakage of order 2 or higher, and make restrictive assumptions about the relationship between $\boldsymbol{X}$ and $Y$. Thus, such methods are ill-suited to our work's 'black-box' leakage localization setting where we make minimal assumptions about the cryptographic implementation being evaluated.

In practice, leakage mitigation will likely be done in a 'white-box' setting where hardware designers understand the implementation and have access to its internal variables. In this setting, $n^{\text{th}}$-order leakage can often be decomposed into first-order leakage of $\geq n$ internal variables, which can then be localized individually with first-order methods. However, such analysis is error-prone and relies on careful analysis of the implementation. For example, Benadjila et al. (2020) released the second order-leaking ASCADv1-fixed dataset and analyzed

its leakage by decomposing it into 2 pairs of first-order-leaking internal variables. Subsequently, Egger et al. (2022) noted additional internal variables which contribute to leakage but were missed in the initial analysis. We view black-box deep learning-based methods such as ours as *complementary* with parametric white-box analysis: the latter provides an interpretable and hyperparameter-free assessment of known-leaky internal variables, while the former can detect leakage stemming from both known and unknown sources.

### 3.2  Neural net attribution-based methods

There is a great deal of prior work on localizing leakage by applying interpretability techniques to neural nets which have been trained to perform side-channel attacks (Masure et al., 2019; Hettwer et al., 2020; Jin et al., 2020; Zaid et al., 2020; Wouters et al., 2020; van der Valk et al., 2021; Golder et al., 2022; Li et al., 2022; Perin et al., 2022; Schamberger et al., 2023; Yap et al., 2023; Li et al., 2024; Yap et al., 2025). Most of these techniques can be summarized as follows: 1) use standard supervised deep learning techniques to train a model $\hat{p} \approx p_{Y|\boldsymbol{X}}$ using data, and 2) use interpretability techniques to estimate the influence of each input feature on the model, on average over the dataset. For example, the Gradient Visualization (GradVis) technique of Masure et al. (2019) estimates the leakiness of $X_t$ by $\mathbb{E}_{\boldsymbol{X},Y} \left| -\frac{\partial}{\partial x_t} \log \hat{p}(Y \mid \boldsymbol{x})|_{\boldsymbol{x}=\boldsymbol{X}} \right|$, and the 1-Occlusion technique of Hettwer et al. (2020) estimates it as $\mathbb{E}_{\boldsymbol{X},Y} |\hat{p}(Y \mid \boldsymbol{X}) - \hat{p}(Y \mid (\boldsymbol{1} - \boldsymbol{I}_t) \odot \boldsymbol{X})|$ where $\boldsymbol{I}_t$ denotes column $t$ of the identity matrix. In this work we consider as baselines the recent OccPOI method (Yap et al., 2025), the $m$-Occlusion and $2^{\text{nd}}$-order $m$-Occlusion techniques (Schamberger et al., 2023), as well as GradVis (Masure et al., 2019), Saliency (Simonyan et al., 2014; Hettwer et al., 2020), 1-Occlusion (Zeiler & Fergus, 2014; Hettwer et al., 2020), LRP (Bach et al., 2015; Hettwer et al., 2020), and Input $*$ Grad (Shrikumar et al., 2017; Wouters et al., 2020). Note that these subsume the deep learning baselines considered by Yap et al. (2025); Schamberger et al. (2023).

Most of these methods are prone to detecting only some leaking measurements while ignoring others. GradVis, Saliency, 1-Occlusion, LRP and Input $*$ Grad compute the leakiness of $X_t$ by occluding or differentiating the input $x_t$ to $\hat{p}(Y \mid X_1, \ldots, X_{t-1}, x_t, X_{t+1}, \ldots, X_T)$ and observing its change in output. However, as discussed in Sec. 2, when many of the measurements carry 'redundant' information, one may have $\mathbb{I}[Y; X_t \mid \{X_1, \ldots, X_T\} \setminus \{X_t\}] \approx 0$ even if $\mathbb{I}[Y; X_t] \gg 0$. In this case a well-fit $\hat{p}$ becomes essentially constant with respect to $x_t$, causing these methods to spuriously estimate low leakiness for $X_t$.

While $m$-Occlusion, $2^{\text{nd}}$-order $m$-Occlusion, and OccPOI occlude multiple inputs simultaneously and are thus less-susceptible to this issue, they have their own shortcomings. $m$-Occlusion is like 1-Occlusion except that it occludes $m$-diameter windows rather than single points, which has an undesirable smoothing effect and is helpful only when the 'redundant' measurements are temporally-local. $2^{\text{nd}}$-order $m$-Occlusion entails occluding all pairs of windows, which is expensive because it requires $\Theta(T^2)$ passes through the dataset with the neural net. Additionally, while it provides the interesting and unique ability to discern whether leakage is first-order or second-order, we find that it gives little improvement over $m$-Occlusion when adapted to estimate the leakiness of a single point. Unlike the other considered methods, OccPOI does not assign a leakiness estimate to every measurement. Rather, it is a heuristic which aims to identify a non-unique minimal-cardinality set of measurements sufficient for $\hat{p}$ to attain some classification performance when all other measurements are occluded. Additionally, it is expensive because it requires $\Omega(T)$ non-parallelizable passes through the dataset with the neural net.

## 4  Our method: Adversarial Leakage Localization (ALL)

Here we propose a novel algorithm called Adversarial Leakage Localization (ALL) for localizing leakage which addresses the shortcomings of prior work. In line with Sec. 2, we have jointly-distributed power/EM radiation traces $\boldsymbol{X} \coloneqq (X_1, \ldots, X_T)$ and secret data $Y$. We seek to define for each $X_t$ a scalar quantifying its 'leakiness,' i.e. its usefulness to attackers for learning $Y$ from $\boldsymbol{X}$.

Intuitively, ALL is based on an adversarial game played between a neural net 'attacker' trained to predict $Y$ from $\boldsymbol{X}$, and a budget-constrained noise distribution trained to 'defend' against the attack by introducing noise to individual measurements to maximize the loss of the classifier. Because of the budget constraint, increasing the noise applied to one measurement requires reducing the noise of other measurements. Thus,

noise cannot simply be applied everywhere, and must be 'triaged' so that leakier measurements get more noise. After training we can surmise the leakiness of a measurement from the amount of noise which has been applied to it.

In this section we first propose a constrained optimization problem which implicitly defines a notion of 'leakiness' which is sensitive to all the terms $\mathbb{I}[Y; X_t \mid \mathsf{S}] : \mathsf{S} \subseteq \{X_1, \ldots, X_T\} \setminus \{X_t\}$. We then derive ALL as a practical deep learning algorithm which approximately solves this optimization problem. We conclude by explicitly contrasting ALL with prior work. Refer to Appendix D for an extended version of this section with proofs and derivations, and Algorithm 2 for pseudocode.

### 4.1 Implicit definition of leakiness through a constrained optimization problem

We define a vector $\boldsymbol{\gamma} \in [0, 1)^T$ which we name the *occlusion probabilities*. $\boldsymbol{\gamma}$ parameterizes a distribution over a binary random vector with range $\{0, 1\}^T$ which we call the *occlusion mask*: $\boldsymbol{\mathcal{A}_\gamma} \equiv (\mathcal{A}_{\gamma,1}, \ldots, \mathcal{A}_{\gamma,T}) \sim p_{\boldsymbol{\mathcal{A}_\gamma}} :$ $\mathcal{A}_{\gamma,t} \sim \text{Bernoulli}(1 - \gamma_t)$. For arbitrary vectors $\boldsymbol{x} \in \mathbb{R}^T$, $\boldsymbol{\alpha} \in \{0, 1\}^T$, let us denote $\boldsymbol{x_\alpha} := (x_t : t = 1, \ldots, T : \alpha_t = 1)$, i.e. the sub-vector of $\boldsymbol{x}$ containing its elements for which the corresponding element of $\boldsymbol{\alpha}$ is 1. We can accordingly use $\boldsymbol{\mathcal{A}_\gamma}$ to obtain random sub-vectors $\boldsymbol{X}_{\boldsymbol{\mathcal{A}_\gamma}}$ of $\boldsymbol{X}$. Note that $\gamma_t$ denotes the probability that $X_t$ will *not* be an element of $\boldsymbol{X}_{\boldsymbol{\mathcal{A}_\gamma}}$.

We assign to each element of $\boldsymbol{\gamma}$ a strictly-increasing 'cost' $c : [0, 1) \to \mathbb{R}_+ : x \mapsto \frac{x}{1-x}$, with the properties $c(0) = 0$ and $\lim_{x \to 1} c(x) = \infty$. We seek to solve the constrained optimization problem

$$\min_{\boldsymbol{\gamma} \in [0,1)^T} \quad \mathcal{L}(\boldsymbol{\gamma}) := \mathbb{I}[Y; \boldsymbol{X}_{\boldsymbol{\mathcal{A}_\gamma}} \mid \boldsymbol{\mathcal{A}_\gamma}] \quad \text{such that} \quad \sum_{t=1}^{T} c(\gamma_t) = C \tag{2}$$

where $C \in \mathbb{R}_{++}$ is a 'budget' hyperparameter. Intuitively, the mutual information term tells us the extent to which the 'occluded' trace $\boldsymbol{X}_{\boldsymbol{\mathcal{A}_\gamma}}$ 'leaks' $Y$, and $\boldsymbol{\gamma}$ is optimized to distribute a fixed 'budget of occlusion probability' among the individual elements of $\boldsymbol{X}$ to minimize this leakage.

As discussed in Appendix D.1, during this optimization process each $\gamma_t$ is 'pushed' up towards 1 in proportion to a weighted sum over *all* values $\mathbb{I}[Y; X_t \mid \mathsf{S}] : \mathsf{S} \subseteq \{X_1, \ldots, X_T\} \setminus \{X_t\}$. Thus, ALL is sensitive to all associations between $Y$ and subsets of $\{X_1, \ldots, X_T\}$. This is in contrast to parametric methods which consider only pairwise associations between each $X_t$ and Y, methods like 1-Occlusion, GradVis, Saliency, Input $*$ Grad and LRP which are sensitive to associations between $Y$ and the sets $\{X_1, \ldots, X_T\}, \{X_1, \ldots, X_T\} \setminus \{X_t\}$, and OccPOI, $m$-Occlusion, and $2^{\text{nd}}$-order $m$-Occlusion, which consider larger yet still tiny subsets of the power set of $\{X_1, \ldots, X_T\}$.

Due to the budget constraint, increasing $\gamma_t$ requires reducing other $\gamma_\tau$, $\tau \neq t$. Let us denote by $\boldsymbol{\gamma}^*$ a solution to Eqn. 2. Each $\gamma_t^*$ will be closer to 1 if $X_t$ is 'leakier' in the sense that it has greater mutual information with $Y$, conditioned on other $X_\tau$, $\tau \neq t$. Thus, we propose using $\gamma_t^*$ to measure the 'leakiness' of $X_t$.

### 4.2 Deep learning-based implementation

We will now re-frame this problem in a way that is amenable to standard deep learning techniques. Refer to Fig. 2 for a diagram.

We first re-parameterize it into an unconstrained problem by defining the variable $\boldsymbol{\eta} := \text{softmax}(\tilde{\boldsymbol{\eta}})$, $\tilde{\boldsymbol{\eta}} \in \mathbb{R}^T$ and letting $\boldsymbol{\gamma}$ be a function from $\mathbb{R}^T \to [0, 1]^T$ satisfying

$$c(\gamma_t(\tilde{\boldsymbol{\eta}})) = C\eta_t \iff \gamma_t(\tilde{\boldsymbol{\eta}}) = \text{sigmoid}(\log C + \log(\text{softmax}(\tilde{\boldsymbol{\eta}})_t)). \tag{3}$$

We can now optimize with respect to $\tilde{\boldsymbol{\eta}}$ instead of $\boldsymbol{\gamma}$, letting us drop the constraint because it is satisfied for any $\tilde{\boldsymbol{\eta}}$. Note that it is cheap and numerically-stable to map $\tilde{\boldsymbol{\eta}}$ to $\boldsymbol{\gamma}$ in PyTorch.

Next, as described in Appendix D.2, we can approximate the mutual information term of Eqn. 2 with a neural net. Note that $\mathbb{I}[Y; \boldsymbol{X}_{\boldsymbol{\mathcal{A}_\gamma}} \mid \boldsymbol{\mathcal{A}_\gamma}] = \sum_{\boldsymbol{\alpha} \in \{0,1\}^T} p_{\boldsymbol{\mathcal{A}_\gamma}}(\boldsymbol{\alpha}) \mathbb{I}[Y; \boldsymbol{X_\alpha}]$ where each $\mathbb{I}[Y; \boldsymbol{X_\alpha}] = \mathbb{E} \log p_{Y|\boldsymbol{X_\alpha}}(Y \mid \boldsymbol{X_\alpha}) - \mathbb{E} \log p_Y(Y)$. The right terms can be dropped because their corresponding terms in the full expression do not depend on $\boldsymbol{\gamma}$. The conditional distributions in the left terms can each be approximated by using

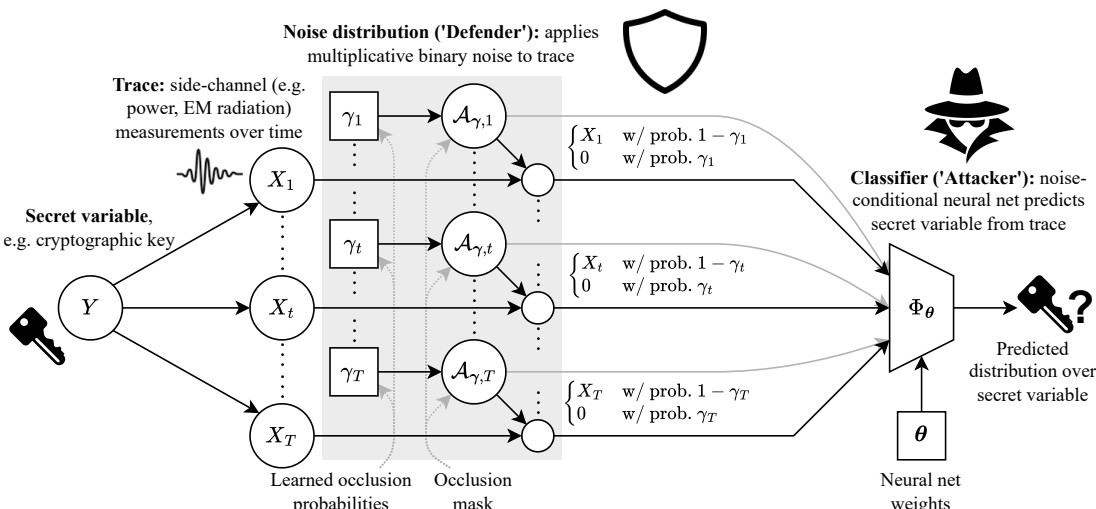

Figure 2: A diagram illustrating our Adversarial Leakage Localization (ALL) algorithm. A classifier $\Phi_{\boldsymbol{\theta}}$ is trained to 'attack' a cryptographic implementation by predicting its secret data $Y$ from power/EM radiation traces $\boldsymbol{X} \equiv (X_1, \ldots, X_T)$. Simultaneously, a noise distribution is trained to 'defend against the attack' by occluding the classifier's individual input features $X_t$ with probabilities $\gamma_t$, subject to a budget constraint which prevents trivially occluding every feature with probability 1. Because of the constraint, increasing $\gamma_t$ necessarily entails decreasing $\gamma_\tau$ for some $\tau \neq t$, so the noise distribution must preferentially apply noise to leakier features. Thus, after training we may interpret $\gamma_t$ as the 'leakiness' of $X_t$.

supervised deep learning with cross-entropy loss to classify $Y$ from $\boldsymbol{X}_{\boldsymbol{\alpha}}$. There are $2^T$ such distributions and it would be infeasible to train this many neural nets independently. Instead, similarly to Lippe et al. (2022), we train a single neural net to estimate all the distributions by occluding its inputs according to $\mathcal{A}_{\boldsymbol{\gamma}}$ and feeding $\mathcal{A}_{\boldsymbol{\gamma}}$ as an auxiliary input. Thus, we can approximate Eqn. 2 with the optimization problem

$$\min_{\tilde{\boldsymbol{\eta}} \in \mathbb{R}^T} \max_{\boldsymbol{\theta} \in \mathbb{R}^P} \quad \mathcal{L}_{\mathrm{adv}}(\tilde{\boldsymbol{\eta}}, \boldsymbol{\theta}) := \mathbb{E} \log \Phi_{\boldsymbol{\theta}} \left( Y \mid \boldsymbol{X} \odot \mathcal{A}_{\boldsymbol{\gamma}(\tilde{\boldsymbol{\eta}})}, \mathcal{A}_{\boldsymbol{\gamma}(\tilde{\boldsymbol{\eta}})} \right) \tag{4}$$

where $\Phi_{\boldsymbol{\theta}} : \mathsf{Y} \times \mathbb{R}^T \times \{0,1\}^T \to (0,1)$ is a neural net with weights $\boldsymbol{\theta}$ and softmax output activation, and $\Phi_{\boldsymbol{\theta}}(y \mid \boldsymbol{x} \odot \boldsymbol{\alpha}, \boldsymbol{\alpha})$ denotes its estimated probability that $Y = y$ given $\boldsymbol{X}_{\boldsymbol{\alpha}} = \boldsymbol{x}_{\boldsymbol{\alpha}}$. This can be approximately solved using alternating SGD-style algorithms, similarly to GANs (Goodfellow et al., 2014).

To use SGD-like algorithms we must estimate $\nabla_{\boldsymbol{\theta}} \mathcal{L}_{\mathrm{adv}}(\tilde{\boldsymbol{\eta}}, \boldsymbol{\theta})$ and $\nabla_{\tilde{\boldsymbol{\eta}}} \mathcal{L}_{\mathrm{adv}}(\tilde{\boldsymbol{\eta}}, \boldsymbol{\theta})$ with Monte Carlo integration. The former is routine in the context of DNN training. However, the latter is nontrivial because $\mathcal{L}_{\mathrm{adv}}$ has the form $\mathbb{E}_{\boldsymbol{\alpha} \sim p_{\tilde{\boldsymbol{\eta}}}} f(\boldsymbol{\alpha})$ where $\boldsymbol{\alpha}$ is discrete. There is a large body of work on gradient estimation for functions of this nature, which can broadly be categorized into unbiased REINFORCE (Williams, 1992)-based estimators with variance reduction strategies, and biased estimators based on relaxing $p_{\tilde{\boldsymbol{\eta}}}$ into a continuous distribution for which we can use the reparameterization trick (Rezende et al., 2014; Kingma & Welling, 2014). In our experiments we use the biased CONCRETE estimator (Maddison et al., 2017) with fixed temperature $\tau = 1$ because it is cheap and simple, and we find it yields nearly the same performance as more-complicated estimators we tried. We conjecture that we get strong performance with this simple and biased estimator because our performance metrics are sensitive only to the *relative* leakiness assigned to measurements, and the bias does not significantly affect this. Note that $f(\boldsymbol{\alpha}) \equiv \mathbb{E} \log \Phi_{\boldsymbol{\theta}}(Y \mid \boldsymbol{X} \odot \boldsymbol{\alpha}, \boldsymbol{\alpha})$ has been defined for $\boldsymbol{\alpha} \in \{0,1\}^T$, but after the CONCRETE relaxation we may have any $\boldsymbol{\alpha} \in [0,1]^T$. Thus, we replace it with the modified function $f(\boldsymbol{\alpha}) \equiv \mathbb{E} \log \Phi_{\boldsymbol{\theta}}(Y \mid \boldsymbol{X} \odot \boldsymbol{\alpha} + \boldsymbol{\varepsilon} \odot (\mathbf{1} - \boldsymbol{\alpha}), \boldsymbol{\alpha})$ where $\boldsymbol{\varepsilon} \sim \mathcal{N}(0,1)^T$.

Our method is mainly sensitive to 3 hyperparameters: the learning rates of $\boldsymbol{\theta}$ and $\tilde{\boldsymbol{\eta}}$, and the budget hyperparameter $C$. Rather than tuning $C$ directly, we find it easier to tune the hyperparameter $\overline{\gamma} := \frac{C}{C+T}$. $\overline{\gamma}$ is equal to the occlusion probability of each measurement when $\tilde{\boldsymbol{\eta}}$ is constant, and is less-sensitive to the data dimensionality $T$ than $C$.

### 4.3 Differences from prior work

Whereas GradVis, Saliency, LRP, Input $*$ Grad and 1-Occlusion effectively perturb single input features to the classifier and analyze the change in its outputs, ALL generally perturbs many inputs simultaneously. This is useful in settings where there are many 'redundant' leaking measurements and the impact of perturbing only one of them is small.

Like ALL, $m$-Occlusion, $2^{\text{nd}}$-order $m$-Occlusion and OccPOI also simultaneously occlude multiple input features. The key differences of our method are: 1) ALL samples from a distribution over all $2^T$ possible occlusion masks optimized to maximally hurt the performance of the classifier, whereas prior work iterates over a tiny subset of possible occlusion masks chosen heuristically. 2) ALL leverages the gradient of classifier loss with respect to relaxed occlusion masks, whereas prior work uses only zeroeth-order information from forward passes with 'hard' occlusion masks. 3) We simultaneously optimize the mask distribution to maximally hurt the classifier, and the classifier weights to be optimal for the current mask distribution. In contrast, prior work trains the classifier with standard supervised learning techniques, then 'interprets' the fixed classifier with occlusion.

## 5 Experimental results

### 5.1 Synthetic datasets where we know 'ground truth' leakiness

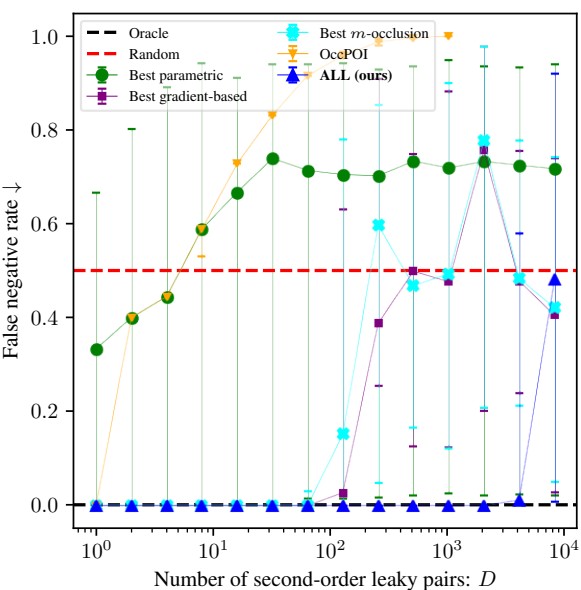

Figure 3: A toy setting described in Sec. 5.1 where ALL (ours) significantly outperforms baselines. We sample 1 non-leaky feature and $D$ second-order leaky pairs, then plot the false negative rate, defined as the proportion of points incorrectly assigned leakiness less than or equal to that of the non-leaky point, as we increase $D$. ALL (ours) succeeds for $D$ up to $64\times$ higher the best prior deep learning-based approach, and the first-order parametric methods completely fail in this setting. Dots denote median and error bars denote min–max over 5 random seeds.

**Toy setting where ALL succeeds and prior work fails** As we will show, these differences lead to significant performance gains, as well as speedup relative to $2^{\text{nd}}$-order $m$-Occlusion and OccPOI.

We generate a sequence of binary-label $2D+1$-feature classification datasets consisting of ordered pairs $(\boldsymbol{X}, Y)$ sampled independently as follows (see Appendix E.1 for details): $Y \sim \mathcal{U}\{0,1\}$, $R \sim \mathcal{U}\{0,1\}$, $M_i \sim \mathcal{U}\{0,1\}$ for $i = 1, \ldots, D, X_R \sim \mathcal{N}(R, 1), X_{M_i} \sim \mathcal{N}(M_i, 1), X_{Y \oplus M_i} \sim \mathcal{N}(Y \oplus M_i, 1)$ for $i = 1, \ldots, D$. Here we denote

by $\oplus$ the exclusive-or operation and $\boldsymbol{X} := (X_R, X_{M_1}, X_{Y \oplus M_i}, \ldots, X_{M_D}, X_{M_D \oplus Y})$. Intuitively, we can view the features $X_R$, $X_{M_i}$, and $X_{Y \oplus M_i}$ as noisy observations of $R$, $M_i$, and $Y \oplus M_i$, respectively. Note that $R$ tells us nothing about $Y$, and while *individually* each variable $M_i$, $Y \oplus M_i$ is independent of $Y$, the *pairs* of variables $\{M_i, Y \oplus M_i\}$ allow us to determine $Y$ through the identity $Y = Y \oplus M_i \oplus M_i$. An ideal leakage localization algorithm should indicate that each feature $X_{M_i}$, $X_{Y \oplus M_i}$ is leaking, while $X_R$ is not.

In Fig. 3, we plot the performance of ALL and prior work as we sweep the value of $D$. While prior deep learning-based methods succeed for small $D$, they fail when $D$ grows large because the individual contribution of each $\{X_{M_i}, X_{Y \oplus M_i}\}$ is 'drowned out' in the sense that $\mathbb{I}[Y; \{X_{M_i}, X_{Y \oplus M_i}\} \mid \{X_{M_j}, X_{Y \oplus M_j} : j \neq i\}]$ vanishes. ALL succeeds for $D$ up to $64\times$ larger than prior work because as the classifier becomes more-reliant on particular features they are subject to a higher occlusion probability, mitigating this effect. For completeness we also include first-order parametric methods in our comparison; these fail because they are not sensitive to the second-order associations between $Y$ and $\{X_{M_i}, X_{Y \oplus M_i}\}$.

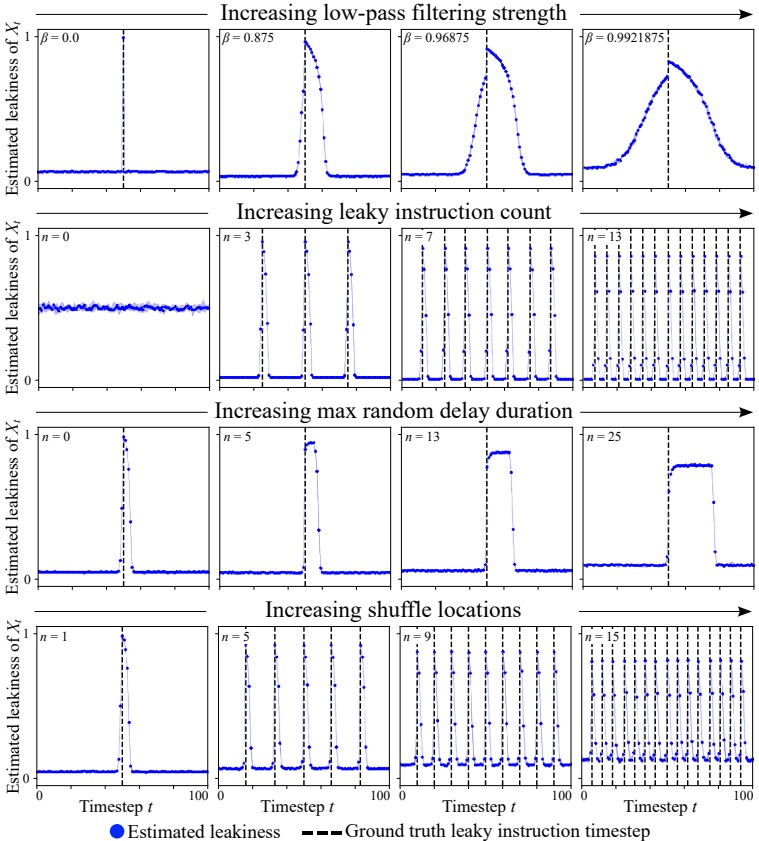

Figure 4: Output $\boldsymbol{\gamma}^* \equiv \boldsymbol{\gamma}(\tilde{\boldsymbol{\eta}}^*)$ of our ALL algorithm when applied to simulated AES-128 power trace datasets based on the Hamming weight model of (Mangard et al., 2007, ch. 4), as described in Appendix E.2. Leakiness estimated by ALL is consistent with the ground truth timestep at which leaky instruction(s) are executed across varying low-pass filtration strength, leaky instruction count, random no-op insertion, and random shuffling.

**Simulated AES-128 datasets**   We next apply ALL to a variety of simulated AES datasets. These are a useful complement to subsequent experiments on real datasets because we can validate ALL's outputs against ground truth knowledge about which timesteps are leaking, as well as gain insight into its behavior when individually varying particular dataset properties. Traces are simulated using the Hamming weight leakage model of (Mangard et al., 2007, ch. 4), which decomposes total power consumption as $X_t = X_{\text{data},t} + X_{\text{op},t} + X_{\text{resid},t}$. Here $X_{\text{data},t}$ is a function of the Hamming weight of the data currently being operated on, $X_{\text{op},t}$ is a function of the operation currently being applied to the data, and $X_{\text{resid},t}$ models

remaining sources of noise as a Gaussian random variable. Further details can be found in Appendix E.2. Note that while this model applies to a particular device studied by Mangard et al. (2007), the relationship between data and power consumption is device-dependent and often eludes simple characterization.

In Fig. 4 we simulate several factors of variation which may be expected to occur in realistic settings, and observe the change in behavior of ALL. *First*, we apply a varying-strength discrete low-pass filter to the simulated traces. As the strength increases, the peak of the estimated leakiness remains centered at the timestep of the leaky instruction while becoming more-diffuse. *Second*, we vary the number of leaky instructions and see that ALL consistently produces similar-height peaks at every leaky instruction. *Third*, we introduce a random delay before the leaky instruction, which causes ALL's peak diffuses over the set of timesteps at which the instruction may occur. *Fourth*, we shuffle the location of the leaky instruction so that it may occur at various points in time. As when varying the number of instructions, ALL produces similar-height peaks at each point in time at which the instruction sometimes occurs.

## 5.2 Real power and EM radiation leakage datasets

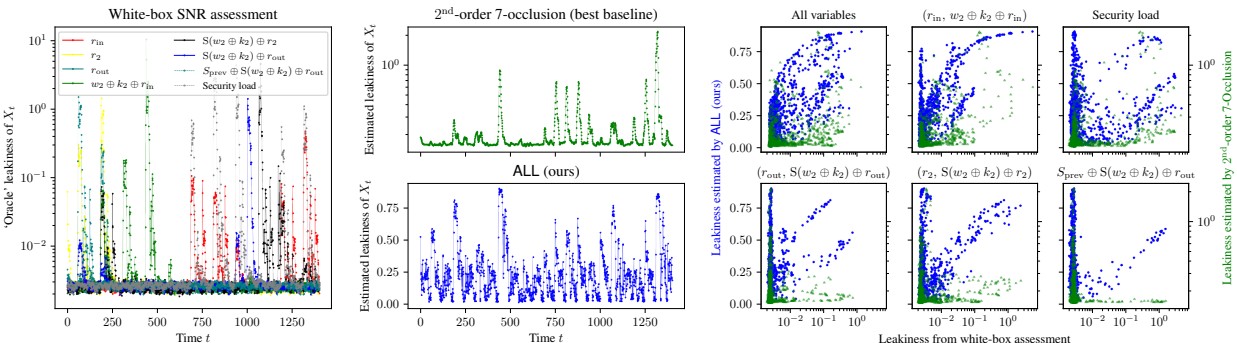

Figure 5: A qualitative comparison on ASCADv1-variable between the estimated leakiness by a white-box SNR-based assessment following Egger et al. (2022), ALL (our method), and $2^{\text{nd}}$-order 7-Occlusion (the strongest baseline). (**left**) Superimposed per-timestep leakiness of 8 internal AES variables which contribute to leakage of our targeted 'secret variable' $Y \equiv \mathrm{S}(w_2 \oplus k_w)$. We consider $Y$ to leak to the extent that at least one of these internal variables leaks. (**center**) The per-timestep leakiness of $Y$, as estimated by ALL and the baseline. Ideally, these estimates should align with peaks in the white-box assessment. (**right**) The per-timestep leakiness as estimated by ALL and the baseline, vs. the mean SNR of subsets of the internal AES variables (indicated in plot titles). In the plot labeled 'All variables', ALL exhibits a stronger positive association with the aggregate white-box assessment than the baseline. For individual variable subsets, ALL consistently produces a |/-shaped structure with a diagonal trend indicating agreement between estimated leakiness and SNR for the variables of interest, and a vertical band corresponding to leakage from other variables. In contrast, the baseline often exhibits an L-shaped structure, with the horizontal band reflecting spuriously low leakiness estimates at timesteps where the internal variables are known to have significant leakage. It appears that the baseline is mainly sensitive to $(r_{\text{in}}, w_2 \oplus k_2 \oplus r_{\text{in}})$ and possibly the security load (which leaks at similar times as $r_{\text{in}}$), but understates or misses leakage due to the other variables.

We compare our method to prior work on 6 publicly-available datasets of real recorded side-channel emissions and metadata covering diverse settings, as described in Appendix E.3.1. See Appendix E.3 for an extended version of this section including full implementation details, additional experiments, and ablation studies.

**Experimental setup** Despite their methodological differences, all considered methods may be viewed as a function which maps a dataset to a sequence of scalars encoding the estimated leakiness of $X_t$ for each $t$. For all deep learning baselines apart from ALL, we first train a supervised classifier to predict secret variables $Y$ from power traces $(X_1, \ldots, X_T)$. We then use the 'importance' of $X_t$ to the classifier's prediction about $Y$, on average over the profiling set, as the estimated leakiness of $X_t$. The definition of 'importance' is prescribed

by the method. For ALL, we simply run our method as described in Sec. 4, then directly use the trained occlusion probability $\gamma_t^*$ as the estimated leakiness of $X_t$.

For the supervised classifiers and the classifier denoted $\Phi_\theta$ in ALL, we use the same ReLU MLP architecture trained with the AdamW optimizer (see Appendix E.3.2). For both supervised classification and ALL training runs, we use the same minibatch size and training step count, tune the important hyperparameters for each dataset using random search with a 50-run budget, and leave the rest at reasonable defaults (see Appendix E.3.3). Supervised classifier hyperparameters are tuned to minimize correct-key rank (similar to maximizing accuracy). For ALL we cannot use this approach, so we instead use a criterion based on occluding the inputs to a frozen supervised classifier and observing its change in performance (see Appendix E.3.5). Baselines are implemented using Captum (Kokhlikyan et al., 2020) where possible. OccPOI and ($2^{\text{nd}}$-order) $m$-Occlusion are adapted to the setting of our paper as described in Appendix E.3.2.

**Performance evaluation strategies and results**   In Fig. 5, in line with previous work (Masure et al., 2019; Wouters et al., 2020; Schamberger et al., 2023; Yap et al., 2025), we compute a white-box leakage assessment on ASCADv1-variable and qualitatively compare it to the outputs of ALL and the strongest baseline. As is standard, we target the variable $Y \equiv \text{S}(w_2 \oplus k_2)$. Because this AES implementation employs Boolean masking, $Y$ does not directly influence the power consumption. Nonetheless, there are 3 pairs of internal variables which *do* directly influence power consumption and can be combined to determine $Y$ (as well as more-complicated identities involving the security load and $S_{\text{prev}} \oplus \text{S}(w_2 \oplus k_2) \oplus r_{\text{out}}$) (Egger et al., 2022). Ideally, a leakage localization algorithm should consider each $X_t$ leaky to the extent that at least one such variable with utility for determining $Y$ leaks. In Fig. 5 ALL successfully identifies the leakage of all 3 pairs of variables, while the best considered baseline clearly identifies leakage from only one of the pairs.

In the interest of scaling our comparison to many baselines and datasets, we next consider quantitative performance metrics. For real-world datasets quantitative performance evaluation is challenging because we lack 'ground truth' knowledge about leakage, and there is no consensus on the best way to do so. We thus employ 4 performance evaluation metrics which are conceptually similar to evaluation strategies in prior work. Because the specific numbers used to encode leakiness by the various baselines are not directly comparable, we use metrics which are sensitive only to the *relative* leakiness of different timesteps – i.e. under which a vector of estimated leakiness values $(\gamma_1^*, \ldots, \gamma_T^*)$ has the same performance as $(f(\gamma_1^*), \ldots, f(\gamma_T^*))$ for any strictly-increasing $f : \mathbb{R} \to \mathbb{R}$.

Table 2 lists performance across four complementary metrics which emphasize different aspects of performance. See Appendix E.3.4 for details about each metric. The *oracle agreement* metric assesses agreement with a white-box leakiness evaluation, and is given by the Spearman rank correlation coefficient between estimated leakiness and the average SNR of known leaky internal variables (similar to Fig. 5). The *template attack minimum traces to disclosure (MTD)*[2] metric assesses the utility of the estimated-leakiest measurements for conducting a side-channel attack – higher-fidelity leakiness estimates are expected to allow leakier measurements to be fed to the attacker, leading to better attack performance. The *forward DNN occlusion test*[2] measures the extent to which the performance of a deep learning-based side-channel attack is preserved as we occlude all but the estimated-leakiest measurements, and is mainly sensitive to spurious or incorrectly-ordered high estimates. The *reverse DNN occlusion test*[2] measures the extent to which performance is preserved as we occlude all but the estimated-least-leaky measurements, and is mainly sensitive to spurious low estimates. As seen in Table 2, ALL outperforms all baselines on the majority of datasets under every metric except for the forward DNN occlusion test.

## 6   Conclusion

We have proposed a novel algorithm for localizing side-channel leakage from cryptographic implementations. Unlike prior work, ours is sensitive to arbitrary statistical associations responsible for leakage and operates in a 'black box' manner, requiring only a supervised learning-style dataset with labels denoting the secret

---

[2]The template attack MTD metric is similar to evaluations done in Masure et al. (2019); Yap et al. (2025). The DNN occlusion tests were inspired by the ZB-KGE and KRPC algorithms of Hettwer et al. (2020). As described in Appendix E.3.4, we alter implementation details to summarize these as scalars and improve reliability on second-order datasets.

Table 2: A comparison of the considered deep learning-based leakage localization algorithms on 6 datasets and under 4 quantitative performance metrics. We report mean $\pm$ 1 standard deviation over 5 random seeds. **Bold numbers** denote the method with the best mean performance (ties broken by lower variance). $\uparrow$ denotes that a higher number is better, and $\downarrow$ denotes that a lower number is better. (**Oracle agreement** $\uparrow$) The Spearman rank correlation coefficient between the predicted leakiness and an 'oracle' leakiness derived from a white-box SNR-based assessment exploiting knowledge of the internal variables contributing to leakage of the target. (**Template attack MTD** $\downarrow$) The minimum traces to disclosure (MTD) of a template attack when using the predicted leakiness for point of interest (feature) selection. (**Forward DNN occlusion test** $\downarrow$) The mean single-trace correct-key rank of a DNN-based attacker when occluding all but the $k$ predicted-leakiest timesteps, on average for $k \in \{1, \dots, T\}$. (**Reverse DNN occlusion test** $\uparrow$) The mean single-trace correct-key rank of a DNN-based attacker when occluding all but the $k$ predicted-*least-leaky* timesteps, on average for $k \in \{1, \dots, T\}$.

| | Method | 2nd-order datasets | | 1st-order datasets | | | |
| --- | --- | --- | --- | --- | --- | --- | --- |
| | | ASCADv1-f | ASCADv1-r | DPAv4 | AES-HD | OTiAiT | OTP |
| **Oracle agreement** $\uparrow$ | GradVis | $0.48 \pm 0.02$ | $0.27 \pm 0.01$ | $0.198 \pm 0.009$ | $0.07 \pm 0.01$ | $0.55 \pm 0.05$ | $0.57 \pm 0.02$ |
| | Saliency | $0.47 \pm 0.02$ | $0.26 \pm 0.01$ | $0.198 \pm 0.008$ | $0.07 \pm 0.01$ | $0.67 \pm 0.06$ | $0.58 \pm 0.02$ |
| | Input $*$ Grad | $0.47 \pm 0.02$ | $0.25 \pm 0.01$ | $0.202 \pm 0.009$ | $0.08 \pm 0.02$ | $0.71 \pm 0.05$ | $0.60 \pm 0.02$ |
| | LRP | $0.47 \pm 0.02$ | $0.25 \pm 0.01$ | $0.202 \pm 0.009$ | $0.08 \pm 0.02$ | $0.71 \pm 0.05$ | $0.60 \pm 0.02$ |
| | OccPOI | $0.07 \pm 0.01$ | $0.064 \pm 0.004$ | $0.030 \pm 0.008$ | $0.044 \pm 0.009$ | $0.07 \pm 0.02$ | $0.01 \pm 0.02$ |
| | 1-Occlusion | $0.47 \pm 0.02$ | $0.25 \pm 0.01$ | $0.202 \pm 0.009$ | $0.08 \pm 0.01$ | $0.71 \pm 0.05$ | $0.60 \pm 0.02$ |
| | $m^*$-Occlusion | $0.49 \pm 0.02$ | $0.41 \pm 0.01$ | $0.32 \pm 0.01$ | $0.18 \pm 0.05$ | $0.72 \pm 0.04$ | $0.77 \pm 0.01$ |
| | $2^{\text{nd}}$-order 1-Occlusion | $0.51 \pm 0.01$ | $0.27 \pm 0.01$ | $0.206 \pm 0.009$ | $0.08 \pm 0.01$ | $0.74 \pm 0.05$ | $0.60 \pm 0.02$ |
| | $2^{\text{nd}}$-order $m^*$-Occlusion | $0.52 \pm 0.01$ | $0.42 \pm 0.01$ | $\mathbf{0.330 \pm 0.009}$ | $0.19 \pm 0.05$ | $0.75 \pm 0.04$ | $0.788 \pm 0.007$ |
| | ALL (ours) | $\mathbf{0.794 \pm 0.006}$ | $\mathbf{0.60 \pm 0.01}$ | $0.317 \pm 0.002$ | $\mathbf{0.22 \pm 0.03}$ | $\mathbf{0.782 \pm 0.001}$ | $\mathbf{0.848 \pm 0.003}$ |
| **Tmpl. attack MTD** $\downarrow$ | GradVis | $686 \pm 100$ | $1162 \pm 1000$ | $2.7 \pm 0.1$ | $20014 \pm 6000$ | $1.4 \pm 0.3$ | $1.378 \pm 0.007$ |
| | Saliency | $726 \pm 100$ | $1412 \pm 2000$ | $2.7 \pm 0.1$ | $19438 \pm 6000$ | $1.14 \pm 0.02$ | $1.379 \pm 0.005$ |
| | Input $*$ Grad | $675 \pm 100$ | $1194 \pm 2000$ | $2.6 \pm 0.1$ | $19893 \pm 6000$ | $1.14 \pm 0.02$ | $1.378 \pm 0.003$ |
| | LRP | $675 \pm 100$ | $1194 \pm 2000$ | $2.6 \pm 0.1$ | $19893 \pm 6000$ | $1.14 \pm 0.02$ | $1.378 \pm 0.003$ |
| | OccPOI | $787 \pm 100$ | $942 \pm 200$ | $71 \pm 30$ | $25000.000$ | $\mathbf{1.08 \pm 0.03}$ | $1.47 \pm 0.05$ |
| | 1-Occlusion | $667 \pm 100$ | $1376 \pm 2000$ | $2.65 \pm 0.08$ | $20011 \pm 6000$ | $1.14 \pm 0.02$ | $1.379 \pm 0.003$ |
| | $m^*$-Occlusion | $673 \pm 70$ | $727 \pm 400$ | $9 \pm 1$ | $16283 \pm 10$ | $1.17 \pm 0.02$ | $1.382 \pm 0.007$ |
| | $2^{\text{nd}}$-order 1-Occlusion | $709 \pm 100$ | $1086 \pm 1000$ | $2.65 \pm 0.08$ | $20222 \pm 6000$ | $1.14 \pm 0.02$ | $1.378 \pm 0.003$ |
| | $2^{\text{nd}}$-order $m^*$-Occlusion | $642 \pm 60$ | $710 \pm 400$ | $9 \pm 1$ | $\mathbf{16033 \pm 700}$ | $1.16 \pm 0.03$ | $1.381 \pm 0.008$ |
| | ALL (ours) | $\mathbf{459 \pm 40}$ | $\mathbf{394 \pm 20}$ | $\mathbf{2.22 \pm 0.01}$ | $17582 \pm 5000$ | $1.11 \pm 0.02$ | $\mathbf{1.363 \pm 0.007}$ |
| **Fwd. DNN occlusion** $\rightarrow$ | GradVis | $108.6 \pm 0.5$ | $96.8 \pm 0.3$ | $9.5 \pm 0.6$ | $125.6 \pm 0.3$ | $1.9 \pm 0.2$ | $1.013 \pm 0.002$ |
| | Saliency | $108.5 \pm 0.4$ | $96.3 \pm 0.4$ | $9.5 \pm 0.7$ | $125.6 \pm 0.3$ | $1.8 \pm 0.1$ | $1.014 \pm 0.001$ |
| | Input $*$ Grad | $108.5 \pm 0.4$ | $96.8 \pm 0.4$ | $9.4 \pm 0.7$ | $125.6 \pm 0.3$ | $\mathbf{1.7 \pm 0.2}$ | $\mathbf{1.013 \pm 0.001}$ |
| | LRP | $108.5 \pm 0.4$ | $96.8 \pm 0.4$ | $9.4 \pm 0.7$ | $125.6 \pm 0.3$ | $\mathbf{1.7 \pm 0.2}$ | $\mathbf{1.013 \pm 0.001}$ |
| | OccPOI | $122.3 \pm 0.8$ | $120.8 \pm 0.2$ | $58 \pm 2$ | $127.4 \pm 0.3$ | $2.6 \pm 0.2$ | $1.09 \pm 0.04$ |
| | 1-Occlusion | $108.5 \pm 0.4$ | $96.7 \pm 0.4$ | $9.4 \pm 0.7$ | $125.6 \pm 0.3$ | $\mathbf{1.7 \pm 0.2}$ | $\mathbf{1.013 \pm 0.001}$ |
| | $m^*$-Occlusion | $108.2 \pm 0.5$ | $\mathbf{95.7 \pm 0.6}$ | $9.0 \pm 0.6$ | $\mathbf{125.3 \pm 0.2}$ | $1.8 \pm 0.2$ | $\mathbf{1.013 \pm 0.001}$ |
| | $2^{\text{nd}}$-order 1-Occlusion | $108.4 \pm 0.4$ | $97.0 \pm 0.4$ | $9.4 \pm 0.7$ | $125.5 \pm 0.3$ | $\mathbf{1.7 \pm 0.2}$ | $\mathbf{1.013 \pm 0.001}$ |
| | $2^{\text{nd}}$-order $m^*$-Occlusion | $108.2 \pm 0.5$ | $95.9 \pm 0.6$ | $\mathbf{9.0 \pm 0.5}$ | $\mathbf{125.3 \pm 0.2}$ | $\mathbf{1.7 \pm 0.2}$ | $\mathbf{1.013 \pm 0.001}$ |
| | ALL (ours) | $\mathbf{107.3 \pm 0.4}$ | $104 \pm 1$ | $9.9 \pm 0.7$ | $125.5 \pm 0.3$ | $1.8 \pm 0.1$ | $1.014 \pm 0.001$ |
| **Rev. DNN occlusion** $\leftarrow$ | GradVis | $125.9 \pm 0.2$ | $127.6 \pm 0.1$ | $122 \pm 1$ | $128.0 \pm 0.3$ | $4.2 \pm 0.4$ | $1.34 \pm 0.07$ |
| | Saliency | $125.8 \pm 0.2$ | $127.4 \pm 0.2$ | $122 \pm 1$ | $128.0 \pm 0.3$ | $5.1 \pm 0.3$ | $1.33 \pm 0.06$ |
| | Input $*$ Grad | $125.7 \pm 0.3$ | $127.5 \pm 0.2$ | $121.9 \pm 0.9$ | $128.1 \pm 0.3$ | $5.2 \pm 0.3$ | $1.34 \pm 0.06$ |
| | LRP | $125.7 \pm 0.3$ | $127.5 \pm 0.2$ | $121.9 \pm 0.9$ | $128.1 \pm 0.3$ | $5.2 \pm 0.3$ | $1.34 \pm 0.06$ |
| | OccPOI | $122.3 \pm 0.4$ | $124.6 \pm 0.2$ | $43 \pm 1$ | $127.0 \pm 0.3$ | $3.6 \pm 0.3$ | $1.09 \pm 0.04$ |
| | 1-Occlusion | $125.8 \pm 0.3$ | $127.4 \pm 0.2$ | $122 \pm 1$ | $128.1 \pm 0.3$ | $5.2 \pm 0.3$ | $1.34 \pm 0.06$ |
| | $m^*$-Occlusion | $126.0 \pm 0.2$ | $127.4 \pm 0.2$ | $121 \pm 1$ | $\mathbf{128.5 \pm 0.2}$ | $5.3 \pm 0.2$ | $1.30 \pm 0.04$ |
| | $2^{\text{nd}}$-order 1-Occlusion | $125.8 \pm 0.3$ | $127.5 \pm 0.2$ | $122.0 \pm 0.9$ | $128.1 \pm 0.3$ | $5.3 \pm 0.2$ | $1.34 \pm 0.06$ |
| | $2^{\text{nd}}$-order $m^*$-Occlusion | $126.1 \pm 0.2$ | $127.4 \pm 0.2$ | $121.3 \pm 0.9$ | $\mathbf{128.5 \pm 0.2}$ | $5.3 \pm 0.2$ | $1.30 \pm 0.04$ |
| | ALL (ours) | $\mathbf{126.4 \pm 0.2}$ | $\mathbf{127.96 \pm 0.06}$ | $\mathbf{125 \pm 1}$ | $128.3 \pm 0.2$ | $\mathbf{5.6 \pm 0.2}$ | $\mathbf{1.39 \pm 0.05}$ |

variable under consideration. In light of the ever-increasing efficacy of deep side-channel attack algorithms and the failure of existing work to detect all leakage they may exploit, our work marks a critical step towards understanding and mitigating the emerging vulnerabilities of cryptographic hardware.

## Broader Impact Statement

The goal of our work is to enhance the security of cryptographic implementations against side-channel attacks by identifying the points in time at which they reveal sensitive information, thereby facilitating targeted defenses and mitigation strategies. We foresee ALL as being a useful complement to widely-adopted parametric leakage localization techniques due to its ability to identify leakage in a 'black box' manner without being limited by the domain knowledge of its users.

Our work is primarily defensive in nature. We do not introduce strategies that directly improve the performance of profiling side-channel attacks, and we solely consider cryptographic datasets which have been made available for research purposes and are already widely studied and understood. Nonetheless, improving the ability to identify and understand the weaknesses of cryptographic systems could potentially benefit attackers as well as defenders. We believe the utility of our work for defense outweighs this risk.

## Acknowledgments

We are thankful to Sakshi Choudhary, Zachary Ellis, Timur Ibrayev, Amogh Joshi, Amitangshu Mukherjee, Deepak Ravikumar, Arjun Roy, and Utkarsh Saxena for helpful discussions and feedback. The authors acknowledge the support from the Purdue Center for Secure Microelectronics Ecosystem – CSME#210205. This work was funded in part by CoCoSys JUMP 2.0 Center, supported by DARPA and SRC.

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

# A    Notation and variable names

See table 3 below for a list of the notation we use, and table 4 for a list of the main variables we define.

Table 3: List of the notation used in our paper.

| Symbol | Description |
|---|---|
| Variable case: $X, x$ | Upper case: random variable, lower-case: actualization |
| $\mathbb{R}, \mathbb{Z}$ | Set of real numbers, set of integers |
| $[a \,..\, b]$ where $a < b$, $a, b \in \mathbb{Z}$ | Interval of integers $\{a, \dots, b\}$ |
| Serif font: $\mathsf{S}$ | Other sets |
| $\mathsf{A} \subseteq \mathsf{B}$ | $\mathsf{A}$ is a non-strict subset of $\mathsf{B}$ |
| $\mathsf{A} \setminus \mathsf{B}$ | Complement of $\mathsf{B}$ in $\mathsf{A}$, i.e. $\mathsf{A} \setminus \mathsf{B} \coloneqq \{x \in \mathsf{A} : x \notin \mathsf{B}\}$ |
| $\mathsf{S}_+$ | Nonnegative elements of $\mathsf{S} \subseteq \mathbb{R}$ |
| $\mathsf{S}_{++}$ | Positive elements of $\mathsf{S} \subseteq \mathbb{R}$ |
| Bold font: $\boldsymbol{x}$ | Vector in $\mathbb{R}^D$ for some $D \in \mathbb{Z}_{++}$ |
| $\boldsymbol{a} \odot \boldsymbol{b}$ | Elementwise product of $\boldsymbol{a}$ and $\boldsymbol{b}$ |
| $f(\boldsymbol{x})$ where $f : \mathbb{R} \to \mathbb{R}$ | Elementwise application of $f$ to $\boldsymbol{x}$ |
| $\nabla_{\boldsymbol{x}} f(\boldsymbol{x}, \dots)$ | Gradient with respect to $\boldsymbol{x}$ |
| $\boldsymbol{x}_{\boldsymbol{\alpha}}$ where $\boldsymbol{\alpha} \in \{0, 1\}^D$ | Sub-vector of $\boldsymbol{x}$ according to $\boldsymbol{\alpha}$: $(x_d : d = 1, \dots, D : \alpha_d = 1)$ |
| $X \sim p$ | $X$ is a random variable with distribution $p$ |
| $p_A$ | The distribution of $A$ |
| $p_{A,B}$ | The joint distribution of $A$ and $B$ |
| $p_{A|B}$ | The conditional distribution of $A$ given $B$ |
| $\boldsymbol{X} \sim p^N$ | $\boldsymbol{X}$ is a random vector with elements $X_1, X_2, \dots \overset{\text{i.i.d.}}{\sim} p$ |
| $\mathcal{N}(\mu; \sigma^2)$ | Normal distribution with mean $\mu$, scale $\sigma$ |
| $\mathcal{U}(\mathsf{S})$ | Uniform distribution over set $\mathsf{S}$ |
| $\mathbb{E}\, f(A, B, \dots)$ | Expectation w.r.t. all random variables $A, B, \dots$ in expression |
| $\mathbb{E}_A\, f(A, B, \dots)$ | Expectation w.r.t. only to $A$ |
| $\mathbb{I}[A; B]$ | Mutual information between $A$ and $B$ |
| $\mathbb{I}[A; B \mid C]$ | Conditional mutual information between $A$ and $B$ given $C$ |
| $A \perp\!\!\!\perp B$, $A \not\perp\!\!\!\perp B$ | $A$ is marginally independent, dependent on $B$ |
| $A \perp\!\!\!\perp B \mid C$, $A \not\perp\!\!\!\perp B \mid C$ | $A$ is conditionally independent, dependent on $B$ given $C$ |

# B    Extended background

Here we provide a high-level overview of the AES algorithm and power side-channel attacks aimed at a machine learning audience. Since our algorithm views the cryptographic algorithm and hardware as a black box to be characterized with data, a deep understanding is not necessary to understand and appreciate our work. Thus, we omit many details and aim to impart an intuitive understanding of these topics. Interested readers may refer to Daemen & Rijmen (2013) for a detailed introduction to the AES algorithm, to Mangard et al. (2007) for a detailed introduction to power side-channel attacks, and to Picek et al. (2023) for a survey of supervised deep learning-based power side-channel attacks on AES implementations. Additionally, note that while for clarity of exposition we focus here on power side-channel attacks on AES implementations, our algorithm requires only a supervised learning-style dataset of side-channel emission traces and metadata which enables computation of the target variable, so is applicable in a more-general setting. We have demonstrated that it works with various target variables on AES, ECC and RSA implementations with both power and EM radiation measurements, and suspect it is relevant in far more contexts.

Table 4: List of the main variables defined in our paper.

| Variable | Description |
|---|---|
| $T \in \mathbb{Z}_{++}$ | Dimensionality of power trace |
| $\boldsymbol{X} \sim p_{\boldsymbol{X}}$ | Power trace (random variable) |
| $\boldsymbol{x} \in \mathbb{R}^T$ | Power trace (actualization) |
| $X_t, x_t$ | Power measurement at time $t$, i.e. $t$-th element of power trace |
| $\mathsf{Y} \subseteq \mathbb{Z}$ | Set of values the targeted variable may take on |
| $Y \sim p_Y$ | Targeted variable (random variable) |
| $y \in \mathsf{Y}$ | Targeted variable (actualization) |
| $\mathsf{D} \subseteq \mathbb{R}^T \times \mathsf{Y}$ | Dataset of power traces/targeted variable pairs, sampled i.i.d. from $p_{\boldsymbol{X},Y}$ |
| $\boldsymbol{\gamma} \in [0,1]^T$ | Occlusion probabilities |
| $\gamma_t$ | Occlusion probability at timestep $t$ |
| $\boldsymbol{\gamma}^* \in [0,1]^T$ | The optimal value of $\boldsymbol{\gamma}$ after solving Eqn. **??** |
| $\tilde{\boldsymbol{\eta}} \in \mathbb{R}^T$ | Unconstrained logits which parameterize $\boldsymbol{\gamma}$ |
| $\mathcal{A}_{\boldsymbol{\gamma}} \sim p_{\mathcal{A}_{\boldsymbol{\gamma}}}$ | Multiplicative binary noise vector parameterized by occlusion probabilities $\boldsymbol{\gamma}$ |
| $\boldsymbol{\alpha} \in \{0,1\}^T$ | Actualization of $\mathcal{A}_{\boldsymbol{\gamma}}$ |
| $c : [0,1) \to \mathbb{R}_+$ | Cost function for occlusion probability elements $\gamma_t$ |
| $C \in \mathbb{R}_{++}$ | Budget for occlusion probabilities: they must satisfy $C = \sum_{t=1}^T c(\gamma_t)$ |
| $\overline{\gamma} \in (0,1)$ | Reparameterized version of $C$ which is more stable w/ data dimensionality |
| $\Phi_{\theta} : \mathbb{R}^T \times [0,1]^T \to \mathbb{R}$ | Noise-conditional neural net w/ weights $\boldsymbol{\theta}$; returns softmax logits for $Y$ |

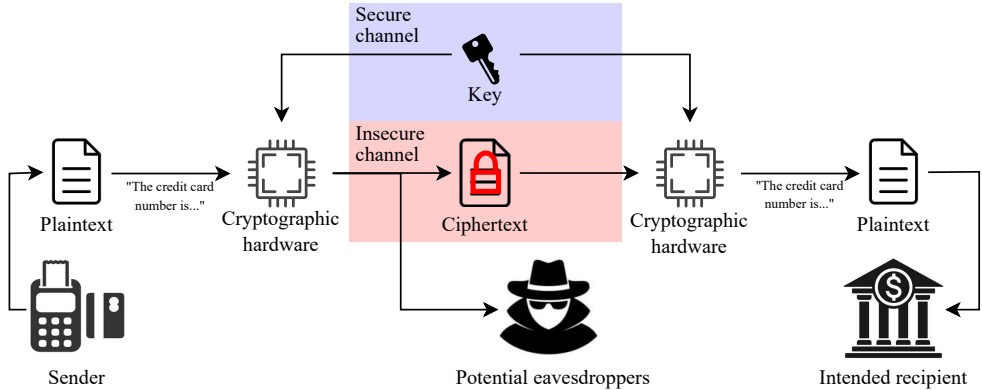

Figure 6: Diagram illustrating the main components of symmetric-key cryptographic algorithms, which enable secure transmission of data over insecure channels where it may be intercepted by eavesdroppers. The data is first partitioned and encoded as a sequence of plaintexts. Each plaintext is transformed into a ciphertext by an invertible function indexed by a cryptographic key. The key is transmitted over a secure channel to intended recipients of the data, allowing them to invert the function and recover the original plaintext. The set of functions is designed so that absent this key, the ciphertext gives no information about the plaintext. Thus, the data remains secure even if eavesdroppers have access to the ciphertext.

## B.1 Cryptographic algorithms

Data is often transmitted over insecure channels which leave it accessible not only to intended recipients, but also to unknown and untrusted parties. For example, when a signal is wirelessly transmitted from one antenna to another, an eavesdropper could set up a third antenna between the two and intercept the signal. Alternately, data stored on a hard drive by one user of a computer may be accessed by a different user. Cryptographic algorithms aim to preserve the privacy of data under such circumstances by transforming it so that it is meaningful only in combination with additional data which is known to its intended recipients but not to the untrusted parties.

We focus here on the ubiquitous advanced encryption standard (AES), which is a symmetric-key cryptographic algorithm. See Fig. 6 for a diagram illustrating the important components of such algorithms. The unencrypted data to be transmitted is encoded and partitioned into a sequence of fixed-length bitstrings called *plaintexts*. The cryptographic algorithm encrypts each plaintext into a *ciphertext* by applying an invertible function from a set of functions indexed by an integer called the *cryptographic key*. This set of functions is designed so that of one were to sample a key and plaintext uniformly at random from the sets of all possible keys and plaintexts, then the plaintext and ciphertext would be marginally independent. Thus, such an algorithm may be used to securely transmit data by ensuring that the sender and recipient of the data know a shared key,[3] and that the key is kept secret from all potential eavesdroppers on the data.

## B.2 Side-channel attacks

Many symmetric-key cryptographic algorithms are believed to be secure in the sense that it is not feasible to determine their cryptographic key by encrypting known plaintexts and observing the resulting ciphertexts. Any such algorithm with a finite number of possible keys is vulnerable to 'brute-force' attacks based on arbitrarily guessing and checking keys until success, but doing so requires checking half of all possible keys in the average case, which is unrealistic for algorithms such as AES which has either $2^{128}$, $2^{192}$, or $2^{256}$ possible keys. To our knowledge the best known such attack against AES reduces the required number of guesses by less than a factor of 8 compared to a naive brute force attack (Mouha, 2021; Tao & Wu, 2015).

However, while algorithms may be secure when considering only their intended inputs and outputs, *hardware executing these algorithms* will inevitably emit measurable physical signals which are statistically associated with their intermediate variables and operations. Examples of such signals include a device's power consumption over time (Kocher et al., 1999), the amount of time it takes to execute a program or instruction (Kocher, 1996; Lipp et al., 2018; Kocher et al., 2019), electromagnetic radiation it emits (Quisquater & Samyde, 2001; Genkin et al., 2016), and sound due to vibrations of its electronic components (Genkin et al., 2014). This phenomenon is called *side-channel leakage*, and can be exploited to determine sensitive data such as a cryptographic key through *side-channel attacks*.

As a simple example of side-channel leakage, consider the following Python function which checks whether a password is correct:

```python
def is_correct(provided_password: str, correct_password: str) -> bool:
    if len(provided_password) != len(correct_password):
        return False
    for i in range(len(provided_password)):
        if provided_password[i] != correct_password[i]:
            return False
    return True
```

Suppose the password consists of $n$ characters, each with $c$ possible values. Consider an attacker seeking to determine the correct password by feeding various guessed passwords until the function returns `True`. Naively, the attacker could simply guess and check all possible $m$-length passwords for $m = 1, \ldots, n$. This would require $O(c^n)$ calls to the function, which would be extremely costly for realistically-large $c$ and $n$. However, an attacker with knowledge of the function's implementation could dramatically reduce this cost by observing that the function's *execution time* depends on `correct_password`. Because the function exits immediately if `len(provided_password) != len(correct_password)`, the attacker can determine the length of `correct_password` in $O(n)$ time by feeding increasing-length guesses to `is_correct` until its execution time increases. Next, because `is_correct` exits the first time it detects an incorrect character, the attacker can sequentially determine each of the characters of `correct_password` by checking all $c$ possible values of each character and noting that the correct value leads to an increase in execution time. Thus, although `is_correct` secure against attackers which use only its intended inputs and outputs, it provides *essentially no security* against attackers which measure its execution time.

---

[3]The key is typically shared using an asymmetric-key cryptographic algorithm such as RSA or ECC. Asymmetric-key cryptography is slow and resource-intensive, so when a sufficiently-large amount of data must be transmitted, it is more-efficient to share the key with an asymmetric-key algorithm and then transmit data using a symmetric-key algorithm than to simply transmit the data with an asymmetric-key algorithm.

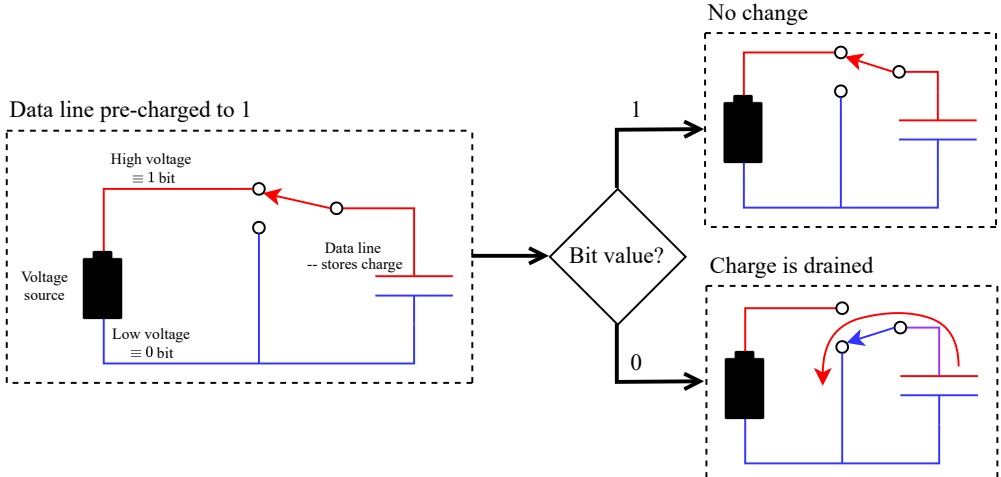

Figure 7: Diagram illustrating one reason there is power side-channel leakage in the device characterized by (Mangard et al., 2007, ch. 4). Data is transmitted over a bus consisting of multiple wires, with one wire representing each bit. Each wire represents a 0 bit as some prescribed 'low' voltage and a 1 bit as a 'high' voltage. Energy is consumed when the voltage of a wire changes from low to high because positive and negative charges, which are attracted to one-another, must be separated to create a high concentration of positive charge on the wire. When 'writing' data to the bus, this particular device first 'pre-charges' all wires to 1, then drains charge from the wires which should represent 0. Thus, because the 0's must be changed to 1's before the next write, energy is consumed in proportion to the number of 0's, thereby creating a statistical association between the device's power consumption and the data it operates on.

In this work we focus on side-channel leakage due to the power consumption over time of a device (as well as EM radiation, which is closely related due to being dominated by the time derivative of power consumption). A device's power consumption is inevitably statistically-associated with the operations it executes and the data it operates on, because these dictate which components are active and the order and manner in which they operate. There are many types of components with different functionality, and components with the same intended functionality are not identical due to imperfect manufacturing processes. These differences impact power consumption. While in general the association between power consumption and data is multifactorial and difficult to describe, in Fig. 7 we illustrate a simple relationship which accounts for a significant portion of the leakage in a device characterized by Mangard et al. (2007).

### B.3 Power side-channel attacks on AES implementations

*Side-channel attacks* are techniques which exploit side-channel leakage to learn sensitive information such as cryptographic keys. There are many categories of attacks, but in this work we focus on a category called *profiled* side-channel attacks on symmetric-key cryptographic algorithms. These attacks assume that the 'attacker' has access to a clone of the actual cryptographic device to be attacked, and the ability to encrypt arbitrary plaintexts with arbitrary cryptographic keys, observe the resulting ciphertexts, and measure the side-channel leakage during encryption. In practice, these assumptions almost certainly overestimate the capabilities of attackers – for example, while in some cases an attacker could plausibly identify the hardware and source code of a cryptographic implementation, purchase copies of this hardware, program them with the source code, and characterize these devices, the nature of the side-channel leakage of these purchased copies would differ from those of the actual device due to PVT (pressure, voltage, temperature) variations (e.g. due to imperfect manufacturing processes, environment, and measurement setup). It has been demonstrated that profiled side-channel attacks can be effective despite this, especially when numerous copies of the target hardware are used for profiling (Das et al., 2019; Danial et al., 2021). Regardless, this type of attack provides an upper bound on the vulnerability of a device to side-channel attacks, which is a useful metric for hardware designers.

While there are diverse types of profiled side-channel attacks, at a high level the following steps encompass the important elements of these attacks:

1. Select some 'sensitive' intermediate variable of the cryptographic algorithm which reveals the cryptographic key (or part of it).

2. Compile a dataset of (side-channel leakage, intermediate variable) pairs by repeatedly randomly selecting a key and plaintext, encrypting the plaintext using the key and recording the resulting ciphertext and side-channel leakage during encryption, and computing the intermediate variable based on knowledge of the cryptographic algorithm.

3. Use supervised learning to train a parametric function approximator to predict intermediate variables from recordings of side-channel leakage during encryption.

4. Measure side-channel leakage during encryptions by the actual target device. Use the trained predictor to predict sensitive variables from side-channel leakage. Potentially, these predictions can be combined to get a better estimate of the key.

In the case of power side-channel attacks on AES, it is generally infeasible to directly target the cryptographic key because care is taken by hardware designers to prevent it from directly influencing power consumption. Instead, it is common to target an intermediate variable which the algorithm directly operates on. A common target is the `SubBytes` output, which is computed as

$$y := \text{Sbox}(k \oplus w) \tag{5}$$

where $k \in \{0,1\}^{n_{\text{bits}}}$ is the key, $w \in \{0,1\}^{n_{\text{bits}}}$ is the plaintext, $n_{\text{bits}} \in \mathbb{Z}_{++}$ is the number of bits of the key and plaintext, $\oplus$ is the bitwise exclusive-or operation, and $\text{Sbox} : \{0,1\}^{n_{\text{bits}}} \to \{0,1\}^{n_{\text{bits}}}$ is an invertible function which is widely known and the same for all AES implementations. Note that if the plaintext is known, the key can be computed as

$$k = \text{Sbox}^{-1}(y) \oplus w. \tag{6}$$

Additionally, it is common to independently target subsets of the bits of the cryptographic key (e.g. the individual bytes). This is reasonable because the common AES target variables are largely leaked by instructions which operate on individual bytes.

### B.3.1 Template attack: example of a classical profiled side-channel attack

In order to underscore the advantage of deep learning over previous side-channel attack algorithms, we will here describe the template attack algorithm of Chari et al. (2003), variations of which are the state-of-the-art non-deep learning based attacks. The attack is based on modeling the joint distribution of power consumption and intermediate variable as a Gaussian mixture model, as described in algorithm 1.

Note that this algorithm assumes that the joint distribution is well-described by a Gaussian mixture model, which may not hold in practice. Additionally, due to the near-cubic runtime of the matrix inversion of each $\Sigma_y$ required to compute the Gaussian density functions, this algorithm requires pruning power traces down to a small number of 'high-leakage' timesteps. Follow-up work (Rechberger & Oswald, 2005) proposed performing principle component analysis on the traces and modeling the coefficients of the top principle components rather than individual timesteps. Nonetheless, these constraints mean that the efficacy of this attack is contingent on simplifying assumptions and judgement of which points are 'leaky' using simple statistical techniques and implementation knowledge, limiting its usefulness as a way for hardware designers to evaluate the amount of side-channel leakage from their device.

### B.3.2 Practical profiled deep learning side-channel attacks on AES implementations

Here we will give a common and concrete setting and method for performing profiled power side-channel attacks on AES implementations, which is used for all of our experiments.

---

**Algorithm 1:** The Gaussian template attack algorithm of Chari et al. (2003)

---

**Input:** Profiling (training) dataset $\mathsf{D} \coloneqq \{(\boldsymbol{x}^{(n)}, y^{(n)} : n \in [1 .. N]\} \subseteq \mathbb{R}^T \times \{0,1\}^{n_{\text{bits}}}$, attack (testing) dataset $\mathsf{D}_{\text{attack}} \coloneqq \{(\boldsymbol{x}_{\text{a}}^{(n)}, w_{\text{a}}^{(n)}) : n \in [1 .. N_{\text{a}}]\} \subseteq \mathbb{R}^T \times \{0,1\}^{n_{\text{bits}}}$, 'points of interest' $\boldsymbol{T}_{\text{poi}} \coloneqq \{t_m : m = 1, \ldots, \tilde{T}\} \subseteq [1 .. T]$

**Output:** Predicted key $k^*$

**1 Function** `get_y` *(k, w)*
**2** $\quad$ **return** $\text{Sbox}(k \oplus w)$ $\qquad\qquad$ // calculate intermediate variable for given key

**3 for** $n \in [1 .. N]$ **do**
**4** $\quad$ $\tilde{\boldsymbol{x}}^{(n)} \leftarrow \left(x_{t_m}^{(n)} : m = 1, \ldots, \tilde{T}\right)$ $\qquad\qquad$ // prune power traces to 'points of interest'

**5 for** $y \in \{0,1\}^{n_{\text{bits}}}$ **do**
$\quad$ // fit a multivariate Gaussian mixture model to the training dataset
**6** $\quad$ $\mathsf{D}_y \leftarrow \left\{\tilde{\boldsymbol{x}}^{(n)} : n \in [1 .. N], y^{(n)} = y\right\}$
**7** $\quad$ $N_y \leftarrow |\mathsf{D}_y|$
**8** $\quad$ $\boldsymbol{\mu}_y \leftarrow \frac{1}{N_y} \sum_{\tilde{\boldsymbol{x}} \in \mathsf{D}_y} \tilde{\boldsymbol{x}}$
**9** $\quad$ $\boldsymbol{\Sigma}_y \leftarrow \frac{1}{N_y-1} \sum_{\tilde{\boldsymbol{x}} \in \mathsf{D}_y} (\tilde{\boldsymbol{x}} - \boldsymbol{\mu}_y)(\tilde{\boldsymbol{x}} - \boldsymbol{\mu}_y)^\top$

**10 for** $n \in [1 .. N_a]$ **do**
**11** $\quad$ $\tilde{\boldsymbol{x}}_{\text{a}}^{(n)} \leftarrow \left(x_{\text{a},t_m}^{(n)} : m = 1, \ldots, \tilde{T}\right)$ $\qquad\qquad$ // prune power traces of attack dataset

$\quad$ // predict key value which maximizes log-likelihood of attack dataset
**12** $k^* \leftarrow \arg\max_{k \in \{0,1\}^{n_{\text{bits}}}} \sum_{n=1}^{N_{\text{a}}} \left[\log \mathcal{N}\left(\tilde{\boldsymbol{x}}^{(n)}; \boldsymbol{\mu}_{\texttt{get\_y}(k, w_{\text{a}}^{(n)})}, \boldsymbol{\Sigma}_{\texttt{get\_y}(k, w_{\text{a}}^{(n)})}\right) + \log N_{\texttt{get\_y}(k, w_{\text{a}}^{(n)})}\right]$
**13 return** $k^*$

---

Consider an AES-128 implementation, which has a 128-bit cryptographic key and plaintext. Typically, attackers target each of the 16 bytes of the key independently rather than attacking the full key at once. This practice tacitly assumes that the bytes of the sensitive variable are statistically-independent given the power trace, which is reasonable because many AES operations (including those which are commonly targeted) are performed independently on the individual bytes. Thus, it is a convenient way to simplify the attack without significantly impacting performance.

Additionally, it is difficult and uncommon to try to directly map power traces to associated cryptographic keys, because great care is taken by hardware designers to ensure that the key does not directly impact power consumption. Instead, attackers generally target 'sensitive' intermediate variables which unavoidably directly impact power consumption and can be combined with the plaintext and ciphertext to learn the key. We consider one such intermediate variable which is referred to as the first `SubBytes` output, and is equal to

$$y \coloneqq \text{Sbox}(k \oplus w), \tag{7}$$

where $k \in \{0,1\}^8$ is one byte of the cryptographic key, $w \in \{0,1\}^8$ is the corresponding byte of the plaintext, $\oplus$ denotes the bitwise exclusive-or operation, and $\text{Sbox} : \{0,1\}^8 \to \{0,1\}^8$ is an invertible function which is publicly-available and the same for all AES implementations. Note that if $w$ is known, as is assumed in the profiled side-channel attack setting, then $k$ can be recovered as

$$k = w \oplus \text{Sbox}^{-1}(y). \tag{8}$$

In the context of profiled power side-channel analysis, one assumes to have a 'profiling' dataset (i.e. a training dataset) and an 'attack' dataset (i.e. a test dataset). Suppose we target $n_{\text{bytes}}$ bytes of the sensitive variable. In our setting, the profiling dataset consists of ordered pairs of power traces and their associated sensitive intermediate variables:

$$\mathsf{D} \coloneqq \left\{(\boldsymbol{x}^{(n)}, y^{(n)}) : n \in [1 .. N]\right\} \subseteq \mathbb{R}^T \times \{0,1\}^{n_{\text{bytes}} \times 8} \tag{9}$$

and the attack dataset consists of ordered pairs of power traces and their associated plaintexts:

$$\mathsf{D}_{\mathrm{a}} := \left\{ (\boldsymbol{x}_{\mathrm{a}}^{(n)}, w_{\mathrm{a}}^{(n)}) : n \in [1 .. N_{\mathrm{a}}] \right\} \subseteq \mathbb{R}^T \times \{0,1\}^{n_{\mathrm{bytes}} \times 8}. \tag{10}$$

Many works prove the concept of their approaches by targeting only a single byte of the sensitive variable. When multiple bytes are targeted, it is common to either train a separate neural network for each byte of the sensitive variable, or to amortize the cost of targeting these bytes by training a single neural network with a shared backbone and a separate head for each byte. In this work we exclusively target single bytes, though it would be straightforward to extend our approach to the multitask learning setting.

Consider a neural network architecture $\Phi : \mathsf{Y} \times \mathbb{R}^T \times \mathbb{R}^P \to \mathbb{R}_+ : (y, \boldsymbol{x}, \boldsymbol{\theta}) \mapsto \Phi(y \mid \boldsymbol{x}; \boldsymbol{\theta})$, where each $\Phi(\cdot \mid \boldsymbol{x}; \boldsymbol{\theta})$ is a probability mass function over $\mathsf{Y}$. In the case of a multi-headed network with each head independently predicting a single byte, we compute this probability mass of $y \in \mathsf{Y}$ as the product of the mass assigned to each of its bytes. We train the network by approximately solving the optimization problem

$$\max_{\boldsymbol{\theta} \in \mathbb{R}^P} \quad \mathcal{L}(\boldsymbol{\theta}) := \frac{1}{N} \sum_{n=1}^{N} \log \Phi(y^{(n)} \mid \boldsymbol{x}^{(n)}; \boldsymbol{\theta}). \tag{11}$$

Given $\hat{\boldsymbol{\theta}} \in \arg\max_{\boldsymbol{\theta} \in \mathbb{R}^P} \mathcal{L}(\boldsymbol{\theta})$, we then identify the key which maximizes our estimated likelihood of our attack dataset and key as follows:

$$\hat{k} \in \arg\max_{k \in \{0,1\}^{n_{\mathrm{bytes}} \times 8}} \quad \sum_{n=1}^{N_{\mathrm{a}}} \log \Phi\left( \left( \mathrm{Sbox}(k_i \oplus w_{\mathrm{a},i}^{(n)}) : i = 1, \ldots, n_{\mathrm{bytes}} \right) \mid \boldsymbol{x}_{\mathrm{a}}^{(n)}; \hat{\boldsymbol{\theta}} \right) \tag{12}$$

where we denote by $k_i$ and $w_i^{(n)}$ the individual bytes of $k$ and $w^{(n)}$.

## C   Extended related work

Here we consider existing work which has been applied to leakage localization in the context of power or EM radiation side-channel analysis. In line with the problem framing given in Sec. 2, we view these methods as functions which map joint emission-target variable distributions $p_{\boldsymbol{X},Y}$ to vectors in $\mathbb{R}^T$ which assign to each emission measurement variable $X_t$ a scalar 'leakiness' measurement. Prior approaches to leakage localization can largely be categorized as either 1) parametric statistics-based methods which check for pairwise associations between $X_t$ and $Y$, or 2) neural net attribution methods which use standard supervised deep learning techniques to train a model $\hat{p}_{Y|\boldsymbol{X}} \approx p_{Y|\boldsymbol{X}}$, then use 'attribution' techniques to estimate the average 'importance' of each feature $X_t$ to the predictions made by the model.

### C.1   First-order parameteric statistics-based methods

In the side-channel attack literature it is common to use parametric first-order statistical methods to localize leakage. In this work we consider the signal to noise ratio (SNR) (Mangard et al., 2007), the sum of squared differences (SOSD) (Chari et al., 2003), and correlation power analysis with a Hamming weight leakage model (CPA) (Brier et al., 2004) due to their popularity and efficacy for 'point of interest' selection (Fan et al., 2014). Below we summarize and discuss these methods.

The SNR is a standard tool for leakage localization and is defined as

$$\mathrm{SNR}(p_{\boldsymbol{X},Y}) := \frac{\mathrm{Var}_{Y \sim p_Y} \mathbb{E}_{\boldsymbol{X} \sim p_{\boldsymbol{X}|Y}}[\boldsymbol{X}]}{\mathbb{E}_{Y \sim p_Y} \mathrm{Var}_{\boldsymbol{X} \sim p_{\boldsymbol{X}|Y}}(\boldsymbol{X})}. \tag{13}$$

The SOSD method was introduced as a point of interest (feature) selection technique for the Gaussian template attack, a parametric side-channel attack based on modeling emission measurements as a multivariate Gaussian mixture model with a component corresponding to each possible value of the target variable,

then using Bayes' rule to estimate the conditional distribution of the target variable given the emission measurements. The SOSD is defined as

$$\text{sosd}(p_{\boldsymbol{X},Y}) := \sum_{y \in \mathsf{Y}} \sum_{y' \in \mathsf{Y}} \left( \mathbb{E}_{\boldsymbol{X} \sim p_{\boldsymbol{X}|Y}(\cdot|y)}[\boldsymbol{X}] - \mathbb{E}_{\boldsymbol{X} \sim p_{\boldsymbol{X}|Y}(\cdot|y')}[\boldsymbol{X}] \right)^2. \tag{14}$$

CPA is based on the assumption that power consumption is a noisy linear function of the Hamming weight of the target variable, which is a useful model for certain devices. It is defined as the elementwise Pearson correlation between emission measurements and the target variable's Hamming weight:

$$\text{cpa}(p_{\boldsymbol{X},Y}) := \frac{\mathbb{E}[(\boldsymbol{X} - \mathbb{E}[\boldsymbol{X}])(\text{HW}(Y) - \mathbb{E}[\text{HW}(Y)])]}{\sqrt{\text{Var}(\boldsymbol{X}) \text{Var}(\text{HW}(Y))}} \tag{15}$$

where $\text{HW} : [0 .. 2^d - 1] \to [0 .. d]$ maps integers to the sum of their bits when writing them as an unsigned integer.

While techniques such as these are invaluable due to their simplicity, interpretability and low cost, they have major shortcomings. Notably, they consider they are sensitive only to pairwise associations between single emission measurements and the target variable and will consider the measurement $X_t$ to be non-leaky when it is nonleaky in isolation but gives exploitable information *when combined* with $X_\tau$ for some $\tau \neq t$, i.e. when $\mathbb{I}[Y; X_t] = 0$ but $\exists \tau \neq t$ such that $\mathbb{I}[Y; X_t \mid X_\tau] > 0$.

Additionally, these techniques make strong assumptions about the nature of $p_{\boldsymbol{X},Y}$ which have generally been observed to hold in practice, but nonetheless introduce the risk of failing to detect leaking measurements. SNR and SOSD are sensitive only to the influence of $Y$ on the mean of $\boldsymbol{X}$, and would fail to identify $X_t$ is leaking if $Y$ changes its distribution while leaving its mean unchanged (e.g. if the variance of $X_t$ changes with $Y$). CPA is sensitive only to associations between $X_t$ and the Hamming weight of $Y$, and additionally assumes a linear relationship between these variables.

While the present work concerns mainly 'black box' leakage localization algorithms which make minimal assumptions about the cryptographic implementation being evaluated, in practice these parametric methods are often employed as tools in white-box analyses of implementations. For example, in our work we consider the ASCADv1 datasets (Benadjila et al., 2020), which use a Boolean masking countermeasure and thus have mainly second-order leakage. This renders the above methods ineffective when directly analyzing leakage of their canonical target variable. However, Egger et al. (2022) identified 4 *pairs* of internal AES variables which individually have first-order leakage and may be combined to determine the target variable. Thus, if one is aware *a priori* that these variables leak and has access to the internal randomly-generated Boolean mask variables, they may individually analyze leakage of these variables with the above methods and accumulate the results. We use such an approach to compute 'ground truth' leakiness measurements when running experiments on the ASCADv1 datasets. We emphasize that this type of analysis is challenging and error-prone: multiple leaking variable identified by Egger et al. (2022) were unknown or overlooked by Benadjila et al. (2020), who introduced the dataset. This underscores the importance of black box techniques such as ours to supplement white box analysis.

## C.2 Neural net attribution methods

There is a great deal of prior work on localizing leakage based on interpretability techniques to determine the relative importance of input features to a deep neural net which has been trained to model $\hat{p}_{Y|\boldsymbol{X}} \approx p_{Y|\boldsymbol{X}}$ using standard supervised deep learning techniques (Masure et al., 2019; Hettwer et al., 2020; Jin et al., 2020; Zaid et al., 2020; Wouters et al., 2020; van der Valk et al., 2021; Wu & Johnson, 2021; Golder et al., 2022; Li et al., 2022; Perin et al., 2022; Schamberger et al., 2023; Yap et al., 2023; Li et al., 2024; Yap et al., 2025). As baselines we consider the recent works Yap et al. (2025); Schamberger et al. (2023) as well as a variety of older neural net attribution techniques which were compared in Masure et al. (2019); Hettwer et al. (2020); Wouters et al. (2020). Note that our choice of deep learning baselines subsumes those of Yap et al. (2025); Schamberger et al. (2023). Since these methods all 'interpret' a trained deep neural net, we view them as functions mapping the data distribution $p_{\boldsymbol{X},Y}$ as well as a model $\hat{p}_{Y|\boldsymbol{X}}$ s.t. softmax $\hat{p}_{Y|\boldsymbol{X}} \approx p_{Y|\boldsymbol{X}}$ to a vector

of leakiness estimates in $\mathbb{R}^T$. Here we summarize and discuss the works Masure et al. (2019); Hettwer et al. (2020); Wouters et al. (2020); Schamberger et al. (2023); Yap et al. (2025) which we consider as baselines.

To our knowledge Masure et al. (2019) were the first to explore neural net interpretability for leakage localization, proposing the Saliency-like GradVis leakage assessment, defined as

$$\text{GradVis}(p_{\boldsymbol{X},Y}; \hat{p}_{Y|\boldsymbol{X}}) \coloneqq \mathbb{E}_{\boldsymbol{X},Y} \left| -\nabla_{\boldsymbol{x}} \log\text{softmax}\, \hat{p}_{Y|\boldsymbol{X}}(Y \mid \boldsymbol{x})|_{\boldsymbol{x}=\boldsymbol{X}} \right|. \tag{16}$$

Hettwer et al. (2020) subsequently compared the 1-Occlusion (Zeiler & Fergus, 2014), Saliency (Simonyan et al., 2014), and layerwise relevance propagation (LRP) (Bach et al., 2015) as leakage localization techniques. The 1-Occlusion technique is based on computing the size of change in the model's prediction as each individual input feature is 'occluded' (replaced by 0), and is defined as

$$\text{1-Occlusion}(p_{\boldsymbol{X},Y}; \hat{p}_{Y|\boldsymbol{X}}) \coloneqq \mathbb{E}_{\boldsymbol{X},Y} \left( \left| \hat{p}_{Y|\boldsymbol{X}}(Y \mid \boldsymbol{X}) - \hat{p}_{Y|\boldsymbol{X}}(Y \mid (\boldsymbol{1} - \boldsymbol{I}_t) \odot \boldsymbol{X}) \right| : t = 1, \ldots, T \right) \tag{17}$$

where $\boldsymbol{I}_t$ denotes the vector in $\mathbb{R}^T$ with element $t$ equal to 1 and all other elements equal to 0. Saliency is defined as

$$\text{Saliency}(p_{\boldsymbol{X},Y}; \hat{p}_{Y|\boldsymbol{X}}) \coloneqq \mathbb{E}_{\boldsymbol{X},Y} \left| \nabla_{\boldsymbol{x}} \hat{p}_{Y|\boldsymbol{X}}(Y \mid \boldsymbol{x})|_{\boldsymbol{x}=\boldsymbol{X}} \right|. \tag{18}$$

LRP (Bach et al., 2015) is a gradient-based explainability technique which is more-complicated than the above, and we refer readers to Bach et al. (2015) for an explanation. Wouters et al. (2020) applied the Input $*$ Grad method (Shrikumar et al., 2017) to leakage localization; as its name suggests, this method is defined as

$$\text{Input} * \text{Grad}(p_{\boldsymbol{X},Y}; \hat{p}_{Y|\boldsymbol{X}}) \coloneqq \mathbb{E}_{\boldsymbol{X},Y} \left| \boldsymbol{X} \odot \nabla_{\boldsymbol{x}} \hat{p}_{Y|\boldsymbol{X}}(Y \mid \boldsymbol{x})|_{\boldsymbol{x}=\boldsymbol{X}} \right|. \tag{19}$$

We find that all these techniques have a tendency to incorrectly assign low leakiness to certain measurements, particularly in scenarios where lots of features have significant leakage. We suspect that a primary reason for this is that these methods rely on perturbing the input $x_t$ to the function $\hat{p}_{Y|\boldsymbol{X}}(y \mid x_1, \ldots, x_t, \ldots, x_T)$, but when $Y$ is almost entirely determined by $X_\tau$ for $\tau \neq t$, it becomes nearly independent of $X_t$ conditioned on $\{X_1, \ldots, X_T\} \setminus \{X_t\}$, i.e. $\hat{p}_{Y|\boldsymbol{X}}(y \mid x_\tau : \tau = 1, \ldots, T) \approx \hat{p}_{Y|\boldsymbol{X}}(y \mid x_\tau : \tau = 1, \ldots, t-1, t+1, \ldots, T)$. We demonstrate this phenomenon with a simple Gaussian mixture model setting in Sec. C.3.

We also consider the recent works Schamberger et al. (2023); Yap et al. (2025) which estimate leakiness by perturbing many inputs simultaneously to a classifier, and do not necessarily suffer from the same issue. Schamberger et al. (2023) presents the $m$-Occlusion technique, which is like 1-Occlusion except that it occludes $m$-diameter windows rather than single points. This could plausibly overcome the aforementioned issue if the 'redundant' points are temporally-local. However, it has an undesirable 'smoothing' effect where the estimated leakiness of a single point is tied to those of nearby points. Schamberger et al. (2023) also proposes to use 2$^\text{nd}$-order $m$-Occlusion to analyze leakiness, where *pairs* of windows are occluded rather than only individual windows. This is computationally-expensive because it requires $\Theta(T^2)$ passes through the dataset where $T$ is the data dimensionality. Additionally, Schamberger et al. (2023) proposes it as a means to determine whether a measurement has first-order leakage or is part of a second-order leaking pair, and does not explore its use for single-measurement leakiness estimation. In our experiments we find it only marginally-better than 1-Occlusion for this task, and not worth the significantly-higher computational cost. Yap et al. (2025) proposes the OccPOI technique, which aims to identify a non-unique minimal set of measurements sufficient for a neural net to attain some chosen classification performance when all other measurements are occluded. This differs from our other considered methods in that it does not assign a leakiness estimate to every measurement, and we find that it is ill-suited for our leakage localization task and performance metrics. Additionally, it is computationally-expensive to run, requiring $\Omega(T)$ *non-parallelizable* passes through the dataset.

## C.3 Numerical experiment illustrating conditional mutual information decay when many redundant leaking measurements are present

Here we provide a simple numerical experiment to illustrate conditional mutual information decay. Consider random variables $Y \sim \mathcal{U}\{-1, 1\}$ and $X_1, \ldots, X_n \overset{\text{i.i.d.}}{\sim} \mathcal{N}(Y, \sigma^2)$. In Fig. 8 we plot the quantity $\mathbb{I}[Y; X_n \mid$

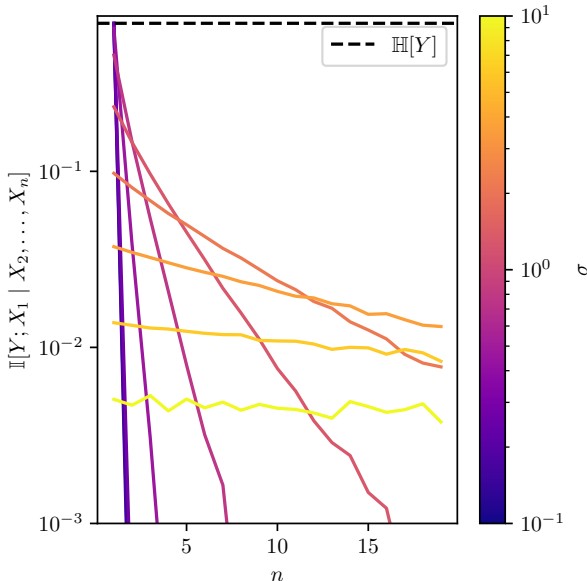

Figure 8: A numerical experiment evaluating the scaling behavior of $\mathbb{I}[Y; X_n \mid X_1, \ldots, X_{n-1}]$ vs. $n$ for various values of $\sigma$, where $Y \sim \mathcal{U}\{-1, 1\}$ and $X_1, \ldots, X_n \stackrel{\text{i.i.d.}}{\sim} \mathcal{N}(Y, \sigma^2)$. Observe that for small $\sigma$, each $X_i$ is approximately a point mass on $Y$ and we have $\mathbb{I}[Y; X_1] \approx \mathbb{H}[Y]$ and $\mathbb{I}[Y; X_n \mid X_1, \ldots, X_{n-1}] \approx 0$ for all $n > 1$. When $\sigma$ is large, each $X_i$ gives us little information about $Y$ and we have $\mathbb{I}[Y; X_n \mid X_1, \ldots, X_{n-1}] \ll \mathbb{H}[Y]$ approximately constant with $n$ sufficiently small.

$X_1, \ldots, X_{n-1}]$ vs. $n$. We see that for various values of $\sigma$ the quantity decays rapidly with $n$, and appears to be well-described by the function $\mathbb{I}[Y; X_1 \mid X_2, \ldots, X_n] \approx Ik^n$ for some $I \in \mathbb{R}_+$, $k \in (0, 1)$. Informally, the fact that conditional mutual information decays with $n$ makes sense for the following reason: each successive $X_i$ can be viewed as an independent 'noisy observation' of $Y$ which reduces uncertainty about it without fully determining it. For all $n$ we have $0 \leq \mathbb{I}[Y; X_1, \ldots, X_n] = \mathbb{I}[Y; X_1] + \mathbb{I}[Y; X_2 \mid X_1] + \ldots \mathbb{I}[Y; X_n \mid X_1, \ldots, X_{n-1}] \leq \mathbb{H}[Y]$. If we assume that each $\mathbb{I}[Y; X_m \mid X_1, \ldots, X_{m-1}]$ is nonnegative, then for their sum to stay bounded they must converge to 0.

## D Extended method with derivations

Given $\boldsymbol{X}, Y \sim p_{\boldsymbol{X},Y}$ as defined in section 2, where $\boldsymbol{X} \coloneqq (X_1, \ldots, X_T)$, we seek to assign to each timestep $t$ a scalar $\gamma_t^*$ indicating the 'amount of leakage' about $Y$ due to $X_t$. All of the quantities $\mathbb{I}[X; X_t \mid \mathsf{S}]$ for $\mathsf{S} \subseteq \{X_1, \ldots, X_T\} \setminus \{X_t\}$ are relevant to the 'leakiness' of $X_t$, but it is not clear how to weight these $2^T$ quantities into a single scalar measurement. Most prior work simply ignores most of these quantities: the first-order parametric methods (Mangard et al., 2007; Brier et al., 2004; Chari et al., 2003) consider only the pairwise terms $\mathbb{I}[Y; X_t]$, and GradVis (Masure et al., 2019), Saliency (Simonyan et al., 2014; Hettwer et al., 2020), 1-Occlusion (Zeiler & Fergus, 2014; Hettwer et al., 2020), LRP (Bach et al., 2015; Hettwer et al., 2020), Input $\ast$ Grad (Shrikumar et al., 2017; Wouters et al., 2020) may loosely be viewed as computing proxies for $\mathbb{I}[Y; X_t \mid X_{\tau \neq t}]$. While $m$-Occlusion, $2^{\text{nd}}$-order $m$-Occlusion (Schamberger et al., 2023), and OccPOI (Yap et al., 2025) are sensitive to more of these terms, they still ignore almost all of them in an *ad hoc* manner.

In this section we propose a constrained optimization problem which implicitly defines an intuitively-reasonable definition of $\gamma_t^*$ which is sensitive to $\mathbb{I}[Y; X_t \mid \mathsf{S}]$ for *all* $\mathsf{S} \subseteq \{X_1, \ldots, X_T\} \setminus \{X_t\}$. We then propose an adversarial deep learning algorithm which lets us approximately solve it by modeling all the conditional distributions $p_{Y|\mathsf{S}}$ in an amortized manner which emphasizes those with a large impact on the objective. While our objective is a sum over $2^T$ occlusion mask-like values, in practice we find we can efficiently optimize it using the reparameterization trick (Kingma & Welling, 2014; Rezende et al., 2014) with a CONCRETE-like

relaxation (Jang et al., 2017; Maddison et al., 2017) of our objective. This lets us exploit first-order gradient information, in contrast to 'hard' occlusion-based methods such as Schamberger et al. (2023); Yap et al. (2025) which leverage only zeroth-order information.

## D.1 Optimization problem

We define a vector $\boldsymbol{\gamma} \in [0,1]^T$ which we name the *occlusion probabilities*. We use $\boldsymbol{\gamma}$ to parameterize a distribution over binary vectors in $\{0,1\}^T$ as follows:

$$\boldsymbol{\mathcal{A}_\gamma} \sim p_{\boldsymbol{\mathcal{A}_\gamma}} \quad \text{where} \quad \mathcal{A}_{\gamma,t} = \begin{cases} 1 & \text{with probability } 1 - \gamma_t \\ 0 & \text{with probability } \gamma_t, \end{cases} \tag{20}$$

i.e. $\boldsymbol{\mathcal{A}_\gamma}$ is a vector of independent Bernoulli random variables where the $t$-th element has parameter $p = 1 - \gamma_t$. For arbitrary vectors $\boldsymbol{x} \in \mathbb{R}^T$, $\boldsymbol{\alpha} \in \{0,1\}^T$, let us denote $\boldsymbol{x_\alpha} \coloneqq (x_t : t = 1, \ldots, T : \alpha_t = 1)$, i.e. the sub-vector of $\boldsymbol{x}$ containing its elements for which the corresponding element of $\boldsymbol{\alpha}$ is 1. We can accordingly use $\boldsymbol{\mathcal{A}_\gamma}$ to obtain random sub-vectors $\boldsymbol{X_{\mathcal{A}_\gamma}}$ of $\boldsymbol{X}$. Note that $\gamma_t$ denotes the probability that $X_t$ will *not* be an element of $\boldsymbol{X_{\mathcal{A}_\gamma}}$ (hence, 'occlusion probability').

We assign to each element of $\boldsymbol{\gamma}$ a 'cost', defined as

$$c : [0,1] \to \mathbb{R}_+ : x \mapsto \begin{cases} \frac{x}{1-x} & x < 1 \\ \infty & x = 1 \end{cases}. \tag{21}$$

We seek to solve the constrained optimization problem

$$\min_{\boldsymbol{\gamma} \in [0,1]^T} \quad \mathcal{L}_{\text{ideal}}(\boldsymbol{\gamma}) \coloneqq \mathbb{I}[Y; \boldsymbol{X_{\mathcal{A}_\gamma}} \mid \boldsymbol{\mathcal{A}_\gamma}] \quad \text{such that} \quad \sum_{t=1}^T c(\gamma_t) = C \tag{22}$$

for hyperparameter $C > 0$. Note that $c$ is strictly-increasing with $c(0) = 0$ and $\lim_{\substack{x \to 1 \\ x < 1}} c(x) = \infty$ so that for finite $C$ any optimal $\boldsymbol{\gamma}$ will be in $[0,1)^T$, and increasing some $\gamma_t$ necessarily entails reducing some other $\gamma_\tau$, $\tau \neq t$. Additionally, for each $t$ we can re-write our objective as

$$\mathcal{L}_{\text{ideal}}(\boldsymbol{\gamma}) = \sum_{\boldsymbol{\alpha} \in \{0,1\}^T} p_{\boldsymbol{\mathcal{A}_\gamma}}(\boldsymbol{\alpha}) \, \mathbb{I}[Y; \boldsymbol{X_\alpha}] \tag{23}$$

$$= \sum_{\substack{\boldsymbol{\alpha} \in \{0,1\}^T \\ \alpha_t = 0}} p_{\boldsymbol{\mathcal{A}_{\gamma,-t}}}(\boldsymbol{\alpha}_{-t}) \left[ (1 - \gamma_t) \, \mathbb{I}[Y; X_t, \boldsymbol{X_\alpha}] + \gamma_t \, \mathbb{I}[Y; \boldsymbol{X_\alpha}] \right] \tag{24}$$

$$= \sum_{\substack{\boldsymbol{\alpha} \in \{0,1\}^T \\ \alpha_t = 0}} p_{\boldsymbol{\mathcal{A}_{\gamma,-t}}}(\boldsymbol{\alpha}_{-t}) \left[ \mathbb{I}[Y; X_t, \boldsymbol{X_\alpha}] - \gamma_t \, \mathbb{I}[Y; X_t \mid \boldsymbol{X_\alpha}] \right], \tag{25}$$

which implies

$$\frac{\partial \mathcal{L}_{\text{ideal}}(\boldsymbol{\gamma})}{\partial \gamma_t} = - \sum_{\substack{\boldsymbol{\alpha} \in \{0,1\}^T \\ \alpha_t = 0}} p_{\boldsymbol{\mathcal{A}_{\gamma,-t}}}(\boldsymbol{\alpha}_{-t}) \, \mathbb{I}[Y; X_t \mid \boldsymbol{X_\alpha}]. \tag{26}$$

## D.2 Estimating mutual information with deep neural nets

We cannot solve equation 22 directly because we lack an expression for $p_{\boldsymbol{X},Y}$. Here we derive an equivalent optimization problem which uses deep learning to characterize $p_{\boldsymbol{X},Y}$ using data.

Consider the family $\{\Phi_{\boldsymbol{\alpha}}\}_{\boldsymbol{\alpha} \in \{0,1\}^T}$ with each element a deep neural net

$$\Phi_{\boldsymbol{\alpha}} : \mathsf{Y} \times \mathbb{R}^{\sum_{t=1}^T \alpha_t} \times \mathbb{R}^P \to [0,1] : (y, \boldsymbol{x_\alpha}, \boldsymbol{\theta}) \mapsto \Phi_{\boldsymbol{\alpha}}(y \mid \boldsymbol{x_\alpha}; \boldsymbol{\theta}). \tag{27}$$

We assume each $\Phi_{\boldsymbol{\alpha}}(\cdot \mid \boldsymbol{x}; \boldsymbol{\theta})$ is a probability mass function over $\mathsf{Y}$ (e.g. the neural net has a softmax output activation). We define the optimization problem

$$\min_{\boldsymbol{\gamma} \in [0,1]^T} \max_{\boldsymbol{\theta} \in \mathbb{R}^P} \quad \mathcal{L}_{\mathrm{adv}}(\boldsymbol{\gamma}, \boldsymbol{\theta}) := \mathbb{E} \log \Phi_{\mathcal{A}_{\boldsymbol{\gamma}}}(Y \mid \boldsymbol{X}_{\mathcal{A}_{\boldsymbol{\gamma}}}; \boldsymbol{\theta}) \quad \text{such that} \quad \sum_{t=1}^{T} c(\gamma_t) = C. \tag{28}$$

**Proposition D.1.** *Consider the objective function $\mathcal{L}_{\mathrm{adv}}$ of equation 28. Suppose there exists some $\boldsymbol{\theta}^* \in \mathbb{R}^P$ such that $\Phi_{\boldsymbol{\alpha}}(y \mid \boldsymbol{x}_{\boldsymbol{\alpha}}; \boldsymbol{\theta}^*) = p_{Y|\boldsymbol{X}_{\boldsymbol{\alpha}}}(y \mid \boldsymbol{x}_{\boldsymbol{\alpha}})$ for all $\boldsymbol{\alpha} \in \{0,1\}^T$, $\boldsymbol{x} \in \mathbb{R}^T$, $y \in \mathsf{Y}$. Then*

$$\boldsymbol{\theta}^* \in \arg\max_{\boldsymbol{\theta} \in \mathbb{R}^P} \mathcal{L}_{\mathrm{adv}}(\boldsymbol{\gamma}, \boldsymbol{\theta}) \quad \forall \boldsymbol{\gamma} \in [0,1]^T. \tag{29}$$

*Furthermore, for all $y \in \mathsf{Y}$ and for all $\boldsymbol{\gamma} \in [0,1]^T$, $\boldsymbol{\alpha} \in \{0,1\}^T$ such that $p_{\mathcal{A}_{\boldsymbol{\gamma}}}(\boldsymbol{\alpha}) > 0$,*

$$\Phi_{\boldsymbol{\alpha}}(y \mid \boldsymbol{x}_{\boldsymbol{\alpha}}; \hat{\boldsymbol{\theta}}) = p_{Y|\boldsymbol{X}_{\boldsymbol{\alpha}}}(y \mid \boldsymbol{x}_{\boldsymbol{\alpha}}) \quad p_{\boldsymbol{X}}\text{-almost surely} \quad \forall \hat{\boldsymbol{\theta}} \in \arg\min_{\boldsymbol{\theta} \in \mathbb{R}^P} \mathcal{L}_{\mathrm{adv}}(\boldsymbol{\gamma}, \boldsymbol{\theta}). \tag{30}$$

*Proof.* Note that since each $\Phi_{\boldsymbol{\alpha}}(\cdot \mid \boldsymbol{x}, \boldsymbol{\theta})$ is a probability mass function over $\mathsf{Y}$, by Gibbs' inequality we have

$$\mathbb{E} \log \Phi_{\boldsymbol{\alpha}}(Y \mid \boldsymbol{X}_{\boldsymbol{\alpha}}; \boldsymbol{\theta}) \le \mathbb{E} \log p_{Y|\boldsymbol{X}_{\boldsymbol{\alpha}}}(Y \mid \boldsymbol{X}_{\boldsymbol{\alpha}}) \quad \forall \boldsymbol{\alpha} \in \{0,1\}^T, \boldsymbol{\theta} \in \mathbb{R}^P. \tag{31}$$

Thus,

$$\mathcal{L}_{\mathrm{adv}}(\boldsymbol{\gamma}, \boldsymbol{\theta}^*) \ge \mathcal{L}_{\mathrm{adv}}(\boldsymbol{\gamma}, \boldsymbol{\theta}) \quad \forall \boldsymbol{\theta} \in \mathbb{R}^P, \boldsymbol{\gamma} \in [0,1]^T, \tag{32}$$

which implies the first claim.

Next, consider some fixed $\boldsymbol{\gamma} \in [0,1]^T$ and $\hat{\boldsymbol{\theta}} \in \arg\min_{\boldsymbol{\theta} \in \mathbb{R}^P} \mathcal{L}_{\mathrm{adv}}(\boldsymbol{\gamma}, \boldsymbol{\theta})$. We must have $\mathcal{L}_{\mathrm{adv}}(\boldsymbol{\gamma}, \hat{\boldsymbol{\theta}}) = \mathcal{L}_{\mathrm{adv}}(\boldsymbol{\gamma}, \boldsymbol{\theta}^*)$. Thus,

$$0 = \mathcal{L}_{\mathrm{adv}}(\boldsymbol{\gamma}, \boldsymbol{\theta}^*) - \mathcal{L}_{\mathrm{adv}}(\boldsymbol{\gamma}, \hat{\boldsymbol{\theta}}) \tag{33}$$

$$= \mathbb{E}\left[\log p_{Y|\boldsymbol{X}_{\boldsymbol{\alpha}}}(Y \mid \boldsymbol{X}_{\boldsymbol{\alpha}}) - \log \Phi_{\boldsymbol{\alpha}}(Y \mid \boldsymbol{X}_{\boldsymbol{\alpha}}; \hat{\boldsymbol{\theta}})\right] \tag{34}$$

$$= \sum_{\boldsymbol{\alpha} \in \{0,1\}^T} p_{\mathcal{A}_{\boldsymbol{\gamma}}}(\boldsymbol{\alpha}) \mathbb{E}\left[\log p_{Y|\boldsymbol{X}_{\boldsymbol{\alpha}}}(Y \mid \boldsymbol{X}_{\boldsymbol{\alpha}}) - \log \Phi_{\boldsymbol{\alpha}}(Y \mid \boldsymbol{X}_{\boldsymbol{\alpha}}; \hat{\boldsymbol{\theta}})\right]. \tag{35}$$

By Gibbs' inequality, each of the expectations in the summation is nonnegative, which implies that whenever $p_{\mathcal{A}_{\boldsymbol{\gamma}}}(\boldsymbol{\alpha}) > 0$ we must have

$$0 = \mathbb{E}\left[\log p_{Y|\boldsymbol{X}_{\boldsymbol{\alpha}}}(Y \mid \boldsymbol{X}_{\boldsymbol{\alpha}}) - \log \Phi_{\boldsymbol{\alpha}}(Y \mid \boldsymbol{X}_{\boldsymbol{\alpha}}; \hat{\boldsymbol{\theta}})\right] \tag{36}$$

$$= \int_{\mathbb{R}^{\sum_{t=1}^{T} \alpha_t}} p_{\boldsymbol{X}_{\boldsymbol{\alpha}}}(\boldsymbol{x}_{\boldsymbol{\alpha}}) \mathbb{KL}\left[p_{Y|\boldsymbol{X}_{\boldsymbol{\alpha}}}(\cdot \mid \boldsymbol{x}_{\boldsymbol{\alpha}}) \parallel \Phi_{\boldsymbol{\alpha}}(\cdot \mid \boldsymbol{x}_{\boldsymbol{\alpha}}; \hat{\boldsymbol{\theta}})\right] d\boldsymbol{x}_{\boldsymbol{\alpha}}. \tag{37}$$

Since $\mathbb{KL}\left[p_{Y|\boldsymbol{X}_{\boldsymbol{\alpha}}}(\cdot \mid \boldsymbol{x}_{\boldsymbol{\alpha}}) \parallel \Phi_{\boldsymbol{\alpha}}(\cdot \mid \boldsymbol{x}_{\boldsymbol{\alpha}}; \hat{\boldsymbol{\theta}})\right] \ge 0$ with equality if and only if $p_{Y|\boldsymbol{X}_{\boldsymbol{\alpha}}}(y \mid \boldsymbol{x}_{\boldsymbol{\alpha}}) = \Phi_{\boldsymbol{\alpha}}(y \mid \boldsymbol{x}_{\boldsymbol{\alpha}}; \hat{\boldsymbol{\theta}})$ $\forall y \in \mathsf{Y}$, this must be the case except possibly for $\boldsymbol{x} \in \mathbb{R}^T$ where

$$\int_{\{\boldsymbol{x}_{\boldsymbol{\alpha}} : \boldsymbol{x} \in \mathbb{R}^T\}} p_{\boldsymbol{X}_{\boldsymbol{\alpha}}}(\boldsymbol{x}_{\boldsymbol{\alpha}}) d\boldsymbol{x}_{\boldsymbol{\alpha}} = 0 \implies \int_{\mathbb{R}^T} p_{\boldsymbol{X}}(\boldsymbol{x}) d\boldsymbol{x} = 0. \tag{38}$$

This implies the second claim. $\qquad \square$

**Corollary D.2.** *Under the assumptions of Proposition D.1, equations 22 and 28 are equivalent.*

*Proof.* Observe that for any $\boldsymbol{\gamma} \in [0,1]^T$,

$$\max_{\boldsymbol{\theta} \in \mathbb{R}^P} \mathcal{L}_{\text{adv}}(\boldsymbol{\gamma}, \boldsymbol{\theta}) = \max_{\boldsymbol{\theta} \in \mathbb{R}^P} \mathbb{E} \log \Phi_{\mathcal{A}_{\boldsymbol{\gamma}}}(Y \mid \boldsymbol{X}_{\mathcal{A}_{\boldsymbol{\gamma}}}(Y \mid \boldsymbol{X}_{\mathcal{A}_{\boldsymbol{\gamma}}}; \boldsymbol{\theta}) \tag{39}$$

$$= \sum_{\boldsymbol{\alpha} \in \{0,1\}^T} p_{\mathcal{A}_{\boldsymbol{\gamma}}}(\boldsymbol{\alpha}) \, \mathbb{E} \log p_{Y \mid \boldsymbol{X}_{\boldsymbol{\alpha}}}(Y \mid \boldsymbol{X}_{\boldsymbol{\alpha}}) \quad \text{by Prop. D.1} \tag{40}$$

$$= - \sum_{\boldsymbol{\alpha} \in \{0,1\}^T} p_{\mathcal{A}_{\boldsymbol{\gamma}}}(\boldsymbol{\alpha}) \, \mathbb{H}[Y \mid \boldsymbol{X}_{\boldsymbol{\alpha}}] \tag{41}$$

$$\equiv \sum_{\boldsymbol{\alpha} \in \{0,1\}^T} p_{\mathcal{A}_{\boldsymbol{\gamma}}}(\boldsymbol{\alpha}) \left[ \mathbb{H}[Y] - \mathbb{H}[Y \mid \boldsymbol{X}_{\boldsymbol{\alpha}}] \right] \quad \text{because } \mathbb{H}[Y] \text{ is not a function of } \boldsymbol{\gamma} \tag{42}$$

$$= \sum_{\boldsymbol{\alpha} \in \{0,1\}^T} p_{\mathcal{A}_{\boldsymbol{\gamma}}}(\boldsymbol{\alpha}) \, \mathbb{I}[Y; \boldsymbol{X}_{\boldsymbol{\alpha}}] \tag{43}$$

$$= \mathbb{I}[Y; \boldsymbol{X}_{\mathcal{A}_{\boldsymbol{\gamma}}} \mid \mathcal{A}_{\boldsymbol{\gamma}}] \tag{44}$$

$$= \mathcal{L}_{\text{ideal}}(\boldsymbol{\gamma}). \tag{45}$$

This implies the result. $\qquad \square$

**Corollary D.3.** *Suppose the assumptions of Proposition D.1 are satisfied, and let $\hat{\boldsymbol{\theta}} \in \arg\min_{\boldsymbol{\theta} \in \mathbb{R}^P} \mathcal{L}_{\text{adv}}(\boldsymbol{\gamma}, \boldsymbol{\theta})$ for some $\boldsymbol{\gamma} \in [0,1]^T$. Consider $\boldsymbol{\alpha} := \boldsymbol{\alpha}' + \boldsymbol{\alpha}''$ where $\boldsymbol{\alpha}', \boldsymbol{\alpha}'' \in \{0,1\}^T$ such that $\alpha'_t = 1 \implies \alpha''_t = 0$ and $\alpha''_t = 1 \implies \alpha'_t = 0$, and $p_{\mathcal{A}_{\boldsymbol{\gamma}}}(\boldsymbol{\alpha}) > 0$. For all $y \in \mathsf{Y}$, it follows immediately from Proposition D.1 that $p_{\boldsymbol{X}}$-almost everywhere we can use our classifiers to compute the pointwise mutual information quantities*

$$\text{pmi}(y; \boldsymbol{x}_{\boldsymbol{\alpha}'} \mid \boldsymbol{x}_{\boldsymbol{\alpha}''}) := \log p_{Y \mid \boldsymbol{X}_{\boldsymbol{\alpha}}}(y \mid \boldsymbol{x}_{\boldsymbol{\alpha}}) - \log p_{Y \mid \boldsymbol{X}_{\boldsymbol{\alpha}''}}(y \mid \boldsymbol{x}_{\boldsymbol{\alpha}''}) \tag{46}$$

$$= \log \Phi_{\boldsymbol{\alpha}}(y \mid \boldsymbol{x}_{\boldsymbol{\alpha}}; \hat{\boldsymbol{\theta}}) - \log \Phi_{\boldsymbol{\alpha}''}(y \mid \boldsymbol{x}_{\boldsymbol{\alpha}''}; \hat{\boldsymbol{\theta}}). \tag{47}$$

This is useful because it allows us to assess leakage from *single* power traces, as opposed to merely summarizing distributions of power traces. There are scenarios where a power measurement might leak for some traces but not for others. For example, a common countermeasure is to randomly delay leaky instructions or swap their order with another instruction so that they do not occur at a deterministic time relative to the start of encryption. One could use pmi computations to determine the timetep at which the leaky instruction has been run in a single trace.

Since it would be impractical to train $2^T$ deep neural networks independently, we implement the family of classifiers by a single neural net with input dropout and with the dropout mask fed to the neural net as an auxiliary input:

$$\Phi : \mathsf{Y} \times \mathbb{R}^T \times \{0,1\}^T \times \mathbb{R}^P \to [0,1] : (y, \boldsymbol{x}, \boldsymbol{\alpha}, \boldsymbol{\theta}) \mapsto \Phi(y \mid \boldsymbol{x} \odot \boldsymbol{\alpha}, \boldsymbol{\alpha}; \boldsymbol{\theta}) \tag{48}$$

where $\Phi_{\boldsymbol{\alpha}}(y \mid \boldsymbol{x}_{\boldsymbol{\alpha}}; \boldsymbol{\theta}) := \Phi(y \mid \boldsymbol{x} \odot \boldsymbol{\alpha}, \boldsymbol{\alpha}; \boldsymbol{\theta})$. This approach was inspired by Lippe et al. (2022).

### D.3 Re-parametrization into an unconstrained optimization problem

We would like to approximately solve equation 28 using an alternating stochastic gradient descent-style approach, similarly to GANs (Goodfellow et al., 2014). Thus, it is convenient to express it as an unconstrained optimization problem. We first define a new vector $\boldsymbol{\eta} \in \Delta^{T-1}$ where $\Delta^{T-1} := \{(\delta_1, \ldots, \delta_T) \in \mathbb{R}_+^T : \sum_{t=1}^T \delta_t = 1\}$ denotes the $T$-simplex. We then define $\boldsymbol{\gamma}$ to be the vector satisfying the equality

$$c(\gamma_t) = C\eta_t \tag{49}$$

$$\implies \frac{\gamma_t}{1 - \gamma_t} = C\eta_t \tag{50}$$

$$\implies \log \gamma_t - \log(1 - \gamma_t) = \log C + \log \eta_t \tag{51}$$

$$\implies \gamma_t = \text{sigmoid}\left(\log C + \log \eta_t\right). \tag{52}$$

If we define $\boldsymbol{\eta} \coloneqq \mathrm{softmax}(\tilde{\boldsymbol{\eta}})$ for $\tilde{\boldsymbol{\eta}} \in \mathbb{R}^T$, then we can express

$$\gamma_t = \mathrm{sigmoid}\left(\log C + \log \tilde{\eta}_t - \mathrm{logsumexp}(\tilde{\boldsymbol{\eta}})\right), \tag{53}$$

which allows us to map the unconstrained vector $\tilde{\boldsymbol{\eta}}$ to $\boldsymbol{\gamma}$ or $\log \boldsymbol{\gamma}$ using numerically-stable PyTorch operations. Our constrained optimization problem 28 is thus equivalent to the following unconstrained problem:

$$\min_{\tilde{\boldsymbol{\eta}} \in \mathbb{R}^T} \max_{\boldsymbol{\theta} \in \mathbb{R}^P} \quad \mathcal{L}(\tilde{\boldsymbol{\eta}}, \boldsymbol{\theta}) \coloneqq \mathbb{E} \log \Phi\left(Y \mid \boldsymbol{X} \odot \mathcal{A}_{\boldsymbol{\gamma}(\tilde{\boldsymbol{\eta}})}, \mathcal{A}_{\boldsymbol{\gamma}(\tilde{\boldsymbol{\eta}})}; \boldsymbol{\theta}\right). \tag{54}$$

### D.4 Implementation details

It is infeasible to exactly compute the expectation with respect to $\mathcal{A}_{\boldsymbol{\gamma}(\tilde{\boldsymbol{\eta}})}$ because doing so would require summing over $2^T$ terms. As is routine in deep learning contexts, we instead approximate the gradient with Monte Carlo integration. Note that our objective[4] takes the form $\mathcal{L}(\tilde{\boldsymbol{\eta}}) = \mathbb{E} f(\mathcal{A}_{\boldsymbol{\gamma}(\tilde{\boldsymbol{\eta}})})$ where $f(\boldsymbol{\alpha}) \coloneqq \mathbb{E}_{\boldsymbol{X},Y} \log \Phi(Y \mid \boldsymbol{X} \odot \boldsymbol{\alpha}, \boldsymbol{\alpha}; \boldsymbol{\theta})$ and the distribution of $\mathcal{A}_{\boldsymbol{\gamma}(\tilde{\boldsymbol{\eta}})}$ depends on $\tilde{\boldsymbol{\eta}}$.

Unbiased estimators for $\nabla_{\tilde{\boldsymbol{\eta}}} \mathcal{L}(\boldsymbol{\eta})$ of this form are usually based on the REINFORCE estimator (Williams, 1992) with control variates, and tend to have complicated implementations and high variance. We tried using the vanilla REINFORCE estimator with simple control variates as well as the more-sophisticated REBAR estimator (Tucker et al., 2017), and found that the former works poorly, while the latter works well. Subsequent ablation studies revealed that the biased CONCRETE estimator (Maddison et al., 2017) works almost as well as REBAR for our application and is considerably simpler, so in this work we use CONCRETE.

The CONCRETE estimator lets us write $\mathcal{A}_{\boldsymbol{\gamma}(\tilde{\boldsymbol{\eta}})}$ as a deterministic function of $\tilde{\boldsymbol{\eta}}$ and $\tilde{\boldsymbol{\eta}}$-independent noise and thereby use the reparameterization trick to estimate $\nabla_{\tilde{\boldsymbol{\eta}}} \mathcal{L}(\tilde{\boldsymbol{\eta}})$ using standard automatic differentiation tools. For binary random variables such as the elements of $\mathcal{A}_{\boldsymbol{\gamma}(\tilde{\boldsymbol{\eta}})}$, the estimator is built on the observation that

$$x \sim \mathrm{Bernoulli}(p) \equiv x = H(\log p - \log(1-p) + \log u - \log(1-u)) \quad \text{for} \quad u \sim \mathcal{U}(0,1) \tag{55}$$

where $H(x) \coloneqq \begin{cases} 1 & x \geq 0 \\ 0 & x < 0 \end{cases}$ denotes the unit step function. $H$ is not amenable to gradient descent because its derivative is zero almost everywhere, but we can approximate it with the tempered sigmoid function $\mathrm{sigmoid}_\tau(x) \coloneqq \mathrm{sigmoid}(x/\tau)$. The temperature parameter $\tau > 0$ can be tuned to control a bias-variance tradeoff for the estimator: $\lim_{\tau \to 0} \mathrm{sigmoid}_\tau(x) = H(x) \; \forall x \in \mathbb{R} \setminus \{0\}$, but variance increases as $O(1/\tau)$ (Shekhovtsov, 2021). We find that results are reasonable when we simply leave $\tau$ fixed at 1, and we do this throughout the present work. We conjecture that this is because all our performance evaluation metrics consider only the *relative* leakiness estimates produced by our method, and the nature of the bias is not to significantly impact the relative sizes of the elements of $\boldsymbol{\gamma}^*$.

Note that while our original loss function

$$\ell(\tilde{\boldsymbol{\eta}}, \boldsymbol{\theta}, \boldsymbol{x}, y, \boldsymbol{\alpha}) \coloneqq \log \Phi(y \mid \boldsymbol{x} \odot \boldsymbol{\alpha}, \boldsymbol{\alpha}; \boldsymbol{\theta}) \tag{56}$$

is written for a 'hard' occlusion mask $\boldsymbol{\alpha} \in \{0,1\}^T$, the relaxed occlusion masks lie inside the open ball $(0,1)^T$. Thus, we must relax our loss function to accept these inputs as well. While the right-hand side of Eqn. 56 is still a valid expression for $\boldsymbol{\alpha}$ in $(0,1)^T$, its optimal value of this loss with respect to $\boldsymbol{\theta}$ does not vary smoothly with $\boldsymbol{\alpha}$ because rescaling the elements of $\boldsymbol{X}$ by nonzero constants does not change its mutual information with $Y$. Thus, we instead use the stochastic relaxed loss function

$$\ell_{\mathrm{relaxed}}(\tilde{\boldsymbol{\eta}}, \boldsymbol{\theta}, \boldsymbol{x}, y, \boldsymbol{\alpha}) \coloneqq \log \Phi(y \mid \boldsymbol{x} \odot \boldsymbol{\alpha} + \boldsymbol{\varepsilon} \odot (\mathbf{1} - \boldsymbol{\alpha}); \boldsymbol{\theta}) \quad \text{where} \quad \boldsymbol{\varepsilon} \sim \mathcal{N}(0,1)^T. \tag{57}$$

In Alg. 2 we provide pseudocode for a practical implementation of ALL, omitting details such as minibatch use for clarity. Also refer to this link for a minimal self-contained PyTorch (Paszke et al., 2019) implementation.

---

[4]ignoring its $\boldsymbol{\theta}$-dependence because differentiating with respect to $\boldsymbol{\theta}$ is straightforward here

---

**Algorithm 2:** Simplified implementation of our Adversarial Leakage Localization (ALL) algorithm.

---

**Input:** Dataset $\mathsf{D} := \{(\boldsymbol{x}^{(n)}, y^{(n)}) : n = 1, \ldots, N\} \subseteq \mathbb{R}^T \times \mathsf{Y}$, mask-conditional classifier architecture
$\tilde{\Phi}_\cdot : \mathbb{R}^T \times [0,1]^T \to \mathbb{R}^{|\mathsf{Y}|}$, initial classifier weights $\boldsymbol{\theta}_0 \sim \mathbb{R}^P$, initial pre-constraint occlusion logits
$\tilde{\boldsymbol{\eta}}_0 \in \mathbb{R}^T$, occlusion budget $\overline{\gamma} \in (0,1)$

**Output:** Per-timestep 'leakiness' estimate $\boldsymbol{\gamma}^* \in [0,1]^T$

---

**1** **Function** getOcclProbLogits ($\tilde{\boldsymbol{\eta}} \in \mathbb{R}^T$: pre-constraint logits of occlusion probabilities)

**2**     **return** $\tilde{\boldsymbol{\eta}} - \mathrm{logsumexp}(\tilde{\boldsymbol{\eta}}) + \log T + \log \overline{\gamma} - \log(1 - \overline{\gamma})$

**3** **Function** sampleFromCONCRETE ($\tilde{\boldsymbol{\gamma}} \in (0,1)^T$: logits of occlusion probabilities)

**4**     **return** $\mathrm{sigmoid}(\mathrm{logsigmoid}(\tilde{\boldsymbol{\gamma}}) - \mathrm{logsigmoid}(-\tilde{\boldsymbol{\gamma}}) + \log \boldsymbol{u} - \log(\mathbf{1} - \boldsymbol{u}))$, $\boldsymbol{u} \sim \mathcal{U}(0,1)^T$

**5** **Function** getMaskedCrossEntropy ($\boldsymbol{\theta} \in \mathbb{R}^P$: classifier weights, $(\boldsymbol{x}, y) \in \mathbb{R}^T \times \mathsf{Y}$: input and label,
$\boldsymbol{\alpha} \in [0,1]^T$: relaxed input mask)

**6**     **return** $\mathrm{logsoftmax}\,\tilde{\Phi}_{\boldsymbol{\theta}}(y \mid (\mathbf{1} - \boldsymbol{\alpha}) \odot \boldsymbol{x} + \boldsymbol{\alpha} \odot \boldsymbol{\varepsilon}, \mathbf{1} - \boldsymbol{\alpha})$, $\boldsymbol{\varepsilon} \sim \mathcal{N}(0,1)^T$

**7** **for** $t = 0, 1, \ldots$ until convergence **do**

**8**     $(\boldsymbol{x}_t, y_t) \leftarrow$ sampleDatapoint($\mathsf{D}$)

**9**     $\tilde{\boldsymbol{\gamma}}_t \leftarrow$ getOcclProbLogits($\tilde{\boldsymbol{\eta}}_t$)

**10**     $\boldsymbol{\alpha}_t \leftarrow$ sampleFromCONCRETE($\tilde{\boldsymbol{\gamma}}_t$)

**11**     $\ell_t \leftarrow$ getMaskedCrossEntropy($\boldsymbol{\theta}_t, (\boldsymbol{x}_t, y_t), \boldsymbol{\alpha}_t$)

**12**     $\boldsymbol{g}_t^\theta \leftarrow \nabla_{\boldsymbol{\theta}} \ell_t$, $\boldsymbol{g}_t^{\tilde{\eta}} \leftarrow -\nabla_{\tilde{\boldsymbol{\eta}}} \ell_t$, $\boldsymbol{\theta}_{t+1} \leftarrow$ OptStep($\boldsymbol{\theta}_t, \boldsymbol{g}_t^\theta$), $\tilde{\boldsymbol{\eta}}_{t+1} \leftarrow$ OptStep($\tilde{\boldsymbol{\eta}}_t, \boldsymbol{g}_t^{\tilde{\eta}}$)

**13** **return** $\mathrm{sigmoid}\,$getOcclProbLogits($\tilde{\boldsymbol{\gamma}}_{t+1}$)

---

# E Extended experimental details and results

## E.1 Toy setting where our method succeeds and prior work fails

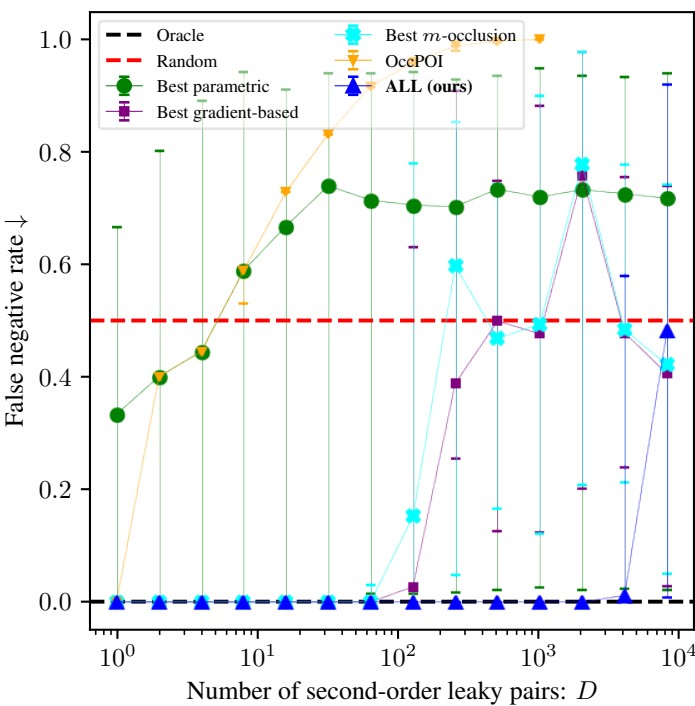

Figure 9: A toy setting where ALL (ours) significantly outperforms baselines. We sample 1 non-leaky feature and $D$ second-order leaky pairs, then plot the false negative rate, defined as the proportion of points incorrectly assigned leakiness less than or equal to that of the non-leaky point, as we increase $D$. ALL (ours) succeeds for $D$ up to 64× higher the best prior deep learning-based approach, and the first-order parametric methods completely fail in this setting. Dots denote median and error bars denote min–max over 5 random seeds.

As previously discussed, first-order parametric statistics-based methods are insensitive to associations of order 2 or higher. Prior deep learning-based leakage localization algorithms tend to exploit few of the available $2^{T-1}$ associations between $X_t$ and $Y$ given subsets of $\{X_1, \ldots, X_T\} \setminus \{X_t\}$, with many of them using only the maximal conditioning set $\{X_1, \ldots, X_T\} \setminus \{X_t\}$ itself. This creates issues when there is a large number of leaky measurements and the individual contribution of each is 'drowned out' in the sense that $\mathbb{I}[Y; X_t \mid \{X_1, \ldots, X_T\} \setminus \{X_t\}]$ vanishes. Here we construct a simple setting where both of these issues are present, and demonstrate that ALL succeeds whereas the prior approaches face issues.

We generate a sequence of binary-label $2D + 1$-feature classification datasets consisting of ordered pairs $(\boldsymbol{X}, Y)$ sampled independently as follows:

$$Y \sim \mathcal{U}\{0, 1\}, \ R \sim \mathcal{U}\{0, 1\}, \ M_i \sim \mathcal{U}\{0, 1\} : i = 1, \ldots, D, \tag{58}$$

$$X_R \sim \mathcal{N}(R, 1), \ X_{M_i} \sim \mathcal{N}(M_i, 1), \ X_{Y \oplus M_i} \sim \mathcal{N}(Y \oplus M_i, 1) : i = 1, \ldots, D. \tag{59}$$

Here we denote by $\oplus$ the exclusive-or operation and $\boldsymbol{X} \equiv (X_R, X_{M_1}, X_{Y \oplus M_1}, \ldots X_{M_D}, X_{Y \oplus M_D})$. Intuitively, we can view $X_R$ as a noisy observation of $R$ and each $X_{M_i}, X_{Y \oplus M_i}$ as noisy observations of $M_i, Y \oplus M_i$, respectively. Here the variable $Y$ is analogous to a targeted sensitive variable, and the values $(M_i, Y \oplus M_i)$ are analogous to the pairs of second-order leaky variables which arise in Boolean masked implementations such as Benadjila et al. (2020).

Clearly $R$, and thus $X_R$, tells us nothing about $Y$. Additionally, the values $M_i$ *in isolation* tell us nothing about $Y$. Similarly, the values $Y \oplus M_i$ in isolation tell us nothing about $Y$ because

$$\mathbb{P}(Y = 0 \mid Y \oplus M_i = 0) = \mathbb{P}(M_i = 0) = \tfrac{1}{2} = \mathbb{P}(M_i = 1) = \mathbb{P}(Y = 1 \mid Y \oplus M_i = 0) \tag{60}$$

and similarly

$$\mathbb{P}(Y = 0 \mid Y \oplus M_i = 1) = \mathbb{P}(M_i = 1) = \tfrac{1}{2} = \mathbb{P}(M_i = 0) = \mathbb{P}(Y = 1 \mid Y \oplus M_i = 1). \tag{61}$$

Despite this, given the *pair* of values $\{M_i, Y \oplus M_i\}$ we can recover $Y$ via the identity

$$M_i \oplus M_i = 0 \implies (Y \oplus M_i) \oplus M_i = Y. \tag{62}$$

Thus, we would like a leakage localization algorithm to indicate that $X_R$ is non-leaky and the values $X_{M_i}$, $X_{Y \oplus M_i}$ are leaky.

All experiments use the hyperparameters shown in Table 5.

Table 5: Hyperparameters used for our toy Gaussian dataset experiments. We use the default PyTorch settings except where specified. $^\dagger$These hyperparameters apply only to ALL. Other hyperparameters are used both for ALL and for our baseline methods.

| Hyperparameter | Value |
|---|---|
| Classifier architecture | ReLU MLP with $1 \times 500$-neuron hidden layer |
| Classifier optimizer | AdamW |
| Classifier learning rate | $10^{-4}$ |
| Classifier weight decay | 0.01 |
| Dataset size | 10k |
| Training steps | 5k |
| Minibatch size | 800 |
| Noise distribution learning rate$^\dagger$ | $10^{-3}$ |
| Budget $\overline{\gamma}^\dagger$ | $1 - 2^{-0.1 \cdot D - 1}$ |

Results can be seen in Fig. 9. We measure the performance of methods by the percent of measurements in $\{X_{M_i} : i = 1, \ldots, D\} \cup \{X_{Y \oplus M_i} : i = 1, \ldots, D\}$ assigned a leakiness greater than or equal to that of $X_R$. For clarity we report for each $D$ the best result out of SNR, SOSD and CPA as 'best parametric', out of GradVis, Saliency, Input $*$ Grad, and LRP as 'best gradient-based', and the best result out of $\{2m + 1 : m \in [0 .. 24], m < 2D + 1\}$-Occlusion as 'best $m$-Occlusion'. Due to their high cost we use OccPOI only for $D$ up to 256, and we do not use $2^{\text{nd}}$-order $m$-Occlusion. Note that in subsequent experiments $2^{\text{nd}}$-order $m$-Occlusion does not significantly outperform first-order occlusion, and OccPOI performs poorly due to identifying only a small number of leaky measurements rather than assigning a leakiness to every measurement.

## E.2 Simulated AES datasets where we have ground truth knowledge about leakage

Here we present experiments done on synthetic AES power traces. These are a useful complement to the experiments on real datasets because 1) here we have ground truth knowledge about which timesteps are leaking, which we can use to validate our model's output, 2) we can generate infinitely-large datasets to eliminate dataset size as a confounding variable in results, and 3) we can observe the change in our technique's behavior as we individually vary particular dataset properties such as low-pass filtering strength and leaky instruction count.

---

**Algorithm 3:** Pseudocode for our synthetic data generation procedure, based on the Hamming weight leakage model of Mangard et al. (2007). For clarity we omit the random delay and shuffling procedures, but these are straightforward and may be found in our code.

---

**Input:**    Dataset size $N \in \mathbb{Z}_{++}$,

        Timesteps per power trace $T \in \mathbb{Z}_{++}$,

        Bit count $n_{\text{bits}} \in \mathbb{Z}_{++}$,

        Operation count $n_{\text{ops}} \in \mathbb{Z}_{++}$,

        Data-dependent noise variance $\sigma_{\text{data}}^2 \in \mathbb{R}_+$,

        Operation-dependent noise variance $\sigma_{\text{op}}^2 \in \mathbb{R}_+$,

        Residual noise variance $\sigma_{\text{resid}}^2 \in \mathbb{R}_+$,

        Low-pass filtering strength $\beta \in [0, 1)$,

        Leaking timestep count $n_{\text{lkg}} \in \mathbb{Z}_+$

**Output:** Synthetic dataset $\mathsf{D} \subseteq \mathbb{R}^T \times [0 \mathbin{..} 2^{n_{\text{bits}}} - 1]$

---

1   $\{k^{(n)} : n \in [1 \mathbin{..} N]\} \sim \mathcal{U}\left(\{0,1\}^{n_{\text{bits}}}\right)^N$      `// AES keys`

2   $\{w^{(n)} : n \in [1 \mathbin{..} N]\} \sim \mathcal{U}\left(\{0,1\}^{n_{\text{bits}}}\right)^N$      `// plaintexts`

3   $\{o_t : t \in [1 \mathbin{..} T]\} \sim \mathcal{U}\left([1 \mathbin{..} n_{\text{ops}}]\right)^N$      `// operations`

4   $\{\tilde{x}_{\text{op}, i} : i \in [1 \mathbin{..} n_{\text{ops}}]\} \sim \mathcal{N}(0, \sigma_{\text{op}}^2)^{n_{\text{ops}}}$      `// operation-dependent power consumption`

5   $\mathsf{T}_{\text{lkg}} \sim \mathcal{U}\left(\begin{bmatrix} [1 \mathbin{..} T] \\ n_{\text{lkg}} \end{bmatrix}\right)$      `// timesteps of leaky instructions`

6   **for** $n \in [1 \mathbin{..} N]$ **do**

7      $y^{(n)} \leftarrow \text{Sbox}(k^{(n)} \oplus w^{(n)})$      `// targeted variable:  first SubBytes output`

8      $\boldsymbol{x}_{\text{resid}}^{(n)} \sim \mathcal{N}(0, \sigma_{\text{resid}}^2)^T$      `// residual power consumption`

9      **for** $t \in \mathsf{T}_{\text{lkg}}$ **do**

10         $d_t^{(n)} \leftarrow y^{(n)}$      `// leaky timesteps:  targeted variable is the data`

11      **for** $t \in [1 \mathbin{..} T] \setminus \mathsf{T}_{\text{lkg}}$ **do**

12         $d_t^{(n)} \sim \mathcal{U}\left(\{0,1\}^{n_{\text{bits}}}\right)$      `// rest of data treated as random`

13      **for** $t \in [1 \mathbin{..} T]$ **do**

14         $x_{\text{data}, t}^{(n)} \leftarrow \sigma_{\text{data}} \left(4 - \text{HW}(d_t^{(n)})\right) / \sqrt{2}$      `// data-dependent power consumption`

15      $\boldsymbol{x}^{(n)} \leftarrow \boldsymbol{x}_{\text{data}}^{(n)} + \boldsymbol{x}_{\text{op}} + \boldsymbol{x}_{\text{resid}}^{(n)}$      `// total power consumption`

16      **for** $t \in [2 \mathbin{..} T]$ **do**

17         $x_t^{(n)} \leftarrow \beta x_{t-1}^{(n)} + (1 - \beta) x_t^{(n)}$      `// Discrete low-pass filtering of` $\boldsymbol{x}^{(n)}$`.`

            `// In practice we prepend 'burn-in' timesteps to allow transient effects to`
               `decay.`

18   **return** $\left\{(\boldsymbol{x}^{(n)}, y^{(n)}) : n \in [1 \mathbin{..} N]\right\}$

---

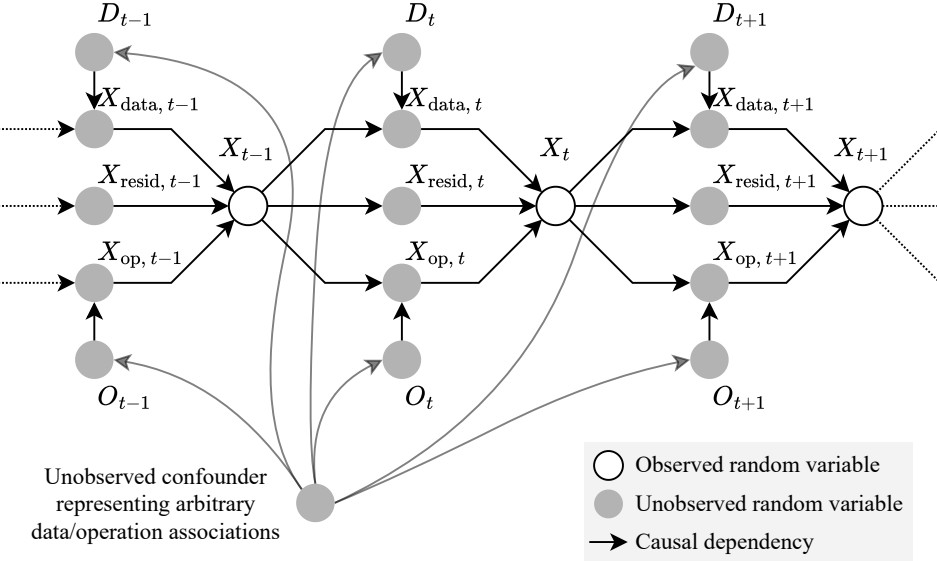

Figure 10: Causal diagram representing the assumed data-generating process for our synthetic AES datasets. We assume this is an I-map for the data-generating process – i.e. we assume independence conditions present in the diagram hold for the data-generating process, but the data-generating process might have additional independence conditions not reflected here. This is an effort to make more precise the Hamming weight leakage model of (Mangard et al., 2007, ch. 4). We represent the data of the AES algorithm by the random variables $D_t$, its operations $O_t$, and its power consumption $X_t$, which is further broken down into data-dependent power consumption $X_{\text{data},t}$, operation-dependent power consumption $X_{\text{op},t}$, and 'residual' power consumption $X_{\text{resid},t}$ (e.g. due to random noise, other processes running in parallel, etc.). We assume that only the composite power consumption $X_t$ is observed.

### E.2.1 Data generation procedure

We base our synthetic data generation procedure on the Hamming weight leakage model of (Mangard et al., 2007, ch. 4), which we will subsequently describe.[5] As above, let us represent our power/EM traces as a random vector $\boldsymbol{X} := (X_t : t = 1, \ldots, T)$ with range $\mathbb{R}^T$. We represent the cryptographic algorithm as sequences of data $\boldsymbol{D} := (D_t : t = 1, \ldots, T)$ and operations $\boldsymbol{O} := (O_t : t = 1, \ldots, T)$, where each $D_t$ has range $\{0,1\}^{n_{\text{bits}}}$ and each $O_t$ has range $[1 .. n_{\text{ops}}]$ for some $n_{\text{bits}}, n_{\text{ops}} \in \mathbb{Z}_{++}$. For each $t \in [1 .. T]$, we can decompose

$$X_t = X_{\text{data}, t} + X_{\text{op}, t} + X_{\text{resid}, t} \tag{63}$$

with dependency structure illustrated in the causal diagram of Fig. 10. Here we have represented by $X_{\text{data}, t}$ the data-dependent component of power consumption, $X_{\text{op}, t}$ the operation-dependent component of power consumption, and $X_{\text{resid}, t}$ the 'residual' power consumption due to random noise, other processes running simultaneously on the same hardware, etc.

Mangard et al. (2007) experimentally characterized the power consumption of a cryptographic device and found that it is reasonable to approximate $X_{\text{data}, t}$ as Gaussian noise with $D_t$-dependent mean, $X_{\text{op}, t}$ as Gaussian noise with $O_t$-dependent mean, and $X_{\text{resid}, t}$ as Gaussian noise with constant mean (which we will assume to be zero because it contains no information and is thus irrelevant). For their device, they found that the mean of $X_{\text{data}, t}$ was roughly proportional to $n_{\text{bits}} - \text{HW}(D_t)$ where $\text{HW} : \{0,1\}^{n_{\text{bits}}} \to [0 .. n_{\text{bits}}] : \boldsymbol{x} \mapsto \sum_{k=1}^{n_{\text{bits}}} x_k$. We adopt these approximations for our synthetic dataset experiments. See Alg. 3 for pseudocode giving a simplified version of our data generation procedure, and refer to our code for full details.

We simulate several factors of variation which can reasonably be expected to occur in realistic settings. We apply discrete-time low-pass filtering to the power traces, to simulate the low-pass filtering which occurs due to measurement apparatus as well as the fact that power consumption does not change instantaneously in real circuits. We allow for the presence of multiple leaky instructions. Additionally, we simulate random delays to the leaky instruction which might result from countermeasures such as random no-op insertion (Coron & Kizhvatov, 2009), and random 'shuffling' where the leaky instruction is randomly placed at one of several points in time, as done in Masure & Strullu (2023).

We emphasize that these approximations are specific to the device studied by Mangard et al. (2007) and may hold to a limited extent or not at all for other devices. For example, the Hamming weight dependence of power consumption stems from the fact that their device 'pre-charges' all its data bus lines to 1, then drains the charge from the lines which should represent 0, thereby consuming power proportional to the number of lines which represent 0. Many devices operate differently, and cryptographic hardware is often designed with the explicit goal of obfuscating this data/power consumption dependence. Thus, while we expect all real cryptographic devices to have some exploitable dependence between data and power consumption given sufficient quality and quantity of data, in many cases the nature of the dependence will likely elude a simple characterization such as this.

### E.2.2 Experimental details

We run experiments on many variations of this dataset and verify that ALL produces outputs which align with our expectations. For all these experiments, our classifier $\Phi_{\boldsymbol{\theta}}$ is a 3-layer multilayer perceptron with hidden dimension 500, ReLU activations, an input dropout rate of 0.1, hidden dropout rate of 0.2, and pre-logits dropout rate of 0.3. We generate data continuously so that our dataset size is effectively infinite. Additional hyperparameters are listed in table 6, and default settings corresponding to Alg. 3 are listed in table 7, which we use except where otherwise stated.

### E.2.3 Results

We vary 4 parameters of the data generation process of Alg. 3 and observe its effect on the output of ALL. We find that ALL consistently produces results consistent with our expectations given the timestep(s) of

---

[5]For clarity we alter the notation and explicitly define a causal structure for the data-generating process. The decomposition of power consumption in Eqn. 63 and the definition of $X_{\text{data}, t}$ in terms of the Hamming weight of the data are based on Mangard et al. (2007), but the additional details are our own.

Table 6: List of hyperparameters used for experiments on synthetic AES datasets. We use the default PyTorch settings unless otherwise stated. Note that in general for `ALL` classfiers we disable input dropout, but for these experiments the dropout rate was set to 0.1 due to an oversight.

| Hyperparameter | Value |
|---|---|
| Classifier architecture | ReLU MLP with $3 \times 500$-neuron hidden layers |
| Input dropout | 0.1 |
| Hidden dropout | 0.2 |
| Output dropout | 0.3 |
| Optimizer for both $\boldsymbol{\theta}$ and $\tilde{\boldsymbol{\eta}}$ | `torch.optim.AdamW` |
| Weight decay for $\boldsymbol{\theta}$ | 0.01 for weights, 0 for biases |
| Weight decay for $\tilde{\boldsymbol{\eta}}$ | 0 |
| Training steps | $10^4$ |
| Minibatch size | $10^3$ |
| Noise budget $\overline{\gamma}$ | 0.5 |
| Weight initializer | `torch.nn.init.xavier_uniform_` |

Table 7: Default synthetic AES dataset configuration, corresponding to the inputs of Alg. 3. Subsequent experiments will use these settings unless otherwise stated.

| Setting | Value |
|---|---|
| Dataset size $N$ | $\infty$ |
| Timesteps per power trace $T$ | 101 |
| Bit count $n_{\text{bits}}$ | 8 |
| Operation count $n_{\text{op}}$ | 32 |
| Data-dependent noise variance $\sigma^2_{\text{data}}$ | 1.0 |
| Operation-dependent noise variance $\sigma^2_{\text{op}}$ | 1.0 |
| Residual noise variance $\sigma^2_{\text{resid}}$ | 1.0 |
| Number of leaky instructions $n_{\text{lkg}}$ | 1 |
| Low-pass filtering strength $\beta$ | 0.5 |
| Maximum random delay size | 0 |
| Possible leaky timestep location count (i.e. shuffling) | 1 |

the leaky instruction(s), and that the variance of its output is quite low in this context where we have an infinitely-large dataset and a long training duration. Here we describe the parameters being swept and justify the output of `ALL` given this.

**Low pass filtering strength** $\beta$   Recall that we are discrete low-pass filtering traces via the recursive function $x_t^{\text{lpf}} \coloneqq (1 - \beta)x_t + \beta x_{t-1}$. See the first row of Fig. 11, where from left to right $\beta$ takes on the values 0, 0.5, 0.75, 0.875, 0.9375, 0.96875, 0.984375, 0.9921875. Note that the peak estimated leakiness always corresponds to the ground-truth leaky instruction timestep. As we increase $\beta$ we see that measurements to the left and right are assigned high leakiness as well, which makes sense for the following reason:

Let us denote by $t^*$ the timestep at which the leaky instruction was executed. We then have $X_{t^*} = c_1 \operatorname{HW}(Y) + c_2 U_{t^*} + c_3 X_{t^*-1}$ where $U_{t^*}$ denotes a 'residual' random variable independent of $Y$, and $c_1$, $c_2$ are appropriate constants. Note that $X_{t^*+1} = \beta X_{t^*} + (1 - \beta)U_{t^*+1} = \beta c_1 \operatorname{HW}(Y) + \beta c_2 U_{t^*} + (1 - \beta)U_{t^*+1}$, so $X_{t^*+1}$ also leaks $Y$. Recursively it is clear that the same can be said for $X_{t^*+2}, X_{t^*+3}, \ldots$. Less-intuitively, $X_{t^*-1}$ also leaks $Y$. This is because although $X_{t^*-1}$ is *marginally* independent of $Y$, $X_{t^*} - c_3 X_{t^*-1}$ has a higher correlation with $\operatorname{HW}(Y)$ than $X_{t^*}$ does – i.e. $X_{t^*-1}$ is dependent on the $Y$-independent noise of $X_{t^*}$, and can be used to reduce the noise. Recursively, since $X_{t^*-2}, X_{t^*-3}, \ldots$ are correlated with $X_{t^*-1}$, they also leak $Y$ by the same mechanism.

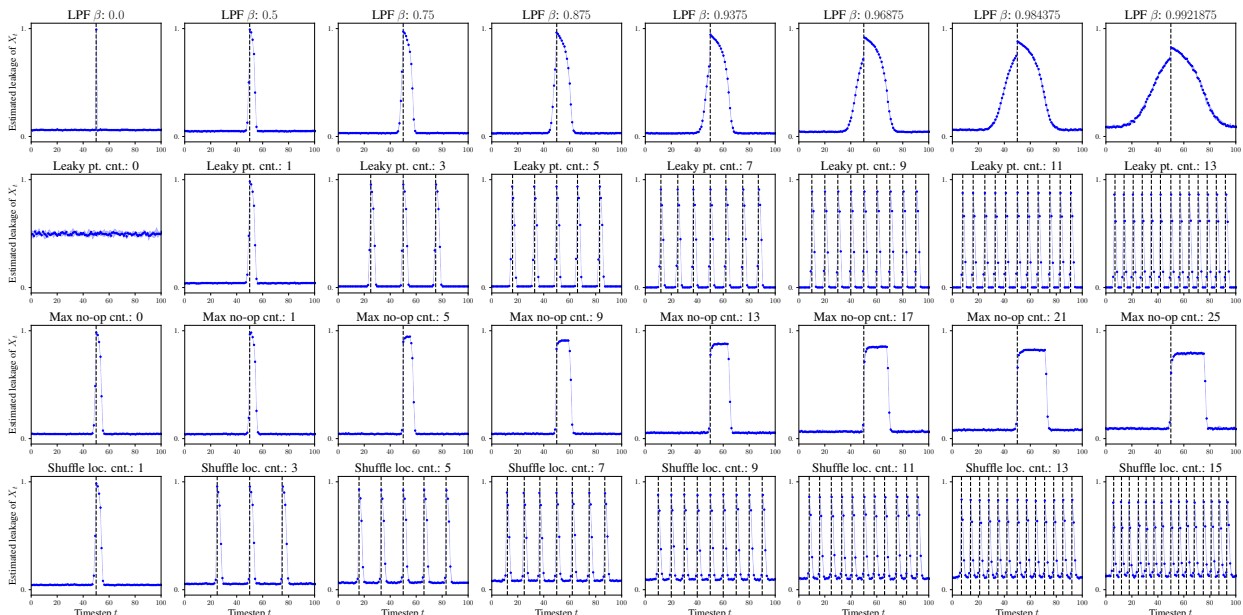

Figure 11: Results of applying ALL (ours) to synthetic AES datasets with varying parameters. The estimated leakage by ALL is denoted by blue dots and the ground truth timestep of the leaking instruction is denoted by the vertical black dotted lines. Blue dots denote mean and shading denotes median over 5 random seeds. Note that the ALL output is consistent with the ground truth leaky instruction timestep in all cases, and the variance between runs is quite low in the infinite data regime. **(first row)** Increasing low pass filtering strength $\beta$ from left to right. **(second row)** Increasing number of leaky instructions from left to right. **(third row)** Increasing random delay insertion from left to right. **(fourth row)** Increasing number of possible shuffling locations for the leaky point from left to right.

**Leaky point count** Here we sweep the number of leaky instructions. See the second row of Fig. 11 where from left to right the leaky point count $n_{\text{lkg}}$ takes on the values 0, 1, 3, 5, 7, 9, 11, 13. As expected, all leaky instruction timesteps correspond to a peak of ALL-estimated leakiness.

**Random delay size** Here we insert random delays – i.e. instead of always occurring at time $t$, the leaky instruction occurs at $t + u$ where $u \sim \mathcal{U}\{0, \ldots, d_{\text{max}}\}$. See the third row of Fig. 11 where from left to right $d_{\text{max}}$ takes on values 0, 1, 5, 9, 13, 17, 21, 25. We see that the estimated leakiness becomes 'spread out' over the interval $[t \mathinner{..} t + d_{\text{max}}]$.

**Shuffle location count** Here we randomly 'shuffle' the leaky instruction – i.e. instead of always occurring at time $t$, the leaky instruction occurs at $t' \sim \mathcal{U}\{t_1, \ldots, t_{n_{\text{shuff}}}\}$. See the fourth row of Fig. 7 where from left to right $n_{\text{shuff}}$ takes on values 1, 3, 5, 7, 9, 11, 13, 15. We see that the leakiness becomes 'spread out' over the set of timesteps at which the leaky instruction might occur.

### E.3 Experiments on real power and EM radiation leakage datasets

Here we run experiments on a variety of publicly-available side-channel attack datasets, where we attempt to localize leakage of their canonical target variable.

#### E.3.1 Datasets

We compare ALL with prior work on the 6 datasets described in Table 8, which consist of traces of power and EM radiation measurements and associated cryptographic variables recorded from real implementations. Note that we evaluate on AES, ECC and RSA implementations implemented on several MCUs and an FPGA

Table 8: A list of the datasets used in our paper, with a summary of their salient attributes. Note that our experiments cover a variety of settings: AES, RSA and ECC implementations on both microcontrollers (MCUs) and a field-programmable gate array (FPGA), both power and EM radiation traces, and various types of countermeasures. For all datasets we localize leakage of the canonical target variable. We denote by subscripts the targeted byte of the variable. We denote by $k_n$, $w_n$, $m_n$, $k_n^*$, $c_n$ the $n$-th byte (counting from 0) of the AES key, plaintext, mask, last round key, and ciphertext, respectively. †As described below, we deviate from the canonical profiling/attack split.

| **Dataset** | ASCADv1 (fixed key) | ASCADv1 (variable key) | DPAv4 (Zaid version) |
|---|---|---|---|
| **Citation** | Benadjila et al. (2020) | Benadjila et al. (2020) | Bhasin et al. (2014); Zaid et al. (2020) |
| **Link** | (here) | (here) | (here) |
| **Algorithm** | AES-128 | AES-128 | AES-128 |
| **Hardware** | ATMega8515 (MCU) | ATMega8515 (MCU) | ATMega163 (MCU) |
| **Emission measured** | Power | Power | Power |
| **Countermeasures** | Boolean masking | Boolean masking | Rotating Sbox mask (known) |
| **Targeted variable** | $\text{Sbox}(k_3 \oplus w_3)$ | $\text{Sbox}(k_3 \oplus w_3)$ | $\text{Sbox}(k_0 \oplus w_0)$ |
| **Dataset size** (profile/attack) | 50k/10k | 200k/100k | 3k/500† |
| **Feature count** $T$ | 0.7k | 1.4k | 4k |

| **Dataset** | AES-HD | One Trace is All it Takes (OTiAiT) | One Truth Prevails (OTP) (1024-bit) |
|---|---|---|---|
| **Citation** | Bhasin et al. (2020) | Weissbart et al. (2019) | Saito et al. (2022) |
| **Link** | (here) | (here) | (here) |
| **Algorithm** | AES-128 | EdDSA w/ Curve2559 | 1024-bit RSA-CRT |
| **Hardware** | XiLinx Virtex-5 (FPGA) | STM32F4 (MCU) | STM32F4 (MCU) |
| **Emission measured** | EM radiation | Power | EM radiation |
| **Countermeasures** | None | None | Dummy load |
| **Targeted variable** | $\text{Sbox}^{-1}(k_{11}^* \oplus c_{11}) \oplus c_7$ | Ephemeral key nibble | Dummy load? |
| **Dataset size** (profile/attack) | 50k/25k | 5.12k/1.28k | 100k/98.304k† |
| **Feature count** $T$ | 1.25k | 1k | 1k |

with various target variables. ALL as well as most of our baseline algorithms are in principle agnostic to most of these details, requiring only a supervised learning-style dataset with power traces and the associated value of the targeted variable as labels. Through these experiments we demonstrate that this is true in practice across a diverse array of settings.

Note that the ASCADv1 datasets have primarily second-order leakage due to their Boolean masking countermeasure, whereas the other 4 datasets have primarily first-order leakage. Our comparisons include both deep learning methods as well as simple first-order parametric methods which are widely used due to their low cost and interpretability. We find that the latter are competitive or superior to the deep learning methods on the first-order datasets but perform significantly worse on the second-order datasets due to failing to exploit second-order leakage. Our experiments do not compellingly show that deep learning methods improve on these simpler methods for first-order datasets, but we nonetheless include them as additional points of comparison between the deep learning methods and to show that ALL works in a variety of settings.

In general we use the canonical target variable and profiling/attack dataset split (note that in the context of profiling side-channel analysis the training dataset is called the profiling dataset, and the test dataset is called the attack dataset). We deviate from the canonical dataset configuration in the following cases:

- We find that the canonical attack dataset of DPAv4 is too small to compute useful oracle leakiness assessments. We thus concatenate the canonical 4.5k-trace profiling dataset and 0.5k-trace attack dataset into a single 5k-length dataset, and use the first 3k traces for profiling and the last 2k to compute the oracle assessments. Since some of our experiments require metadata which is only available for the attack dataset, for everything other than oracle assessment computation we use the canonical attack dataset and leave the remaining 1.5k traces unused.

- We use the version of DPAv4 which was preprocessed and distributed by Zaid et al. (2020) here rather than the original version. This version has shortened traces which have been cropped around the leaky instruction, and has the rotating Sbox mask effectively 'disabled' by providing the masked SubBytes variable as the target.

- The One Truth Prevails (OTP) dataset consists of approximately 64M traces and has a high label imbalance. To save computational resources we extract a 100k-trace randomly-selected balanced subset, which we find is more than sufficient for strong supervised classification and leakage localization performance. See our code for details.

### E.3.2 Implementation details for the leakage localization algorithms

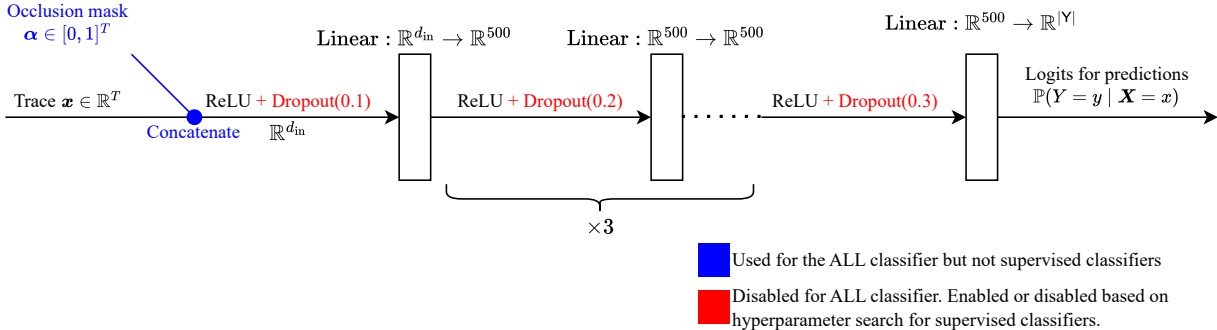

Figure 12: Diagram of the multilayer perceptron architecture used for classifiers in the deep learning methods, based on the architecture proposed in Wang et al. (2017) for time-series classification.

For ALL and all considered baseline methods the classifier uses the simple ReLU + Dropout MLP architecture of Wang et al. (2017) shown in Fig. 12, which has 3 500-neuron hidden layers, input dropout rate of 0.1, hidden dropout rate of 0.2, and output dropout rate of 0.3. For the deep learning baselines we enable or disable the input, hidden and output dropout based on our hyperparameter search outcome, and for ALL we leave it disabled. The classifier for ALL takes the occlusion mask as an auxiliary input, which we implement by concatenating it with the masked trace.

We also explored convolutional architectures, but preliminary experiments indicated that these achieved weaker classification and leakage localization performance across the board, as well as training more-slowly than the MLP architecture due to a higher layer count. We suspect that the inductive biases of convolutional layers are not useful for the datasets we consider. As a sanity check for this design choice, we run the deep learning baseline methods using both our MLP architecture and a handful of open-weight classifiers which were released with Wouters et al. (2020) on the datasets for which they are available.

All deep learning methods are implemented in PyTorch (Paszke et al., 2019). Most non-deep methods are implemented with Numpy (Harris et al., 2020), with a handful of compute-intensive methods implemented with Numba (Lam et al., 2015). We use Scipy (Virtanen et al., 2020) implementations of statistical methods where available.

We use the AdamW optimizer (Loshchilov & Hutter, 2018; Kingma & Welling, 2014) with the default PyTorch settings $\beta_1 = 0.9$, $\beta_2 = 0.999$, $\lambda = 0.01$, $\epsilon = 10^{-8}$ and the learning rate chosen through hyperparameter search. Weight decay is applied only to the weights of the linear layers, not to the biases. We use a minibatch size of 256. Weights are initialized with the uniform Glorot initialization (Glorot & Bengio, 2010) `torch.nn.init.xavier_uniform_` (default for Keras) rather than the default PyTorch initialization, and we find that this is a critical detail – many supervised classifier runs on the ASCADv1 datasets completely fail to generalize beyond the training dataset when the default PyTorch weight initialization is used. We randomly set aside 20% of the profiling datasets for validation and use the remaining 80% for training. We standardize the traces as $x \mapsto \frac{x-\mu}{\max(\sigma, 10^{-6})}$ where $\mu$ and $\sigma$ denote the elementwise sample mean and standard deviation computed using the profiling dataset.

In general we measure the performance of supervised classifiers with their *mean rank* rather than their accuracy, as accuracy tends to be low and too-coarse in the context of side-channel analysis. Given a label $y \in \mathsf{Y}$ and predicted label distribution $\hat{p}_Y \in \Delta^{|\mathsf{Y}|-1}$ (e.g. the softmaxed output of a classifier neural net), we define the rank as the number of possible labels assigned at least as much probability mass as the true label,

i.e.
$$\text{Rank}(\hat{p}_Y; y) \coloneqq |\{y' \in \mathsf{Y} : \hat{p}_Y(y') \geq \hat{p}_Y(y)\}|. \tag{64}$$

This metric has range $[1 \mathbin{..} |\mathsf{Y}|]$, with lower being better.

**Implementation of the baseline methods**  We use Captum (Kokhlikyan et al., 2020) implementations of the Saliency, Input $*$ Grad, LRP, and $m$-Occlusion methods. We implement SNR, SOSD, CPA, GradVis, and $2^{\text{nd}}$-order $m$-Occlusion ourselves, and we implement a PyTorch version of OccPOI based on the Keras implementation released by the authors here.

Note the following choices we have made in implementing and evaluating OccPOI (Yap et al., 2025):

- OccPOI differs from the other methods we consider in that rather than assigning a leakiness value to every measurement, it aims to identify a non-unique subset of measurements which are sufficient for a classifier to attain some specified performance level when all other measurements are occluded. Since our evaluation metrics require a leakiness value for every measurement, we assign a leakiness of 0 to measurements not identified by OccPOI.

- Their method uses the attack dataset for probing the classifier's sensitivity to input features, which introduces data contamination in the context of our performance metrics. We cannot easily use the validation dataset instead because the attack procedure requires having many traces corresponding to a fixed AES key, which is generally only available for the attack dataset. We find that ALL significantly outperforms OccPOI despite this contamination, so as the aim of our work is to demonstrate the efficacy of ALL, we allow OccPOI to use the attack dataset.

- Since OTiAiT and OTP do not have attack datasets which facilitate this kind of multi-trace prediction, for these datasets we simply use the mean rank of the classifier on the attack dataset as our performance metric.

- Yap et al. (2025) proposes an extension of OccPOI which ranks the leakiness of the points it has identified using a 1-occlusion-like strategy. We use this extension in our implementation.

- Yap et al. (2025) proposes an extension of OccPOI where they apply it repeatedly on the residual measurements not selected during the last iteration. We do not use this extension because it is very computationally expensive, requiring $O(T)$ applications of OccPOI which each require $\Omega(T)$ *non-parallelizable* passes through the attack dataset. In preliminary experiments this extension performed better than basic OccPOI, but still far below the performance of the other considered methods despite requiring orders of magnitude more wall clock time.

- Similarly to Yap et al. (2025), to save compute we use only a subset of the attack datasets for classifier evaluation. We use 1.8k traces for ASCADv1-fixed, 2.8k for ASCADv1-variable, 140 for DPAv4, 25k for AES-HD, 100 for OTiAiT, and 100 for OTP. These are approximately 10–100$\times$ the necessary number of traces to successfully attack the AES methods.

- Yap et al. (2025) define their performance threshold to be that the classifier correctly predicts the AES key after accumulating all traces in the attack dataset, and we use the same threshold for the ASCADv1 datasets, AES-HD, and DPAv4. For the non-AES dataset OTiAiT and OTP, our threshold is that the mean rank of the classifier rises by 0.1 relative to its mean rank when no measurements are occluded.

As we will show, OccPOI attains significantly lower performance than other methods according to our performance metrics, due to not assigning a leakiness to most measurements. This reflects that the aim of Yap et al. (2025) is somewhat different from the present work: Yap et al. (2025) heavily emphasized the usefulness of OccPOI as a feature selection tool with leakage localization being an auxiliary goal, whereas our work is concerned solely with leakage localization. We consider OccPOI as a baseline because it is similar in spirit to our work, but these results demonstrate that it is ill-suited to the task considered in our paper.

We make the following choices in implementing $m$-Occlusion and $2^{\text{nd}}$-order $m$-Occlusion (Schamberger et al., 2023):

- Schamberger et al. (2023) explore different ways to occlude measurements and conclude that it works well to replace measurements by their average value over the profiling dataset, whereas other heuristics such as replacing them with 0 or with Gaussian noise works poorly. We thus replace measurements by their mean. Since we are element-wise standardizing the traces, this is the same as replacing them by 0.

- Schamberger et al. (2023) propose $2^\text{nd}$-order $m$-Occlusion as a means of estimating the leakiness of *pairs* of windows, which is useful for discerning whether a measurement has first-order leakage or is part of a second-order leaking pair of measurements. They do not propose a means of using it to estimate the leakiness of individual measurements. We choose to define the leakiness of $X_t$ as the *average* leakiness of the pairs $\{\{X_t, X_{t'}\} : t' \in [1 .. T]\}$.

- Schamberger et al. (2023) uses a large stride for $2^\text{nd}$-order $m$-Occlusion, and thus get leakiness values for windows of measurements rather than single measurements. We set the stride to 1 for consistency with our other baselines.

- These methods introduce the occlusion window size $m$ as a new hyperparameter which must be tuned. For $m$-Occlusion we tune $m$ by testing successive odd-numbered window sizes starting from 1 until oracle agreement performance starts decreasing, and using the window size which maximizes oracle agreement. We denote this as $m^*$-Occlusion. Note that this introduces some data contamination, which we accept because our aim is to demonstrate the efficacy of ALL, and ALL outperforms $m$-Occlusion despite the contamination.

- For $2^\text{nd}$-order $m$-Occlusion we use the optimal value of $m$ for $m$-Occlusion. We denote this $2^\text{nd}$-order $m^*$-Occlusion. We don't do another sweep because $2^\text{nd}$-order $m$-Occlusion is very computationally-expensive, requiring $\Theta(T^2)$ passes through the dataset.

- Like Yap et al. (2025), Schamberger et al. (2023) uses the attack dataset to evaluate the sensitivity of the classifier to occluding measurements. To avoid data contamination, and for compatibility with the OTiAiT and OTP datasets which do not have the same kind of attack dataset as the AES implementations, we track the average change in logits over the profiling dataset as we occlude measurements (similarly to 1-Occlusion).

### E.3.3 Hyperparameter tuning procedure

Table 9: Outcome of a 50-trial random hyperparameter search for the supervised classification models used by the deep learning baseline methods. All trials are early stopped at the point of lowest validation rank, and we choose the hyperparameter configuration which minimizes the lowest validation rank, with ties broken based on validation loss. Models are trained with AdamW and weight decay applied only to the weights of dense and conv1d layers. We use the default PyTorch settings everywhere unless otherwise stated.

| Hyperparameter | Search space | Selected value | | | | | |
|---|---|---|---|---|---|---|---|
| | | ASCADv1 (fixed) | ASCADv1 (variable) | DPAv4 (Zaid) | AES-HD | OTiAiT | OTP |
| Learning rate | $\bigcup_{m=1}^{9}\bigcup_{n=3}^{5}\{m \cdot 10^{-n}\}$ | $3 \cdot 10^{-4}$ | $2 \cdot 10^{-4}$ | $4 \cdot 10^{-3}$ | $9 \cdot 10^{-5}$ | $8 \cdot 10^{-3}$ | $6 \cdot 10^{-3}$ |
| LR schedule | {constant, cos annealing} | cos annealing | cos annealing | constant | cos annealing | cos annealing | cos annealing |
| Input dropout | $\{0.0, 0.1\}$ | 0.0 | 0.0 | 0.0 | 0.0 | 0.0 | 0.1 |
| Hidden dropout | $\{0.0, 0.2\}$ | 0.2 | 0.2 | 0.2 | 0.2 | 0.2 | 0.0 |
| Output dropout | $\{0.0, 0.3\}$ | 0.3 | 0.0 | 0.0 | 0.3 | 0.3 | 0.0 |
| Training steps | n/a | 20k | 40k | 10k | 20k | 1k | 1k |
| **Classification performance of chosen model** | | | | | | | |
| Test rank ↓ | | $101 \pm 1$ | $79.1 \pm 0.2$ | $5.6 \pm 0.7$ | $124.7 \pm 0.2$ | $1.010 \pm 0.007$ | $1.00171 \pm 0.00007$ |
| Test loss ↓ | | $5.59 \pm 0.04$ | $5.380 \pm 0.009$ | $3.0 \pm 0.2$ | $5.65 \pm 0.04$ | $0.10 \pm 0.08$ | $0.0085 \pm 0.0006$ |
| Traces to AES key disclosure ↓ | | $179 \pm 71$ | $288 \pm 126$ | $1.4 \pm 0.5$ | $4355 \pm 1915$ | n/a | n/a |

All the deep learning methods we consider require hyperparameter tuning. Work in other deep learning subfields such as Gulrajani & Lopez-Paz (2021) has emphasized the importance of a fair hyperparameter tuning process and realistic model selection criterion when comparing the performance of different algorithms, and in this work we aim to follow these recommendations. Accordingly, all methods are tuned with a 50-trial random hyperparameter search. Note that while our main performance evaluation metric is the oracle

Table 10: Outcome of a 50-trial random hyperparameter search for adversarial leakage localization. Models are trained with AdamW and weight decay applied only to the weights of the dense and conv1d layers. We use the default PyTorch settings everywhere unless otherwise stated. For ASCADv1 (fixed), ASCADv1 (variable) and AES-HD we find it helpful to 'pretrain' the classifier with fixed $\tilde{\eta}$ before beginning the simultaneous phase of training, but for DPAv4, OTiAiT, and OTP we find this unnecessary.

| Hyperparameter | Search space | Selected value | | | | | |
| --- | --- | --- | --- | --- | --- | --- | --- |
| | | ASCADv1 (fixed) | ASCADv1 (variable) | DPAv4 (Zaid) | AES-HD | OTiAiT | OTP |
| $\boldsymbol{\theta}$ learning rate (pretrain) | $\bigcup_{m=1}^{10}\{m\cdot 10^{-4}\}$ | $10^{-4}$ | $10^{-4}$ | n/a | $10^{-3}$ | n/a | n/a |
| $\overline{\gamma}$ (pretrain) | n/a | 0.5 | 0.5 | n/a | 0.5 | n/a | n/a |
| $\boldsymbol{\theta}$ learning rate | $\bigcup_{m=1}^{9}\bigcup_{n=4}^{6}\{m\cdot 10^{-n}\}$ | $6\cdot 10^{-6}$ | $7\cdot 10^{-5}$ | $2\cdot 10^{-5}$ | $10^{-4}$ | $10^{-4}$ | $4\cdot 10^{-4}$ |
| $\frac{\tilde{\boldsymbol{\eta}}\text{ learning rate}}{\boldsymbol{\theta}\text{ learning rate}}$ | $\bigcup_{m=1}^{9}\bigcup_{n=0}^{2}\{m\cdot 10^{n}\}$ | 50 | 6 | 9 | 20 | 3 | 3 |
| $\overline{\gamma}$ | $\bigcup_{m=1}^{19}\{0.05\cdot m\}$ | 0.3 | 0.4 | 0.7 | 0.85 | 0.8 | 0.65 |
| Training steps (pretrain) | n/a | 10k | 20k | 0 | 10k | 0 | 0 |
| Training steps | n/a | 10k | 20k | 10k | 10k | 1k | 1k |

agreement, it would be unrealistic to use this for model selection, even when computed on a validation dataset, because it relies on 'white box' knowledge about cryptographic implementations that we assume not to have at training time. Instead, for ALL we use the model selection criterion proposed in Sec. E.3.5. The prior deep learning methods are based on 'interpreting' a classifier trained with supervised learning, and in line with prior work we tune its associated hyperparameters to optimize classification performance via minimizing the early-stopped mean rank on a validation dataset. We also visualize the distribution of results over the hyperparameter sweep for each method.

ALL is sensitive to the noise budget parameter $\overline{\gamma}$ and the learning rates of the classifier weights $\boldsymbol{\theta}$ and the noise distribution parameter $\tilde{\boldsymbol{\eta}}$. We consistently find that $\tilde{\boldsymbol{\eta}}$ should have a higher learning rate than $\boldsymbol{\theta}$, so in order to focus the search on better hyperparameter configurations we tune the *ratio* of the learning rate of $\tilde{\boldsymbol{\eta}}$ to that of $\boldsymbol{\theta}$ rather than the learning rate of $\tilde{\boldsymbol{\eta}}$. For ASCADv1 (fixed, variable) and AES-HD we find that performance is much better if we pretrain the classifier for half of the training steps with fixed $\tilde{\boldsymbol{\eta}} = \mathbf{0}$ and noise budget $\overline{\gamma} = 0.5$, then proceed as normal for the next half. Thus, for these trials, we first do a 10-trial grid search of the learning rate for $\boldsymbol{\theta}$ to minimize mean rank during this pretraining phase, then tune all hyperparameters as normal using this trained classifier as the starting point for the remaining 40 trials. In preliminary experiments we explored tuning the $\beta_1$, $\beta_2$, $\epsilon$ and weight decay strength of the AdamW optimizer as well as the number of $\tilde{\boldsymbol{\eta}}$ steps per $\boldsymbol{\theta}$ step, but chose to leave these fixed for the final search because they had little impact on performance. See Table 10 for the hyperparameter search space and chosen configurations for ALL.

The deep learning baselines are based on 'interpreting' a trained supervised classifier and require tuning this classifier – we tune its learning rate, dropout rates, and decide whether to use a constant or cosine decay learning rate schedule. In preliminary experiments we also explored tuning the $\beta_1$, $\beta_2$, $\epsilon$ and weight decay strength $\lambda$ of the AdamW optimizer, but chose to leave them at fixed values for the final search because they had little effect on classification performance. See Table 9 for the hyperparameter search space, chosen configurations, and resulting classification performance for the supervised classifiers.

### E.3.4 Performance evaluation methods

Unlike for the experiments on synthetic datasets, here we lack ground truth knowledge about the leakiness of individual measurements. It is challenging to evaluate the performance of leakage localization algorithms in this setting, and there is currently no consensus about the best way to do so. We consider 4 quantitative performance evaluation strategies which are conceptually-similar to performance evaluation strategies of prior work. To account for the varying 'shapes' of leakage assessments returned by the compared methods, all of our evaluation metrics are sensitive only to the *relative* leakiness assigned to measurements.

**'Oracle' leakiness via SNR with relevant leaking first-order variables**  The present work is concerned with 'black box' leakage localization algorithms which require only a supervised learning-style dataset of traces and associated target variable values, and minimal *a priori* knowledge about the cryptographic

implementations being analyzed. However, it is also possible to analyze devices in a 'white box' manner which does incorporate *a priori* knowledge. In particular, while second-order datasets such as ASCADv1 (fixed and variable) are not amenable to black box analysis with the first-order parametric methods such as SNR, it is possible to study the implementation and 'decompose' the second-order leakage into first-order leakage of *pairs* of internal variables, then use parametric methods to analyze leakage of these variables individually. As our main performance evaluation strategy, we use such a white box analysis to compute per-measurement leakiness predictions, which we treat as an 'oracle' against which to compare output. This is a useful way to validate deep learning methods because 1) it is interpretable and hyperparameter-free, and 2) it lets us check whether an output is consistent with an analysis which a domain expert might do.

For the ASCADv1 datasets, we use the white box analysis of Egger et al. (2022) to compute 'oracle' leakiness estimates for each of the measurements. We use the canonical target variable for both datasets, which is $\text{Sbox}(k_2 \oplus w_2)$ where Sbox is an invertible function which is publicly-known and shared by all AES implementations, $\oplus$ denotes the bitwise exclusive-or operation and $k_2$ and $w_2$ denote byte 2 of the key and plaintext, respectively, with indexing starting from 0. The underlying AES-128 implementation of these datasets uses Boolean masking, as shown in Alg. 1 of Benadjila et al. (2020). By design, this Boolean masking prevents the algorithm from ever directly operating on $\text{Sbox}(k_2 \oplus w_2)$, making all power measurements $X_t$ nearly marginally statistically-independent of $\text{Sbox}(k_2 \oplus w_2)$.

Alg. 1 does directly operate on the following pairs of variables: $(r_2, \text{Sbox}(k_2 \oplus w_2) \oplus r_2)$, $(r_{\text{out}}, \text{Sbox}(k_2 \oplus w_2) \oplus r_{\text{out}})$, $(r_{\text{in}}, k_2 \oplus w_2 \oplus r_{\text{in}})$. The variables $r_2, r_{\text{out}}, r_{\text{in}}$ are called *masks* and are internal variables which are randomly generated during each encryption. Thus, each of these variables *is* marginally dependent on some measurements $X_t$ and can be detected with first-order parametric methods. Each of these pairs is said to leak $\text{Sbox}(k_2 \oplus w_2)$ because given both, one can calculate $\text{Sbox}(k_2 \oplus w_2)$ using the identity $a \oplus b \oplus b = a$. In addition to these pairs of variables, Egger et al. (2022) identified that the variables $\text{Sbox}(S_{\text{prev}} \oplus r_{\text{in}}) \oplus r_{\text{out}}$ and a 'security load' $S_{\text{prev}} \oplus \text{Sbox}(w_2 \oplus k_2) \oplus r_{\text{out}}$ also have a strong first-order association with power consumption and might contribute to leakage, where for byte 2 $S_{\text{prev}} = \text{Sbox}(k_{11} \oplus w_{11}) \oplus r_{11}$.

All 8 of these internal variables may be computed using the metadata published with the ASCADv1 datasets. As our oracle assessment for the ASCADv1 datasets, we compute the SNR of each of these 8 variables using the attack dataset and average them together. We can then qualitatively assess agreement between the output of leakage localization algorithms and these oracle assessments. For the 4 first-order datasets, we use as our oracle assessment the SNR of the target itself from the attack dataset, which amounts to an assumption that there is no leakage of order 2 or higher.

We use the Spearman rank correlation coefficient as a scalar summary of this agreement. This quantity is defined as the Pearson correlation coefficient between the *ranks* of a pair of sequences, and is useful for our purposes because it tells us the extent to which leakage localization algorithms assign the same *relative* leakiness to measurements as the oracle, while being insensitive to differences in their 'shape'.

Most prior work (Masure et al., 2019; Wouters et al., 2020; Schamberger et al., 2023; Yap et al., 2025) has used white box assessments similar to this for qualitative evaluation of leakage localization algorithms. To our knowledge, ours is the first work to summarize agreement with a scalar and use it for large-scale comparison between a large number of methods.

We refer to this performance evaluation strategy as *oracle agreement*. Note that we use the word 'oracle' for clarity of exposition, and we believe this is the least-flawed of the evaluation metrics we consider, but it does not give us genuine ground truth leakiness measurements. It is sensitive only to first-order leakage of variables which can be identified *a priori* as leaky, and will ignore any other exploitable variables. Additionally, SNR is not perfectly sensitive even to first-order leakage: it relies on changes in the expected values $\mathbb{E}[X_t \mid Y = y]$ with $y$ and will not detect dependencies which do not influence the mean (e.g. if $X_t$ is a Gaussian random variable with $Y$-independent mean but $Y$-dependent variance).

**DNN occlusion tests**    Hettwer et al. (2020) proposed a variety of tests based on plotting the performance of a trained supervised classifier as its input features are successively occluded in order of their leakiness. The intuition is that leakier features should have a larger impact on the performance of the classifier, so the rate at which its performance changes as we successively occlude its inputs tells about the extent to which

---

**Algorithm 4:** Pseudocode for the DNN occlusion tests.

---

**Input:** Trained supervised classifier $\Phi^* : \mathbb{R}^T \times \mathsf{Y} \to [0,1]$, attack dataset $\mathsf{D}_{\text{attack}} \subseteq \mathbb{R}^T \times \mathsf{Y}$, leakiness estimates $\boldsymbol{\ell} \in \mathbb{R}^T$, direction $d \in \{\text{'forward', 'reverse'}\}$
**Output:** Area under DNN occlusion curve $\bar{r}$

---

1   $\boldsymbol{m}_0 \leftarrow \boldsymbol{0}$                               `// occlusion mask`
2   $\boldsymbol{\ell}_{\text{idx}} \leftarrow \texttt{argsort}(\boldsymbol{\ell})$          `// indices of sorted leakiness values, from low-high`
3   **if** $d = $ 'forward' **then**
4      |   $\boldsymbol{\ell}_{\text{idx}} \leftarrow \texttt{reverseOrder}(\boldsymbol{\ell}_{\text{idx}})$             `// sort from high-low instead`
5   **else if** $d = $ 'reverse' **then**
6      |   pass
7   **for** $t = 1, \ldots, T$ **do**
8      |   $\boldsymbol{m}_t \leftarrow \boldsymbol{m}_{t-1} + \boldsymbol{I}_{\ell_{\text{idx},\,t}}$ `// un-occlude` $t$`-th least (reverse) or most (forward)-leaky feature`
9      |   $r_t \leftarrow \frac{1}{|\mathsf{D}_{\text{attack}}|} \sum_{\boldsymbol{x},y \in \mathsf{D}_{\text{attack}}} \text{Rank}\left(\Phi^*(\cdot \mid (\boldsymbol{1} - \boldsymbol{m}_t) \odot \boldsymbol{x}); y\right)$          `// record average classifier`
        |   `performance on attack dataset under this occlusion mask`
10 **return** $\frac{1}{T} \sum_{t=1}^{T} r_t$                      `// area under the DNN occlusion curve`

---

these inputs were leaky. In a similar spirit, we propose 2 evaluation metrics which we name the *forward* and *reverse DNN occlusion tests*.

See Alg. 4. For the *forward DNN occlusion test* we initially occlude all the input features of a trained classifier, then successively un-occlude one feature at a time from most- to least-leaky as predicted by the leakage localization algorithm under test. At each occlusion level we measure the performance of the classifier on the attack dataset in terms of mean rank. We then report the average performance across all occlusion levels. For a 'good' leakage assessment we expect the average performance to be better (lower) because useful features are un-occluded at a greater proportion of occlusion levels. Conversely, for a 'bad' leakage assessment we expect the average performance to be worse because useful features stay occluded for longer. The *reverse DNN occlusion test* is identical except that we un-occlude features from least- to most-leaky. For this test we expect the average performance to be worse (higher) for a 'good' leakage assessment and better (lower) for a 'good' leakage assessment. In general we expect the forward test to be sensitive to the extent to which the predicted-leakiest measurements are truly among the leakiest (similar to true/false positives), and the reverse test to be sensitive to the extent to which the predicted-nonleaky features are truly nonleaky (similar to true/false negatives).

A major limitation of the DNN occlusion tests is that they rely on an imperfect DNN classifier, and are only sensitive to associations insofar as the classifier exploits them. Additionally, we use the same architecture, training procedure and hyperparameters for these classifiers as for those 'interpreted' by the neural net attribution baseline methods, so the test may be 'biased' in favor of these. Nonetheless, we consider them a useful supplement to the oracle agreement metric because they do not suffer from the same restrictive assumptions about the nature of associations.

**Feature selection efficacy for Gaussian template attack**   Similarly to Masure et al. (2019); Yap et al. (2023), we also evaluate leakage localization assessments based on their ability to do feature selection for Gaussian template attacks. To carry out this test, we first select the top 20 measurements with the highest estimated leakiness. We then perform a Gaussian template attack (Chari et al., 2003) using these measurements. The Gaussian template attack is a well-known parametric side-channel attack based on modeling $p_{\boldsymbol{X}|Y}$ with a Gaussian mixture model with one component per value $Y$ may take on, then using Bayes' rule to estimate $p_{Y|\boldsymbol{X}}$. The leakier these features are, the more-performant we expect the attack to be.

For AES datasets accuracy is often low when predicting $Y$ from a single value of $\boldsymbol{X}$. Thus, attack datasets typically consist of many traces $\boldsymbol{X}_1, \ldots, \boldsymbol{X}_M$ recorded with a *fixed* AES key but varying plaintext. The target variable $Y$ is typically chosen so that given the corresponding plaintext and ciphertext, $Y$ is a known invertible function of the key. One can thereby make many predictions about the key using these traces, then 'accumulate' these predictions through the identity $\log p_{\boldsymbol{X}_1, \ldots, \boldsymbol{X}_M | K} = \sum_{m=1}^{M} \log p_{\boldsymbol{X}_m | K}$ where $K$ denotes the

key. A common performance metric for attacks is the *minimum traces to disclosure (MTD)*, given by the number of traces one must accumulate before the true key has the highest predicted probability mass (lower is better). We use this metric to measure the performance of Gaussian template attacks on AES datasets. For OTiAiT and OTP, which are not AES datasets, we simply use the mean rank of the target variable on the attack dataset.

Note that for the second-order ASCADv1 datasets, the algorithms we consider do not reveal which of the leaky internal AES variables a measurement leaks. This is problematic when selecting features for a template attack, because a successful attack must have features corresponding to both of a pair of second-order leaky variables. If we simply used the top 20 predicted-leakiest features for an attack, this would be left to chance and make the performance metric unreliable. To address this issue, for the second-order datasets we instead segment the $T$ measurements into 10 bins each containing $\lfloor \frac{T}{10} \rfloor$ consecutive measurements, then select the top 2 leakiest measurements from each bin.

The main shortcoming of this performance metric is that it is only sensitive to the 20 predicted-leakiest measurements and ignores all others (e.g. it cannot detect that leaky measurements have been assigned spuriously-low leakiness). Additionally, it assumes that the relationship between measurements and the target variable is well-described by a Gaussian mixture model, which may not hold in practice. However, unlike the oracle agreement metric it does not rely on human-identified first-order leaky variables, and unlike the DNN occlusion tests it is hyperparameter-free and may be biased towards different associations than the DNN classifiers. Thus, it is also a useful supplement to the aforementioned metrics.

### E.3.5   Model selection criterion

While we consider the oracle agreement metric our most straightforward and useful performance metric, we cannot use it for model selection (e.g. choosing hyperparameters, or early-stopping runs). This is because it relies on 'white box' knowledge of the internal first-order leaky variables of second-order algorithms and knowledge of their random masks, which we assume not to have at training time. Thus, we must devise a model selection criterion which does not rely on this information.

Note that we can freely use the forward and reverse DNN occlusion tests and the template attack feature selection test for model selection by running them on our validation dataset rather than the attack dataset. Additionally, we find that because 'good' runs typically converge to similar leakiness assessments whereas 'bad' runs resemble random noise, a reasonable model selection heuristic is to 1) compute the average leakiness value for each measurement over all hyperparameter tuning runs, then 2) use the Spearman rank correlation coefficient between the average leakiness assessments and those for a particular run as a proxy for the run's performance. We refer to this model selection strategy as the *mean agreement criterion.*

In Fig. 13 we visualize the relationship between the oracle agreement and the forward and reverse DNN occlusion tests as well as the mean agreement criterion. We find that the forward and reverse DNN occlusion criterion are weakly correlated with oracle agreement, and that the mean agreement is strongly correlated for every dataset apart from OtiAiT. However, while we can consistently discard bad runs using these criteria, they typically lead to selection of suboptimal models. In this work we select ALL models using a 'composite criterion' based on ranking runs according to the forward and reverse DNN occlusion tests and mean agreement criterion, then selecting the model with the highest mean ranking across these 3 criteria. More research into model selection strategies for leakage localization algorithms is warranted.

### E.3.6   Summary of experiments

We report and visualize our results in a variety of ways, which we list and summarize here.

**Visualization of the best-performing leakage localization results found by ALL**   In Fig. 14 we visualize the oracle leakiness measurements and qualitatively compare them with the best ALL runs found during our hyperparameter sweeps. For the ASCADv1 datasets we draw distinct curves for the individual leaky variables – note the similarity between these plots and (Egger et al., 2022, Fig. 3a). We see a strong visual resemblance between the ALL outputs and the oracle leakiness assessments, despite the nonlinear relationship between them. Additionally, there are few points assigned a low oracle leakiness but high *relative*

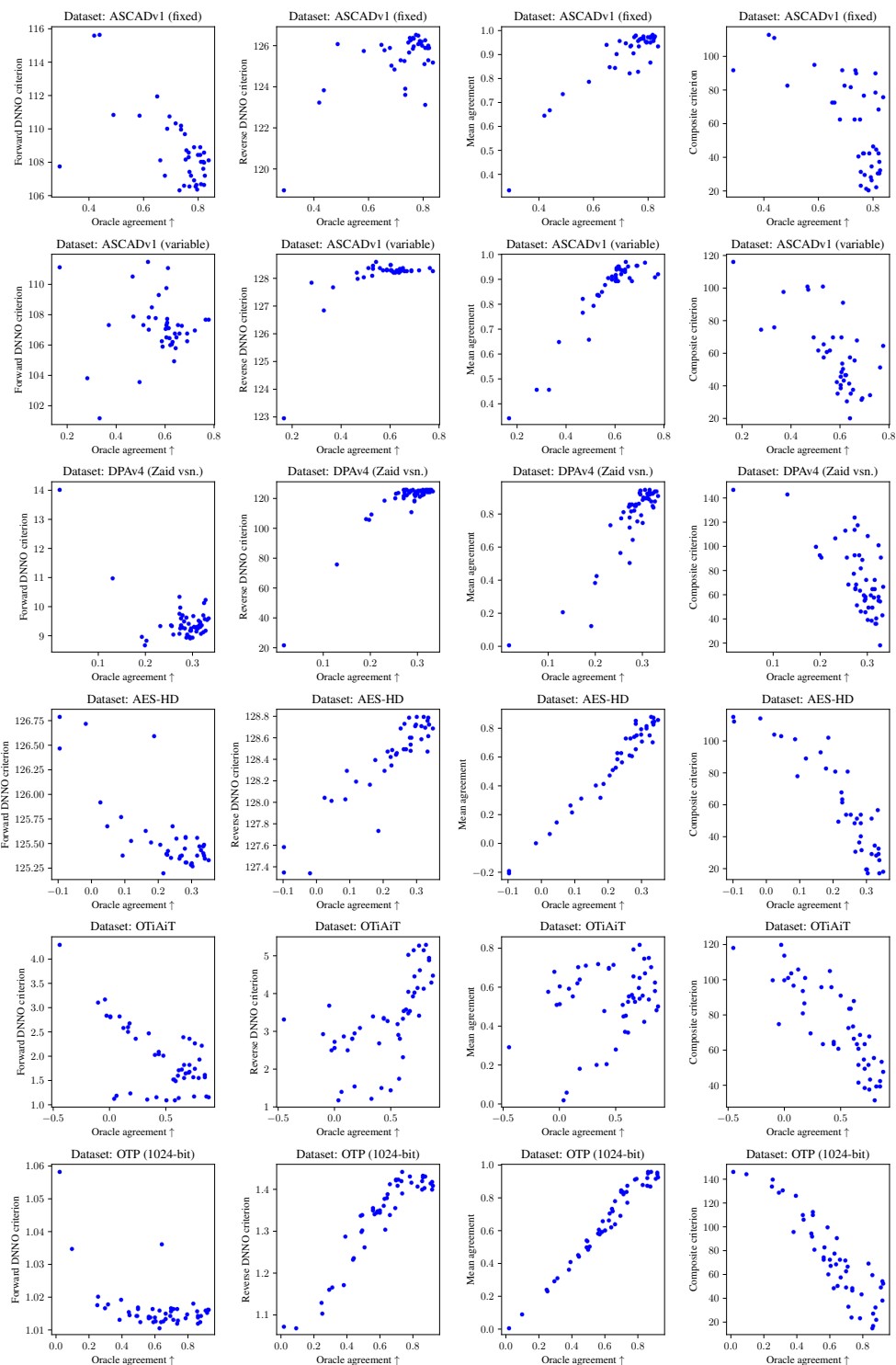

Figure 13: Visualization of the relationship between various model selection criteria from Sec. E.3.5 and the oracle agreement for ALL runs produced during hyperparameter search. We find consistently across datasets that the forward/reverse DNN occlusion and mean agreement criteria are consistently positively or negatively correlated with the oracle agreement, though this correlation is often weak. We achieve slightly better results using a composite criterion which uses considers the 'votes' according to all these criterion, and we adopt this composite criterion when selecting ALL models for comparison with baselines. However, this criterion often selects suboptimal models, and future research on leakage localization model selection strategies is warranted.

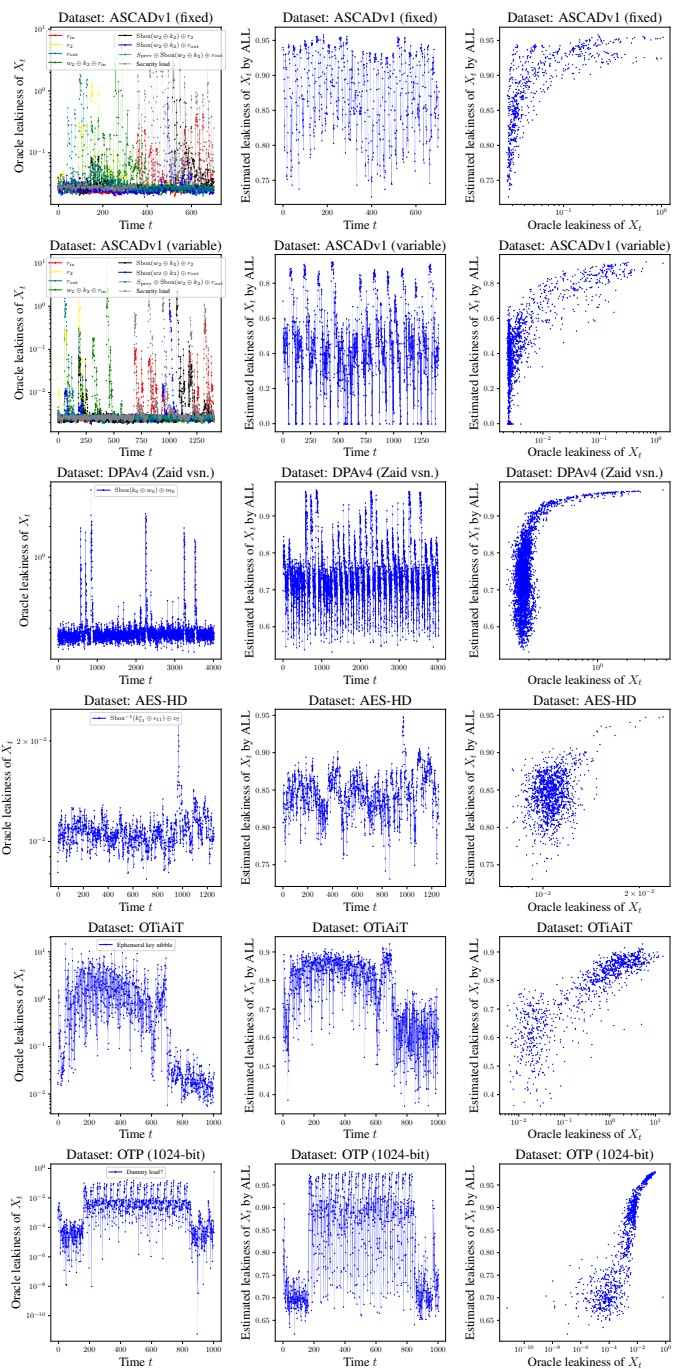

Figure 14: A plot which qualitatively compares the estimated leakiness of the best-performing ALL runs with the oracle leakiness values. (**left column**) A plot of oracle leakiness of $X_t$ vs. timestep $t$. For the second-order ASCADv1 datasets (top two rows), note that our plots are similar to (Egger et al., 2022, Fig. 3a) as they are based on the same first-order variables; differences are because we measure leakiness with SNR whereas they use CPOI. (**middle column**) A plot of estimated leakiness of $X_t$ vs. timestep $t$ according to ALL, for the best-performing ALL run as measured by oracle agreement. We want this column to look like the left column, up to a strictly-increasing nonlinear transform. (**right column**) A plot of the ALL-estimated leakiness of $X_t$ vs. the oracle leakiness of $X_t$. These curves show that the ALL estimates are good in the sense that they tend to apply similar *relative* leakiness values to the measurements as the oracle.

ALL-predicted leakiness or vice-versa. Most of the disagreement appears to lie in the predicted relative leakiness within groups of high-leakiness or low-leakiness measurements.

**ALL training curves**   In Fig. 15 we plot the evolution of various metrics over time during the ALL training procedure for the runs chosen using the model selection criterion of Sec. E.3.5. In Fig. 16 we compare the oracle agreement at different timesteps on ASCADv1-variable for ALL and selected baselines. In general we observe that for successful ALL runs the classifier validation rank drops significantly below the random-guessing threshold at some point in training, though it may begin to rise again as the noise distribution trains adversarially against it. These curves are generally smooth and we do not observe significant training instability. Unfortunately, we are not aware of a reliable way to predict oracle agreement performance from the training curves.

**Sensitivity of ALL to hyperparameters**   The main hyperparameters of ALL are the noise budget $\overline{\gamma}$ and the learning rates of the classifier weights $\boldsymbol{\theta}$ and the noise distribution parameter $\tilde{\boldsymbol{\eta}}$. In Fig. 17 we evaluate the sensitivity of ALL to these hyperparameters by varying them individually using the optimal configuration with respect to oracle agreement as a starting point. We find in general that performance generally varies smoothly with these hyperparameters and stays significantly above random-guessing over a large search space, which is desirable from a standpoint of hyperparameter tuning.

**Attack performance and training curves of supervised classifiers**   All of the deep learning-based baseline methods are based on 'interpreting' a fixed classifier which has been trained using supervised learning to predict the target variable $Y$ from the trace $\boldsymbol{X}$. In Fig. 18 we plot the cross-entropy loss and mean rank over time for these classifiers during training. Additionally, for the AES datasets we plot the rank of the correct key as we accumulate traces from the attack dataset. For reference we superimpose the correct key rank during trace accumulation for the following publicly-available open-weight classifiers: the $\text{CNN}_{\text{Best}}$ and $\text{MLP}_{\text{Best}}$ model of Benadjila et al. (2020), both of which are available here, and the models of Zaid et al. (2020) and their simplified versions from Wouters et al. (2020) distributed here. Also note that in Table 9 we list the early-stopped validation cross-entropy loss, rank, and minimum traces to disclosure (MTD) for these models. We find that our classifiers are able to 'successfully' attack all datasets (i.e. they can successfully predict the key by accumulating all traces in the provided attack dataset), and they achieve comparable MTD to these open-weight models on ASCADv1-fixed, ASCADv1-variable, and DPAv4, and somewhat-worse MTD on the AES-HD dataset.

**$m$-Occlusion window size sweep and smoothing effect**   We consider as baselines $m$-Occlusion and $2^{\text{nd}}$-order $m$-Occlusion. These baselines introduce the occlusion window size as an additional hyperparameter which must be tuned. In Fig. 19 we plot the oracle agreement performance of $m$-Occlusion as we sweep $m$, and qualitatively show how the resulting leakiness vector is smoothed out with increasing $m$. In subsequent experiments, we denote by $m^*$ the optimal value of $m$ found in these experiments, and report results for 1-Occlusion, $m^*$-Occlusion, $2^{\text{nd}}$-order 1-Occlusion and $2^{\text{nd}}$-order $m^*$-Occlusion (due to the high cost of $2^{\text{nd}}$-order $m$-Occlusion we do not separately sweep its window size). Note that this introduces some data leakage into the results, as we are doing validation with our test metric which incorporates implementation knowledge that we assume not to have at training time. Because the goal of this work is to demonstrate the efficacy of ALL, and ALL generally outperforms $m^*$-Occlusion and $2^{\text{nd}}$-order $m^*$-Occlusion despite the data leakage, we consider this acceptable.

Compared to 1-Occlusion, $m$-Occlusion has two major differences: it occludes multiple inputs simultaneously to increase the influence on classifier predictions, and it has a 'smoothing' effect which causes nearby measurements to be assigned similar leakiness values. The latter effect can be easily simulated using average-pooling, so we also plot the performance of ALL as we average-pool it with stride 1 and kernel size $m$. We find that while $m$-Occlusion significantly improves performance over 1-Occlusion on the DPAv4 and AES-HD datasets, on these same datasets we can significantly improve the performance of ALL by average-pooling, and ALL convincingly outperforms $m$-Occlusion when accounting for this. We conjecture that the smoothing effect provides a useful inductive bias for these datasets, and emphasize that it can easily be applied to any other leakage localization technique.

Table 11: Performance comparison between various leakage localization algorithms according to the oracle agreement metric (**larger is better**) described in Sec. E.3.4. Results are reported as mean $\pm$ standard deviation over 5 random seeds. The best result is $\boxed{\text{boxed}}$ and the best deep learning result is underlined. We consider a result to be 'best' if its mean lies inside of the error bars of the result with the highest mean.

| Method | 2nd-order datasets | | 1st-order datasets | | | |
| | ASCADv1 (fixed) | ASCADv1 (random) | DPAv4 (Zaid vsn.) | AES-HD | OTiAiT | OTP (1024-bit) |
|---|---|---|---|---|---|---|
| Random | $-0.00 \pm 0.04$ | $-0.02 \pm 0.01$ | $0.01 \pm 0.01$ | $-0.01 \pm 0.02$ | $0.02 \pm 0.03$ | $0.03 \pm 0.03$ |
| SNR | $0.031$ | $-0.092$ | $0.344$ | $0.185$ | $\boxed{0.989}$ | $0.944$ |
| SOSD | $-0.253$ | $0.272$ | $0.259$ | $0.063$ | $0.886$ | $0.803$ |
| CPA | $0.521$ | $-0.095$ | $\boxed{0.420}$ | $\boxed{0.303}$ | $0.630$ | $\boxed{0.945}$ |
| GradVis | $0.48 \pm 0.02$ | $0.27 \pm 0.01$ | $0.198 \pm 0.009$ | $0.07 \pm 0.01$ | $0.55 \pm 0.05$ | $0.57 \pm 0.02$ |
| Saliency | $0.47 \pm 0.02$ | $0.26 \pm 0.01$ | $0.198 \pm 0.008$ | $0.07 \pm 0.01$ | $0.67 \pm 0.06$ | $0.58 \pm 0.02$ |
| Input $*$ Grad | $0.47 \pm 0.02$ | $0.25 \pm 0.01$ | $0.202 \pm 0.009$ | $0.08 \pm 0.02$ | $0.71 \pm 0.05$ | $0.60 \pm 0.02$ |
| LRP | $0.47 \pm 0.02$ | $0.25 \pm 0.01$ | $0.202 \pm 0.009$ | $0.08 \pm 0.02$ | $0.71 \pm 0.05$ | $0.60 \pm 0.02$ |
| OccPOI | $0.07 \pm 0.01$ | $0.064 \pm 0.004$ | $0.030 \pm 0.008$ | $0.044 \pm 0.009$ | $0.07 \pm 0.02$ | $0.01 \pm 0.02$ |
| 1-Occlusion | $0.47 \pm 0.02$ | $0.25 \pm 0.01$ | $0.202 \pm 0.009$ | $0.08 \pm 0.01$ | $0.71 \pm 0.05$ | $0.60 \pm 0.02$ |
| $m^*$-Occlusion | $0.49 \pm 0.02$ | $0.41 \pm 0.01$ | $\underline{0.32 \pm 0.01}$ | $0.18 \pm 0.05$ | $0.72 \pm 0.04$ | $0.77 \pm 0.01$ |
| 1-Occlusion$^2$ | $0.51 \pm 0.01$ | $0.27 \pm 0.01$ | $0.206 \pm 0.009$ | $0.08 \pm 0.01$ | $0.74 \pm 0.05$ | $0.60 \pm 0.02$ |
| $m^*$-Occlusion$^2$ | $0.52 \pm 0.01$ | $0.42 \pm 0.01$ | $\underline{0.330 \pm 0.009}$ | $0.19 \pm 0.05$ | $0.75 \pm 0.04$ | $0.788 \pm 0.007$ |
| WoutersNet 1-Occlusion | $0.18 \pm 0.03$ | n/a | $0.21 \pm 0.02$ | $0.11 \pm 0.03$ | n/a | n/a |
| WoutersNet $m^*$-Occlusion | $0.20 \pm 0.03$ | n/a | $0.29 \pm 0.02$ | $\underline{0.21 \pm 0.04}$ | n/a | n/a |
| WoutersNet GradVis | $0.19 \pm 0.03$ | n/a | $0.21 \pm 0.02$ | $0.11 \pm 0.03$ | n/a | n/a |
| WoutersNet Input $*$ Grad | $0.18 \pm 0.03$ | n/a | $0.21 \pm 0.02$ | $0.11 \pm 0.03$ | n/a | n/a |
| WoutersNet Saliency | $0.19 \pm 0.03$ | n/a | $0.21 \pm 0.02$ | $0.11 \pm 0.03$ | n/a | n/a |
| ZaidNet 1-Occlusion | $0.24 \pm 0.03$ | n/a | $0.19 \pm 0.01$ | $0.13 \pm 0.02$ | n/a | n/a |
| ZaidNet $m^*$-Occlusion | $0.25 \pm 0.04$ | n/a | $0.273 \pm 0.009$ | $\underline{0.21 \pm 0.05}$ | n/a | n/a |
| ZaidNet GradVis | $0.25 \pm 0.03$ | n/a | $0.19 \pm 0.01$ | $0.13 \pm 0.02$ | n/a | n/a |
| ZaidNet Input $*$ Grad | $0.25 \pm 0.04$ | n/a | $0.19 \pm 0.01$ | $0.13 \pm 0.02$ | n/a | n/a |
| ZaidNet Saliency | $0.25 \pm 0.03$ | n/a | $0.19 \pm 0.01$ | $0.13 \pm 0.02$ | n/a | n/a |
| ALL (ours) | $\boxed{0.794 \pm 0.006}$ | $\boxed{0.60 \pm 0.01}$ | $0.317 \pm 0.002$ | $\underline{0.22 \pm 0.03}$ | $\underline{0.782 \pm 0.001}$ | $\underline{0.848 \pm 0.003}$ |

**Quantitative comparison between ALL and baseline methods using oracle agreement, forward and reverse DNN occlusion tests, and template attack feature selection test** In Tables 11, 12, 13, 14 we compare the performance of ALL with our baseline methods according to the oracle agreement, forward DNN occlusion, reverse DNN occlusion, and template attack feature selection tests, respectively. As a sanity check against our supervised neural net architecture and training procedure, we also include results for GradVis, Saliency, LRP, Input $*$ Grad, 1-Occlusion, and $m^*$-Occlusion computed using the models of Zaid et al. (2020) and their simplified versions from Wouters et al. (2020) distributed here.

We find that ALL outperforms all prior deep learning-based leakage localization algorithms on all datasets except for DPAv4 according to the oracle agreement metric. The first-order parametric methods outperform the deep learning methods on the first-order datasets but generally do poorly on the second-order ASCADv1 datasets due to not being sensitive to second-order associations. Unsurprisingly, SNR and SOSD achieve near-random performance on ASCADv1-fixed and SNR and CPA achieve near-random performance on ASCADv1-variable. Surprisingly, CPA performs fairly well on ASCADv1-fixed and SOSD performs fairly well on ASCADv1-variable. We are not sure why this is the case, but conjecture it is because these methods have some natural proclivity to 'rule out' measurements which are not at useful points in time for these particular datasets (e.g. those which are not close to a clock edge). Note that this surprisingly-strong performance compared to the deep learning baselines does not appear to carry over to the evaluations with the DNN occlusion tests or the template attack feature selection test.

On average, ALL is the best method on the majority of datasets according to the reverse DNN occlusion test and template attack test, but results are mixed according to the forward DNN occlusion test. However, we find that the DNN occlusion tests have high variance, and in general there is a large overlap in error bars (note the large number of boxed and underlined methods in Tables 12 and 13).

Table 12: Performance comparison between various leakage localization algorithms according to the forward DNN occlusion test (**smaller is better**) described in Sec. E.3.4. Results are reported as mean ± standard deviation over 5 random seeds. The best result is $\boxed{\text{boxed}}$ and the best deep learning result is underlined. We consider a result to be 'best' if its mean lies inside of the error bars of the result with the highest mean.

| Method | 2nd-order datasets | | 1st-order datasets | | | |
| --- | --- | --- | --- | --- | --- | --- |
| | ASCADv1 (fixed) | ASCADv1 (random) | DPAv4 (Zaid vsn.) | AES-HD | OTiAiT | OTP (1024-bit) |
| Random | $111 \pm 1$ | $111.7 \pm 0.8$ | $20 \pm 2$ | $126.8 \pm 0.1$ | $1.30 \pm 0.05$ | $1.065 \pm 0.010$ |
| SNR | 117.525 | 118.448 | 11.275 | $\boxed{125.053}$ | $\boxed{1.209}$ | 1.015 |
| SOSD | 115.516 | 106.072 | 11.455 | 125.365 | 1.213 | 1.032 |
| CPA | 111.449 | 117.349 | 11.811 | 125.267 | 2.049 | 1.015 |
| GradVis | $108.6 \pm 0.5$ | $96.8 \pm 0.3$ | $9.5 \pm 0.6$ | $125.6 \pm 0.3$ | $1.9 \pm 0.2$ | $\boxed{1.013 \pm 0.002}$ |
| Saliency | $108.5 \pm 0.4$ | $96.3 \pm 0.4$ | $9.5 \pm 0.7$ | $125.6 \pm 0.3$ | $\underline{1.8 \pm 0.1}$ | $\boxed{1.014 \pm 0.001}$ |
| Input $*$ Grad | $108.5 \pm 0.4$ | $96.8 \pm 0.4$ | $9.4 \pm 0.7$ | $125.6 \pm 0.3$ | $\underline{1.7 \pm 0.2}$ | $\boxed{1.013 \pm 0.001}$ |
| LRP | $108.5 \pm 0.4$ | $96.8 \pm 0.4$ | $9.4 \pm 0.7$ | $125.6 \pm 0.3$ | $\underline{1.7 \pm 0.2}$ | $\boxed{1.013 \pm 0.001}$ |
| OccPOI | $122.3 \pm 0.8$ | $120.8 \pm 0.2$ | $58 \pm 2$ | $127.4 \pm 0.3$ | $2.6 \pm 0.2$ | $1.09 \pm 0.04$ |
| 1-Occlusion | $108.5 \pm 0.4$ | $96.7 \pm 0.4$ | $9.4 \pm 0.7$ | $125.6 \pm 0.3$ | $\underline{1.7 \pm 0.2}$ | $\boxed{1.013 \pm 0.001}$ |
| $m^*$-Occlusion | $108.2 \pm 0.5$ | $\boxed{95.7 \pm 0.6}$ | $\boxed{9.0 \pm 0.6}$ | $\underline{125.3 \pm 0.2}$ | $1.8 \pm 0.2$ | $\boxed{1.013 \pm 0.001}$ |
| 1-Occlusion$^2$ | $108.4 \pm 0.4$ | $97.0 \pm 0.4$ | $9.4 \pm 0.7$ | $125.5 \pm 0.3$ | $\underline{1.7 \pm 0.2}$ | $\boxed{1.013 \pm 0.001}$ |
| $m^*$-Occlusion$^2$ | $108.2 \pm 0.5$ | $\boxed{95.9 \pm 0.6}$ | $\boxed{9.0 \pm 0.5}$ | $\underline{125.3 \pm 0.2}$ | $\underline{1.7 \pm 0.2}$ | $\boxed{1.013 \pm 0.001}$ |
| WoutersNet 1-Occlusion | $109.1 \pm 0.7$ | n/a | $\boxed{8.7 \pm 0.8}$ | $\underline{125.4 \pm 0.3}$ | n/a | n/a |
| WoutersNet $m^*$-Occlusion | $109.0 \pm 0.8$ | n/a | $\boxed{9.0 \pm 0.7}$ | $\underline{125.4 \pm 0.2}$ | n/a | n/a |
| WoutersNet GradVis | $109.3 \pm 0.8$ | n/a | $\boxed{8.8 \pm 0.8}$ | $\underline{125.4 \pm 0.3}$ | n/a | n/a |
| WoutersNet Input $*$ Grad | $109.1 \pm 0.7$ | n/a | $\boxed{8.7 \pm 0.8}$ | $\underline{125.4 \pm 0.3}$ | n/a | n/a |
| WoutersNet Saliency | $109.2 \pm 0.8$ | n/a | $\boxed{8.7 \pm 0.8}$ | $\underline{125.4 \pm 0.3}$ | n/a | n/a |
| ZaidNet 1-Occlusion | $110.0 \pm 0.4$ | n/a | $\boxed{8.7 \pm 0.7}$ | $125.5 \pm 0.2$ | n/a | n/a |
| ZaidNet $m^*$-Occlusion | $109.5 \pm 0.9$ | n/a | $\boxed{9.0 \pm 0.7}$ | $\underline{125.4 \pm 0.2}$ | n/a | n/a |
| ZaidNet GradVis | $109.7 \pm 0.6$ | n/a | $\boxed{8.7 \pm 0.7}$ | $125.5 \pm 0.2$ | n/a | n/a |
| ZaidNet Input $*$ Grad | $109.9 \pm 0.3$ | n/a | $\boxed{8.7 \pm 0.7}$ | $125.5 \pm 0.2$ | n/a | n/a |
| ZaidNet Saliency | $109.7 \pm 0.6$ | n/a | $\boxed{8.7 \pm 0.7}$ | $125.5 \pm 0.2$ | n/a | n/a |
| ALL (ours) | $\boxed{107.3 \pm 0.4}$ | $104 \pm 1$ | $9.9 \pm 0.7$ | $125.5 \pm 0.3$ | $\underline{1.8 \pm 0.1}$ | $1.014 \pm 0.001$ |

Table 13: Performance comparison between various leakage localization algorithms according to the reverse DNN occlusion test(**larger is better**) described in Sec. E.3.4. Results are reported as mean ± standard deviation over 5 random seeds. The best result is $\boxed{\text{boxed}}$ and the best deep learning result is underlined. We consider a result to be 'best' if its mean lies inside of the error bars of the result with the highest mean.

| Method | 2nd-order datasets | | 1st-order datasets | | | |
| --- | --- | --- | --- | --- | --- | --- |
| | ASCADv1 (fixed) | ASCADv1 (random) | DPAv4 (Zaid vsn.) | AES-HD | OTiAiT | OTP (1024-bit) |
| Random | $111.9 \pm 0.5$ | $114 \pm 3$ | $25 \pm 4$ | $126.7 \pm 0.2$ | $1.276 \pm 0.010$ | $1.079 \pm 0.010$ |
| SNR | $110.679$ | $123.284$ | $\boxed{125.924}$ | $\boxed{128.599}$ | $5.267$ | $\boxed{1.373}$ |
| SOSD | $112.074$ | $126.765$ | $108.267$ | $128.216$ | $4.957$ | $\boxed{1.369}$ |
| CPA | $119.875$ | $117.885$ | $114.857$ | $128.367$ | $3.792$ | $\boxed{1.364}$ |
| GradVis | $125.9 \pm 0.2$ | $127.6 \pm 0.1$ | $122 \pm 1$ | $128.0 \pm 0.3$ | $4.2 \pm 0.4$ | $1.34 \pm 0.07$ |
| Saliency | $125.8 \pm 0.2$ | $127.4 \pm 0.2$ | $122 \pm 1$ | $128.0 \pm 0.3$ | $5.1 \pm 0.3$ | $1.33 \pm 0.06$ |
| Input $*$ Grad | $125.7 \pm 0.3$ | $127.5 \pm 0.2$ | $121.9 \pm 0.9$ | $128.1 \pm 0.3$ | $5.2 \pm 0.3$ | $1.34 \pm 0.06$ |
| LRP | $125.7 \pm 0.3$ | $127.5 \pm 0.2$ | $121.9 \pm 0.9$ | $128.1 \pm 0.3$ | $5.2 \pm 0.3$ | $1.34 \pm 0.06$ |
| OccPOI | $122.3 \pm 0.4$ | $124.6 \pm 0.2$ | $43 \pm 1$ | $127.0 \pm 0.3$ | $3.6 \pm 0.3$ | $1.09 \pm 0.04$ |
| 1-Occlusion | $125.8 \pm 0.4$ | $127.4 \pm 0.2$ | $122 \pm 1$ | $128.1 \pm 0.3$ | $5.2 \pm 0.3$ | $1.34 \pm 0.06$ |
| $m^*$-Occlusion | $126.0 \pm 0.2$ | $127.4 \pm 0.2$ | $121 \pm 1$ | $\underline{128.5 \pm 0.2}$ | $5.3 \pm 0.2$ | $1.30 \pm 0.04$ |
| 1-Occlusion$^2$ | $125.8 \pm 0.3$ | $127.5 \pm 0.2$ | $122.0 \pm 0.9$ | $128.1 \pm 0.3$ | $5.3 \pm 0.2$ | $1.34 \pm 0.06$ |
| $m^*$-Occlusion$^2$ | $126.1 \pm 0.2$ | $127.4 \pm 0.2$ | $121.3 \pm 0.9$ | $\underline{128.5 \pm 0.2}$ | $5.3 \pm 0.2$ | $1.30 \pm 0.04$ |
| WoutersNet 1-Occlusion | $121.9 \pm 0.7$ | n/a | $116 \pm 5$ | $128.2 \pm 0.1$ | n/a | n/a |
| WoutersNet $m^*$-Occlusion | $122.4 \pm 0.8$ | n/a | $119.3 \pm 0.7$ | $\underline{128.4 \pm 0.2}$ | n/a | n/a |
| WoutersNet GradVis | $121.8 \pm 0.7$ | n/a | $116 \pm 5$ | $128.2 \pm 0.1$ | n/a | n/a |
| WoutersNet Input $*$ Grad | $121.9 \pm 0.7$ | n/a | $116 \pm 5$ | $128.2 \pm 0.1$ | n/a | n/a |
| WoutersNet Saliency | $121.8 \pm 0.7$ | n/a | $116 \pm 4$ | $128.2 \pm 0.1$ | n/a | n/a |
| ZaidNet 1-Occlusion | $122 \pm 3$ | n/a | $116 \pm 3$ | $128.1 \pm 0.2$ | n/a | n/a |
| ZaidNet $m^*$-Occlusion | $123 \pm 3$ | n/a | $119 \pm 2$ | $\underline{128.4 \pm 0.3}$ | n/a | n/a |
| ZaidNet GradVis | $122 \pm 3$ | n/a | $117 \pm 3$ | $128.1 \pm 0.2$ | n/a | n/a |
| ZaidNet Input $*$ Grad | $122 \pm 3$ | n/a | $116 \pm 3$ | $128.1 \pm 0.2$ | n/a | n/a |
| ZaidNet Saliency | $122 \pm 3$ | n/a | $117 \pm 3$ | $128.1 \pm 0.2$ | n/a | n/a |
| ALL (ours) | $\boxed{126.4 \pm 0.2}$ | $\boxed{127.96 \pm 0.06}$ | $\underline{125 \pm 1}$ | $128.3 \pm 0.2$ | $\boxed{5.6 \pm 0.2}$ | $\boxed{1.39 \pm 0.05}$ |

Table 14: Performance comparison between various leakage localization algorithms according to the template attack feature selection test (**smaller is better**) described in Sec. E.3.4. Results are reported as mean $\pm$ standard deviation over 5 random seeds. The best result is boxed and the best deep learning result is underlined. We consider a result to be 'best' if its mean lies inside of the error bars of the result with the highest mean.

| Method | 2nd-order datasets | | 1st-order datasets | | | |
| | ASCADv1 (fixed) | ASCADv1 (random) | DPAv4 (Zaid vsn.) | AES-HD | OTiAiT | OTP (1024-bit) |
|---|---|---|---|---|---|---|
| Random | $1293 \pm 1000$ | $56421 \pm 40000$ | $449 \pm 100$ | $25000.000$ | $1.2 \pm 0.1$ | $1.44 \pm 0.02$ |
| SNR | $5496.010$ | $54834.990$ | $2.590$ | $17159.690$ | $\boxed{1.058}$ | $1.385$ |
| SOSD | $7733.870$ | $3237.250$ | $91.410$ | $17520.530$ | $1.059$ | $1.398$ |
| CPA | $3826.440$ | $100000.000$ | $10.540$ | $18974.510$ | $1.101$ | $1.385$ |
| GradVis | $686 \pm 100$ | $1162 \pm 1000$ | $2.7 \pm 0.1$ | $20014 \pm 6000$ | $1.4 \pm 0.3$ | $1.378 \pm 0.007$ |
| Saliency | $726 \pm 100$ | $1412 \pm 2000$ | $2.7 \pm 0.1$ | $19438 \pm 6000$ | $1.14 \pm 0.02$ | $1.379 \pm 0.005$ |
| Input $*$ Grad | $675 \pm 100$ | $1194 \pm 2000$ | $2.6 \pm 0.1$ | $19893 \pm 6000$ | $1.14 \pm 0.02$ | $1.378 \pm 0.003$ |
| LRP | $675 \pm 100$ | $1194 \pm 2000$ | $2.6 \pm 0.1$ | $19893 \pm 6000$ | $1.14 \pm 0.02$ | $1.378 \pm 0.003$ |
| OccPOI | $787 \pm 100$ | $942 \pm 200$ | $71 \pm 30$ | $25000.000$ | $\underline{1.08 \pm 0.03}$ | $1.47 \pm 0.05$ |
| 1-Occlusion | $667 \pm 100$ | $1376 \pm 2000$ | $2.65 \pm 0.08$ | $20011 \pm 6000$ | $1.14 \pm 0.02$ | $1.379 \pm 0.003$ |
| $m^*$-Occlusion | $673 \pm 70$ | $727 \pm 400$ | $9 \pm 1$ | $\boxed{16283 \pm 10}$ | $1.17 \pm 0.02$ | $1.382 \pm 0.007$ |
| 1-Occlusion$^2$ | $709 \pm 100$ | $1086 \pm 1000$ | $2.65 \pm 0.08$ | $20222 \pm 6000$ | $1.14 \pm 0.02$ | $1.378 \pm 0.003$ |
| $m^*$-Occlusion$^2$ | $642 \pm 60$ | $710 \pm 400$ | $9 \pm 1$ | $\boxed{16033 \pm 700}$ | $1.16 \pm 0.03$ | $1.381 \pm 0.008$ |
| WoutersNet 1-Occlusion | $6454 \pm 4000$ | n/a | $2.9 \pm 0.6$ | $20278 \pm 3000$ | n/a | n/a |
| WoutersNet $m^*$-Occlusion | $4408 \pm 4000$ | n/a | $11.7 \pm 0.6$ | $\boxed{16124 \pm 300}$ | n/a | n/a |
| WoutersNet GradVis | $6230 \pm 4000$ | n/a | $2.9 \pm 0.5$ | $19539 \pm 5000$ | n/a | n/a |
| WoutersNet Input $*$ Grad | $4988 \pm 4000$ | n/a | $3.0 \pm 0.6$ | $20546 \pm 4000$ | n/a | n/a |
| WoutersNet Saliency | $5878 \pm 4000$ | n/a | $2.8 \pm 0.5$ | $20151 \pm 4000$ | n/a | n/a |
| ZaidNet 1-Occlusion | $2236 \pm 2000$ | n/a | $2.6 \pm 0.3$ | $20696 \pm 2000$ | n/a | n/a |
| ZaidNet $m^*$-Occlusion | $2485 \pm 2000$ | n/a | $9 \pm 2$ | $\boxed{16124 \pm 300}$ | n/a | n/a |
| ZaidNet GradVis | $2560 \pm 3000$ | n/a | $3.0 \pm 0.8$ | $22790 \pm 3000$ | n/a | n/a |
| ZaidNet Input $*$ Grad | $3234 \pm 3000$ | n/a | $2.5 \pm 0.3$ | $21600 \pm 3000$ | n/a | n/a |
| ZaidNet Saliency | $2295 \pm 2000$ | n/a | $2.4 \pm 0.4$ | $22561 \pm 3000$ | n/a | n/a |
| ALL (ours) | $\boxed{459 \pm 40}$ | $\boxed{394 \pm 20}$ | $\boxed{2.22 \pm 0.01}$ | $17582 \pm 5000$ | $1.11 \pm 0.02$ | $\boxed{1.363 \pm 0.007}$ |

**Distribution of performance during hyperparameter sweeps**   In Fig. 20 we show the distribution of performance during the random hyperparameter searches for ALL and selected baselines. We see that ALL convincingly outperforms all baselines besides $m^*$-Occlusion on all datasets. Additionally, it has a higher peak performance than $m^*$-Occlusion on every dataset except for DPAv4, and a higher median performance on ASCADv1-fixed, ASCADv1-variable, and AES-HD. As previously noted and considered again in subsequent ablation studies, we can further improve the performance of ALL relative to $m^*$-Occlusion by mimicking its smoothing effect with stride-1 average-pooling.

### E.3.7  Ablation studies

ALL has many differences from prior work, so here we run ablation studies to evaluate the impact of some of the important individual differences. For each ablated design decision, we run a new 50-trial hyperparameter sweep and plot the distribution of performance in terms of oracle agreement. Results are shown in Fig. 21, with the salient results from Fig. 20 copied over for reference. We ablate the following design decisions:

**Heavy input dropout for the supervised classifiers used by baselines**   See the distributions labeled '$m^*$-Occl + heavy dropout'. The ALL classifier is trained on occluded inputs with the possible 'heaviness' of the occlusion chosen spanning a wide range and chosen as a hyperparameter to optimize our proxy for oracle agreement. In contrast, the baseline methods are based on 'interpreting' fixed classifiers which have been trained with input dropout chosen from $\{0.0, 0.1\}$ to optimize classification performance. A plausible conjecture is that ALL has strong performance because the heavy input corruption 'encourages' the classifier to compensate by leveraging a wider variety of input-output associations, whereas because the supervised classifiers train on uncorrupted or lightly-corrupted inputs, they have no such 'incentive'. We test this assumption by tuning the classifiers with input dropout chosen from $\{0.05, 0.1, \dots, 0.95\}$ (the same search space that ALL uses for $\overline{\gamma}$). We plot the performance distribution for $m^*$-occlusion and 1-occlusion. For clarity we omit GradVis, Saliency, LRP and Input $*$ Grad, but these have similar trends as 1-occlusion. We did not test OccPOI or second-order occlusion because they are costly and in our prior experiments second-order $m$-occlusion performs similarly to $m$-occlusion and OccPOI performs poorly compared to baselines.

We see that this heavy input dropout leads to significant improvements to the maximum and median performance on all datasets, though ALL still convincingly outperforms 1-occlusion and $m$-occlusion except on DPAv4 and OTiAiT. To our knowledge no prior work has explored the effect of regularization strategies on leakage localization performance of methods which 'interpret' supervised classifiers, and this result suggests that such research may be fruitful.

**Adversarial $\rightarrow$ 'Cooperative' leakage localization**   See the distributions labeled 'ALL (cooperative)'. The intuition behind ALL is that we train a noise distribution to distribute a fixed amount of noise to minimize the performance of a classifier, and we then interpret noisier measurements as leakier. Along the lines of this intuition, we could also train it to distribute noise to *maximize* the performance of a classifier, then interpret less-noisy measurements as leakier. We try this approach and find that it typically degrades performance relative to the adversarial version of the algorithm. We conjecture that this is because the adversarial approach encourages the classifier to rely on a diverse assortment of input-output associations, whereas the cooperative approach does the opposite.

**Omitting the noise conditioning to the classifier**   See the distributions labeled 'ALL (unconditional)'. We feed the occlusion mask as an auxiliary input to the ALL classifiers, motivated by our theory which views it as a family of classifiers trained in an amortized manner via conditioning a single neural net. We test omitting this auxiliary input and find mixed results. On ASCADv1-fixed and DPAv4, ALL achieves stronger performance without this auxiliary input, whereas on AES-HD it achieves stronger performance with this input and on ASCADv1-variable, OTiAiT, and OTP results are approximately the same. For future work it is likely justifiable to simplify ALL by omitting this conditioning.

**Average pooling ALL to mimic the smoothing effect of $m^*$-Occlusion**   See the distributions labeled 'ALL + AvgPool($m^*$)'. Recall that we have chosen the occlusion window size $m^*$ by sweeping it over successive odd numbers until finding a local maximum in oracle agreement. We find that this consistently improves

performance. One plausible explanation for the performance improvement is that large-window occlusion has a 'smoothing' effect which amounts to an assumption that temporally-close measurements have similar leakiness. This is likely true for some datasets, and as 1-occlusion is always included in the window size search space, $m^*$ occlusion will never degrade performance relative to 1-occlusion. To test this explanation we average pool ALL with stride 1, kernel size $m^*$, and zero-padding to preserve dimensionality, which creates a similar smoothing effect. Our results seem to support this explanation, with average pooling significantly improving the performance of ALL on the DPAv4 and AES-HD datasets, where $m^*$-occlusion also has significantly stronger performance than 1-occlusion. Note that using oracle agreement to choose an occlusion/pooling window size causes data contamination, so while it is fair to compare results for $m^*$-average-pooled ALL to those of $m^*$-occlusion, they cannot fairly be compared to any other baseline.

### E.3.8 Theoretical and empirical computational cost of deep learning methods

In table 15 we list the theoretical computational complexity of the considered methods, as well as the measured wall clock time to run them. While for fairness our experiments use an equal number of training steps for ALL and supervised learning, note that in general we suspect the latter will converge in fewer training steps; thus, these measurements likely overestimate the practical wall-clock time of supervised training. Additionally, note that OccPOI takes significantly more time than $m$-occlusion despite having the same computational complexity; this is because it requires $\Omega(T)$ *sequential* forward passes through the model, whereas all forward passes may be done in parallel for $m$-occlusion. There is additional variance for the runtime of this method because we choose the attack dataset size to be as small as possible while comfortably allowing the classifier to attain a correct-key rank of $0$ – e.g. AES-HD takes the longest because we use the full attack dataset. We omit the parametric statistics-based methods from consideration, but these are done on the CPU and take a negligible amount of time compared to the deep learning methods.

Table 15: Comparison of the computational cost of methods considered in our work. We denote by $C_F$ and $C_B$ the cost of a forward and backward pass through our neural net respectively, $N$ the dataset size, $n_{\text{sup}}$ and $n_{\text{all}}$ the number of epochs required for supervised learning and ALL respectively, and $T$ the data dimensionality. We omit the parametric statistics-based baseline methods because they are done on the CPU and take a negligible amount of time relative to the deep learning methods. Runtimes are reported as mean $\pm$ standard deviation over 5 runs. [†]These methods are used to 'interpret' a trained supervised classifier, but for comparison we report the cost of running them *after* training the classifier. In practice one would also incur the cost of supervised training (first row).

| Method | Total FLOPS | A6000 minutes per trial | | | | | |
| --- | --- | --- | --- | --- | --- | --- | --- |
| | | ASCADv1 (fixed) | ASCADv1 (random) | DPAv4 (Zaid) | AES-HD | OTiAiT | OTP |
| Supervised training | $\Theta(Nn_{\text{sup}}(C_F + C_B))$ | $2.09 \pm 0.01$ | $3.68 \pm 0.02$ | $1.98 \pm 0.04$ | $1.78 \pm 0.03$ | $0.17 \pm 0.02$ | $0.448 \pm 0.009$ |
| GradVis[†] | $\Theta(N(C_F + C_B))$ | $0.0675 \pm 0.0008$ | $0.263 \pm 0.002$ | $0.0080 \pm 0.0001$ | $0.0384 \pm 0.0003$ | $0.0080 \pm 0.0001$ | $0.0666 \pm 0.0005$ |
| Saliency[†] | $\Theta(N(C_F + C_B))$ | $0.078 \pm 0.001$ | $0.300 \pm 0.003$ | $0.0086 \pm 0.0002$ | $0.048 \pm 0.002$ | $0.0085 \pm 0.0002$ | $0.0881 \pm 0.0004$ |
| Input $*$ Grad[†] | $\Theta(N(C_F + C_B))$ | $0.080 \pm 0.002$ | $0.3023 \pm 0.0007$ | $0.0086 \pm 0.0002$ | $0.049 \pm 0.002$ | $0.0086 \pm 0.0002$ | $0.086 \pm 0.003$ |
| LRP[†] | $\Theta(N(C_F + C_B))$ | $0.080 \pm 0.001$ | $0.3045 \pm 0.0008$ | $0.0088 \pm 0.0002$ | $0.050 \pm 0.002$ | $0.00861 \pm 0.00007$ | $0.087 \pm 0.005$ |
| $m$-Occlusion[†] | $\Theta(NC_FT)$ | $0.1235 \pm 0.0004$ | $0.952 \pm 0.003$ | $0.0644 \pm 0.0003$ | $0.1781 \pm 0.0007$ | $0.018 \pm 0.002$ | $0.247 \pm 0.002$ |
| $2^{\text{nd}}$-order $m$-Occlusion[†] | $\Theta(NC_FT^2)$ | $16.6 \pm 0.1$ | $327 \pm 1$ | $81.1 \pm 0.5$ | $68.4 \pm 0.6$ | $4.42 \pm 0.02$ | $70.6 \pm 0.3$ |
| OccPOI[†] | $\Omega(NC_FT)$ | $2.29 \pm 0.09$ | $7.6 \pm 0.4$ | $0.709 \pm 0.009$ | $36.5 \pm 0.6$ | $0.042 \pm 0.002$ | $0.0437 \pm 0.0004$ |
| ALL (Ours) | $\Theta(Nn_{\text{all}}(C_F + C_B))$ | $3.2 \pm 0.2$ | $4.7 \pm 0.2$ | $2.44 \pm 0.05$ | $2.28 \pm 0.05$ | $0.200 \pm 0.005$ | $0.323 \pm 0.004$ |

## F  Limitations

For real side-channel leakage datasets we lack ground truth knowledge about the leakiness of each measurement, and all evaluation metrics considered in our paper have limitations. In particular:

- The 'oracle' leakage assessments used in the main paper ignore leakage of order greater than 1 except where higher-order leakage may be decomposed into first-order leakage of multiple variables, similarly to the analysis of Egger et al. (2022). This relies on careful analysis of implementations by humans, and is subject to error and oversights. Additionally, the SNR is not guaranteed to detect even first-order leakage – it is sensitive only to the influence of the secret variable $Y$ on the *mean* of each

$X_t \mid Y$, and will not detect cases where the distribution of $X_t \mid Y$ changes with the mean remaining fixed (e.g. if $X_t$ is Gaussian distributed with $Y$-dependent variance and $Y$-independent mean).

- The DNN occlusion tests and similar metrics proposed by Hettwer et al. (2020) are sensitive only to the associations between $\boldsymbol{X}$ and $Y$ that the neural net exploits. The superior performance of ALL compared to prior deep learning-based algorithms, as well as work in other domains such as Geirhos et al. (2020); Hermann & Lampinen (2020), suggests that DNNs are prone to exploiting some but not all of the associations at their disposal. Additionally, such metrics may be biased in favor of deep learning methods due to both leveraging SGD-trained DNNs.

- Evaluation via feature selection for a Gaussian template attack as done by Masure et al. (2019); Yap et al. (2023) is sensitive only to the top $\approx 10$ leakiest measurements identified by an algorithm and ignores all others, which is a major limitation. Additionally, for second-order datasets a template attack will be unsuccessful unless *both* of a leaking pair of variables (e.g. $r_3$ and $\mathrm{Sbox}(w_3 \oplus k_3) \oplus r_3$ for ASCADv1) are leaked through the selected measurements, and because no method considered in our work except for second-order occlusion has any ability to discern the particular variable leaking through a measurement, this would rely on random chance.

We believe it is important for work in this domain to use a variety of evaluation strategies to compensate for these individual limitations. This is similar to image synthesis research where it is common to use precision, recall, and FID score to avoid the individual limitations of each of these metrics.

ALL (alongside the other deep learning-based method we consider) has major advantages over manual analysis and simpler parametric methods. However, it also comes with its own limitations:

- Deep learning methods such as ours have a far larger computational cost than parametric methods. Additionally, ALL is more computationally-expensive than the prior deep learning algorithms apart from OccPOI and second-order occlusion. This increased cost is not fully reflected in our reported runtimes because while for fairness we have run both ALL and supervised learning-based methods for the same number of training steps and post-hoc early-stopped the latter, in practice we find that supervised learning usually converges to good solutions in fewer training steps than ALL.

- Deep learning methods such as ours require hyperparameter tuning and will give incorrect leakiness estimates if poorly tuned. We view ALL as *complementary* with traditional approaches rather than as a replacement. Traditional approaches can cheaply exploit domain/implementation knowledge and detect many types of leakage for a low computational cost. ALL can then be used to search for additionally leaking measurements not detected by these approaches.

Additionally, a limitation of the literature on deep side-channel leakage localization which we do not address in this work is that experiments are done at a smaller scale than the state of the art in deep side-channel attacks, and the considered algorithms would likely need to be significantly scaled up for practical use. The largest-scale experiments in this and prior work consider the ASCADv1-variable dataset, which consists of 300k 1400-measurement power traces and can be attacked with simple MLP and CNN architectures. The 1400-length power traces are extracted from longer 250k-length traces via downsampling and cropping in the general vicinity of known leaky instructions which were themselves located with a white box analysis. We believe that a critical direction for future work in this area is scaling existing methods to larger-scale architectures applied to uncropped high-dimensionality datasets, such as the transformer architecture of Bursztein et al. (2023) and the raw ASCADv2 dataset (Masure & Strullu, 2023), which consists of 800k 1M-measurement power traces.

Our experiments consider mainly temporally-synchronized power trace datasets, which we believe is reasonable because in practice implementation designers are likely able to collect synchronized traces, as the dataset authors have done (e.g. by using the clock line of the hardware to trigger an oscilloscope). In practice a violation of this condition would mean that leakage appears 'spread out' (e.g. see row 3 of Fig. 4), making it more-challenging for designers to identify its source.

A limitation of ALL, alongside the parametric statistical methods and OccPOI, is that they produce only a single vector summarizing leakage over the entire dataset, rather than for individual traces. As observed by Wouters et al. (2020), the neural net attribution methods can also be used to assign leakiness estimates to individual traces, though to our knowledge no work has systematically studied this ability. This would be a useful direction for future work, and would likely require innovation in performance evaluation strategies beyond those introduced in this paper or prior work.

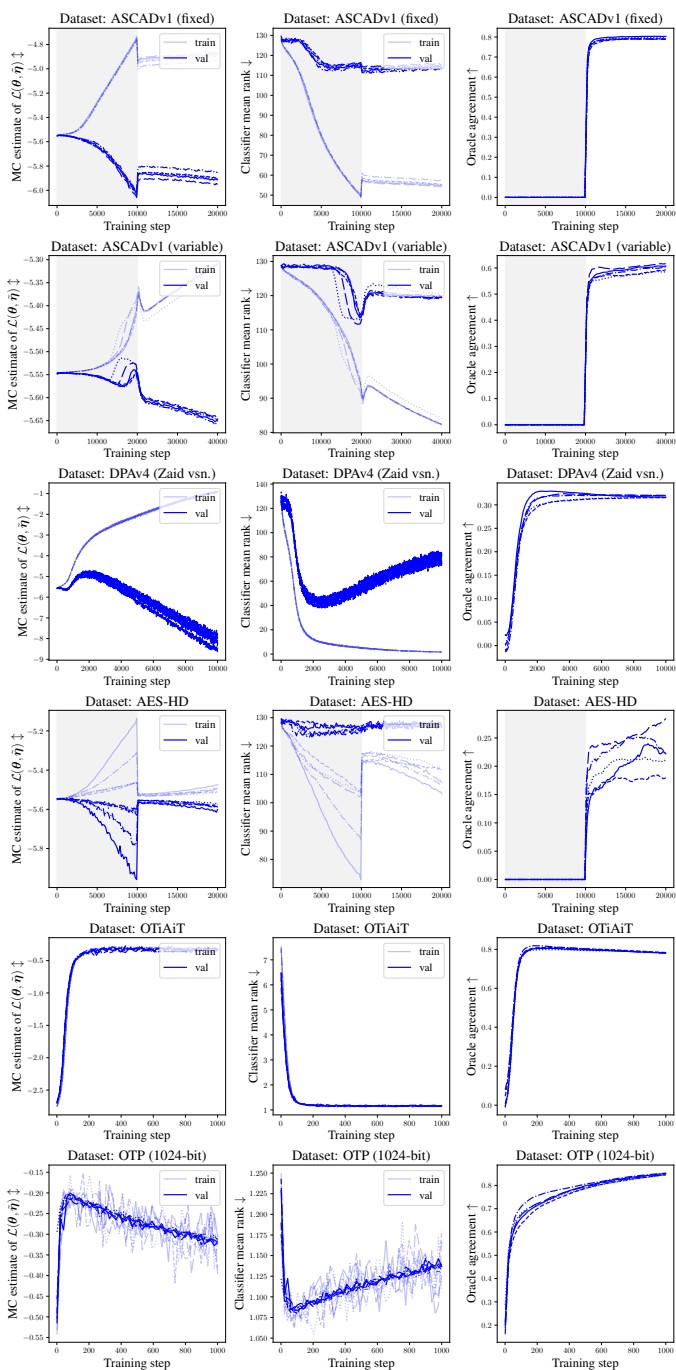

Figure 15: Training curves for ALL with the hyperparameter configuration chosen using our composite model selection criterion. Grey shaded regions denote 'pretraining' phases where we optimize $\boldsymbol{\theta}$ but not $\tilde{\boldsymbol{\eta}}$. Note that jumps in the traces happen because we set $\overline{\gamma} = 0.5$ for pretraining and change it to the setting chosen through hyperparameter search for training. (**left column**) The per-minibatch estimates of our objective function $\mathcal{L}(\boldsymbol{\theta}, \tilde{\boldsymbol{\eta}})$ (i.e. negative cross-entropy classification loss of the classifier) during training. We are optimizing $\boldsymbol{\theta}$ to maximize this value and $\tilde{\boldsymbol{\eta}}$ to minimize it. (**center column**) The mean rank of the correct label in the logits of the classifier (lower corresponds to higher classifier performance). We use this instead of accuracy for its finer granularity. (**right column**) The performance in terms of oracle agreement over the course of training; higher is better, as it indicates that the current ALL leakiness estimates are closer to being a strictly-increasing function of the oracle assessments.

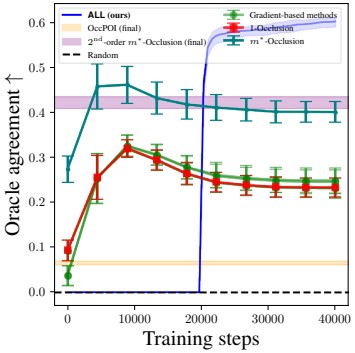

Figure 16: A comparison of the evolution of oracle agreement vs. training steps for ALL and selected baselines. Note that the oracle agreement for ALL is flat for the first 20k training steps and jumps up for the remaining steps because we leave all elements of $\gamma$ fixed at 0.5 during our 'pretraining' phase of training, and update it only during the second half. For $2^{\text{nd}}$-order $m^*$-Occlusion and OccPOI, due to their high computational cost we report only the final performance after training via horizontal lines, rather than the evolution of performance during training.

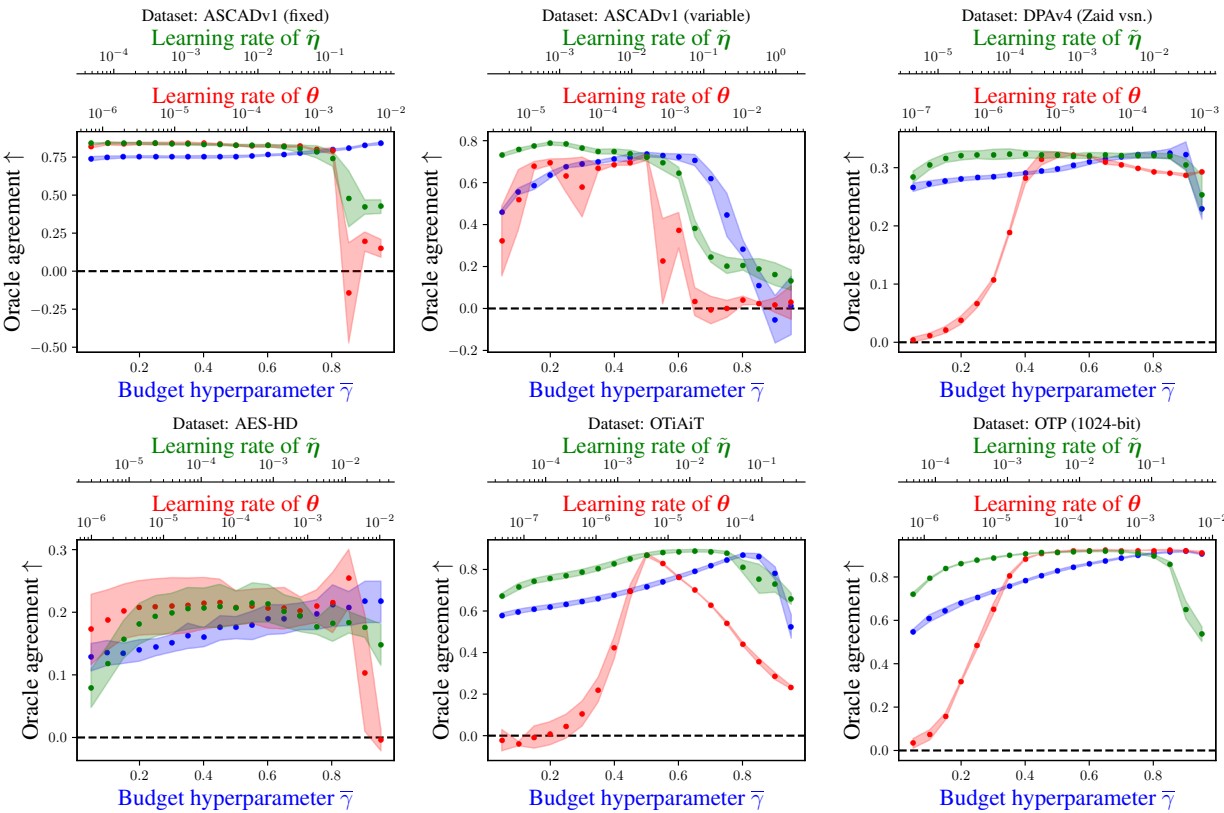

Figure 17: A plot of the performance of ALL as we perturb its 3 main hyperparameters: the noise budget $\overline{\gamma}$ and the learning rates of the classifier weights $\theta$ and the noise distribution parameter $\tilde{\eta}$. We find that performance varies smoothly with these hyperparameters and stays significantly better than random guessing over a large region of the search space, which is useful from a standpoint of hyperparameter tuning. We consider $\overline{\gamma}$ values in `np.arange(0.05, 1.0, 0.05)` and learning rates scaled by values in `np.logspace(-2, 2, 19)` relative to their optimal values according to the oracle agreement metric. All hyperparameters other than those being perturbed are left at their optimal values according to the oracle agreement metric. We repeat trials for 5 random seeds and report their mean with dots and $\pm 1$ standard deviation with shading.

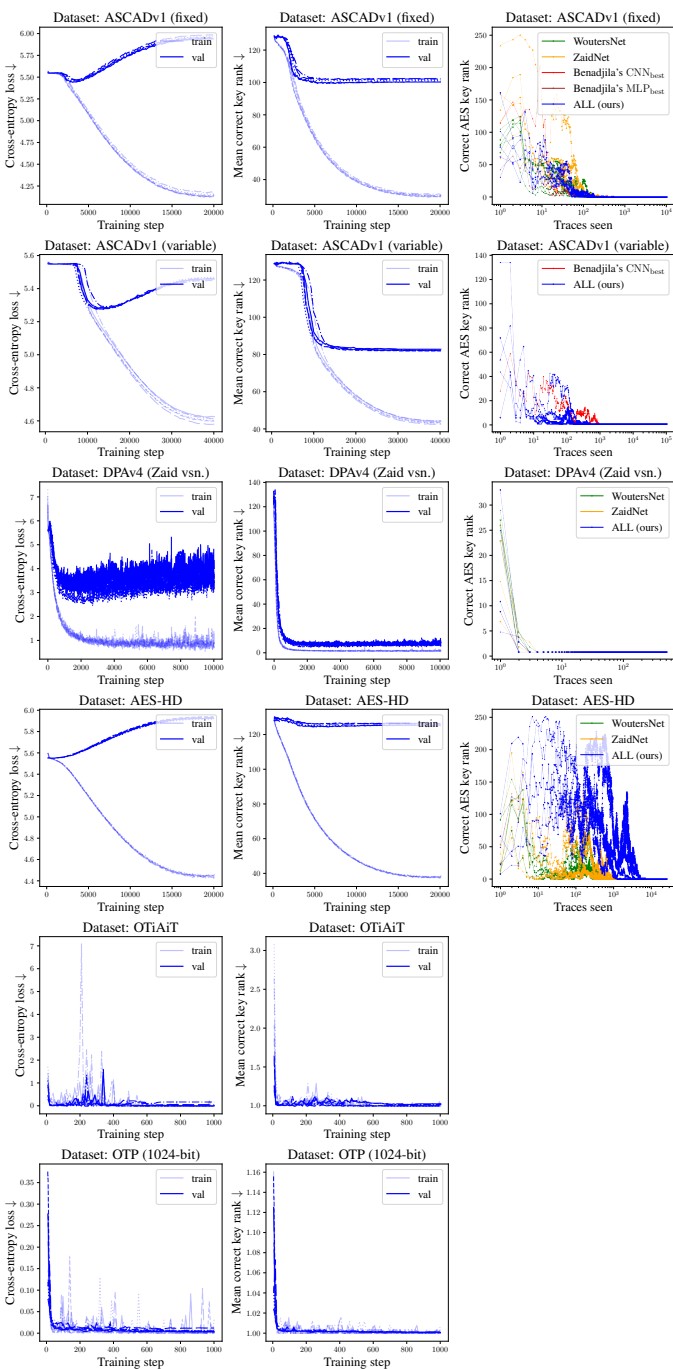

Figure 18: Training curves and attack performance of the supervised classifiers which are 'interpreted' by the deep learning baseline methods. (**Left column**) The training + validation cross-entropy loss vs. training steps for the supervised classifiers. (**center column**) The training + validation rank vs. training steps for the supervised classifiers. (**right column**) The rank of the correct key as we accumulate predictions on the attack dataset for the supervised classifiers, which is a common way of evaluating attack performance in the side-channel literature. For reference, we superimpose the results using open-weight classifiers provided by Benadjila et al. (2020) and Wouters et al. (2020). Note that our classifiers can successfully attack all datasets. They get comparable performance to the open-weight classifiers on ASCADv1-fixed, ASCADv1-variable and DPAv4, and somewhat worse performance on AES-HD.

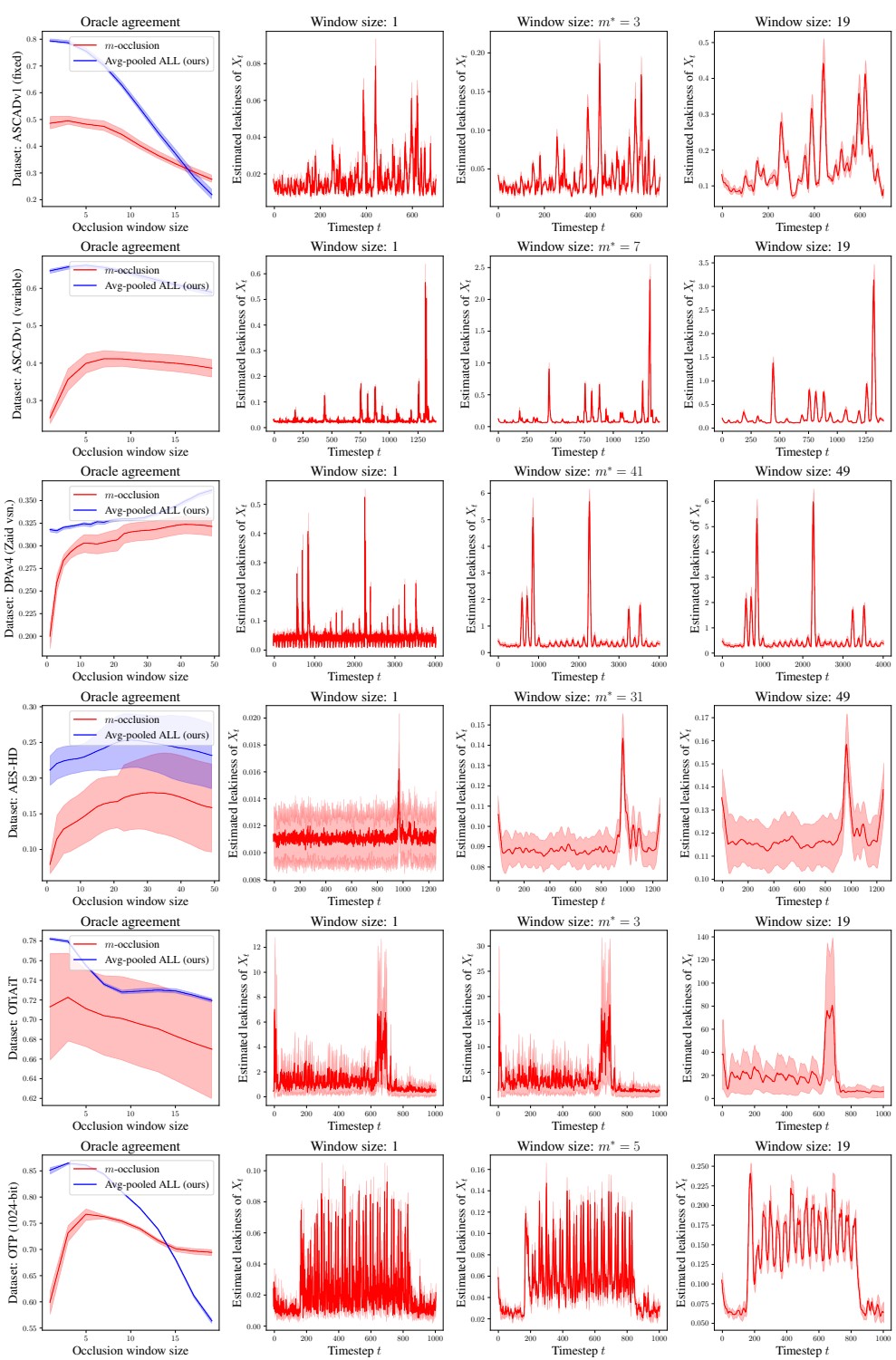

Figure 19: A sweep of the window size $m$ for $m$-Occlusion. In the **first column** we plot the oracle agreement vs. window size (red curve). For reference we also plot the output of ALL (blue curve) as we average-pool it with stride 1 and kernel size $m$. Note that ALL consistently outperforms $m$-Occlusion for a wide range of window sizes. In the **second, third and fourth columns** we plot the $m$-Occlusion leakiness assessment for $m = 1$, optimal $m$, and maximum considered $m$. Observe that there is a smoothing effect as we increase $m$.

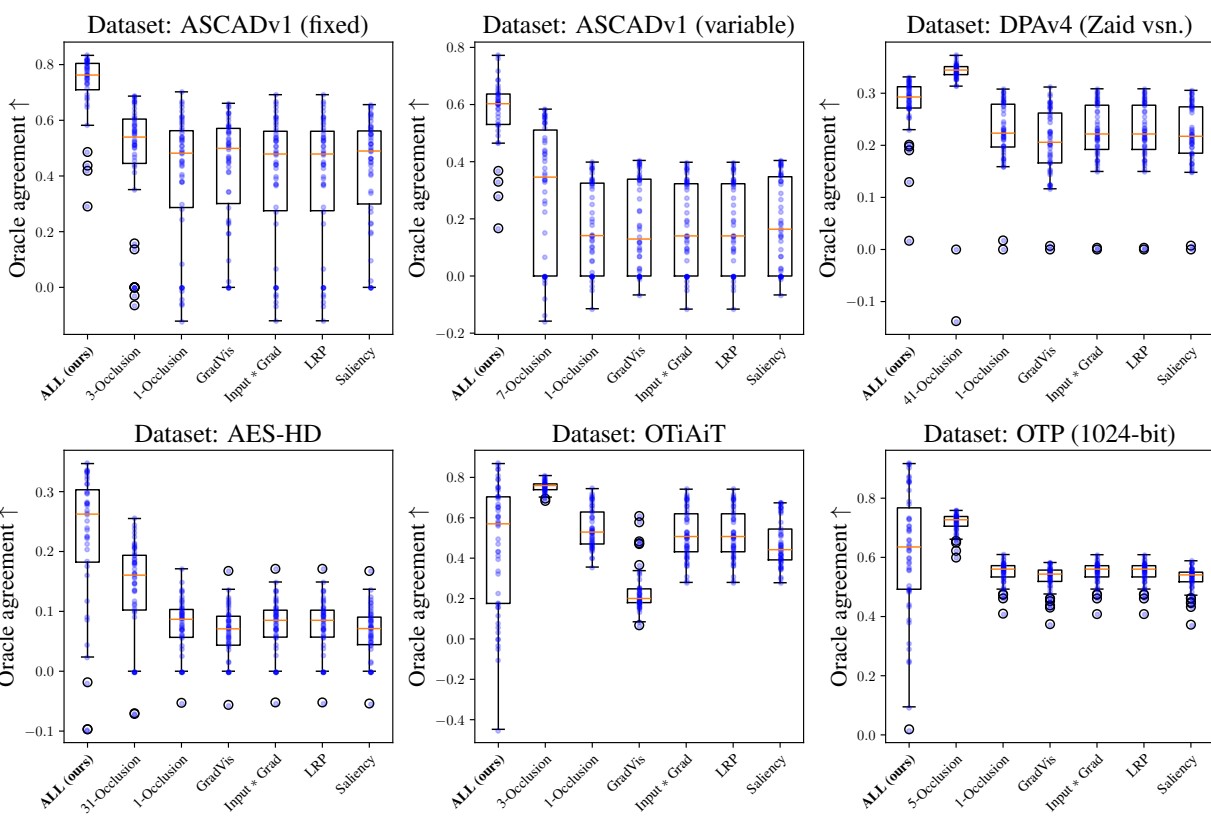

Figure 20: Distribution of performance of ALL and selected baseline methods during a 50-trial random hyperparameter search. Note that ALL generally outperforms baselines over a wide range of configurations. Blue dots denote individual samples, and boxes extend from first quartile to third quartile with a line at the median and whiskers extending to the furthest dot lying within 1.5× the interquartile range from the box.

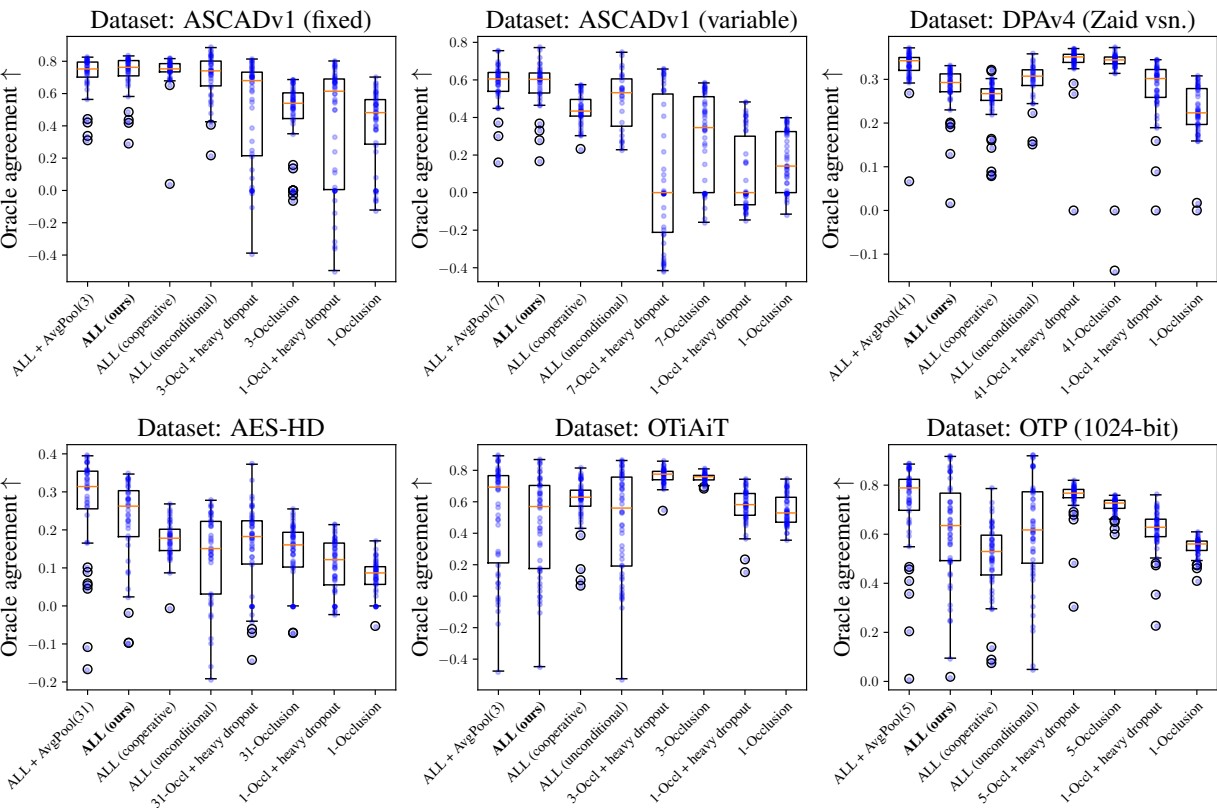

Figure 21: Ablation studies where we evaluate the influence of individual aspects of ALL on its performance gains relative to prior work, as described in Sec. E.3.7. These are plots of the distribution of oracle agreement values after a 50-trial random hyperparameter search with each ablation in place. 'ALL (ours)' denotes our method without modification. 'ALL + AvgPool($m^*$)' denotes an average-pooled version of ALL to mimic the smoothing effect of $m^*$-Occlusion. 'ALL (cooperative)' denotes ALL with the adversarial objective replaced by a 'cooperative' objective where both the classifier and noise distribution are trained to maximize the performance of the classifier. 'ALL (unconditional)' denotes ALL without the occlusion masks being fed as an auxiliary input to the classifier. '$m$-Occl + heavy dropout' denotes $m$-Occlusion with the input dropout to the classifier chosen from $\{0.05, 0.1, \ldots, 0.95\}$ rather than $\{0.0, 0.1\}$ as in our other experiments. '$m$-Occlusion' denote the results with the unmodified $m$-Occlusion techniques.

