# OpenReview forum: "Learning to Localize Leakage of Cryptographic Sensitive Variables"
_TMLR — Accepted by TMLR_

### Review · Reviewer_g2z4 · 2025-11-17

**Summary Of Contributions:**

1. The authors have proposed a deep-learning-based novel algorithm for localizing side-channel leakage from cryptographic implementations, while also introducing a principled information-theoretic quantity for measuring the ‘leakiness’.

2. Wide experimental comparisons are conducted, including multiple baselines and benchmark datasets.

3. The code is provided.

4. Extend versions of background/related work/method introductions, etc, are provided in the appendix.

**Additional Comments:**

I have to admit that I do not have strong expertise in this specific subfield. As such, my review focuses on general clarity, methodological soundness, and logical consistency rather than deep domain-specific technical details

**Audience:**

Yes

**Audience Explanation:**

This work aims to develop a deep-learning-based method to localize leakage of cryptographic sensitive variables, which is a valued practical application of machine learning algorithms.

**Claims And Evidence:**

Yes

**Claims Explanation:**

The wide empirical evaluations on multiple datasets and across various baselines demonstrate the improved performance of the proposed method. My question here is how could the authors ensure that the comaprisons are fair?

**Requested Changes:**

1. The paper uses a relatively complex system of symbols. The authors have tables of symbols in the appendix. Would it be better to move (part of) the tables into the main body and also include the dimensions of the symbols  (where applicable)?

2. The authors use Eq.(3) for re-parameterization invovling sigmoid function. As one of the properties of the sigmoid function is the startution effect, will it also influence the algorithm?

3. Do the authors have insights into the complexity of the proposed method (e.g., comparing with others or the scalability of larger and more complex problems)?

---

> ### Author Response · Authors · 2026-01-17
>
> Thank you for your positive review. We have updated our manuscript in response to your requested changes.
> ## How did we ensure that the comparisons are fair?
> - As discussed in our response to Reviewer ozmX, we have expanded our results section to clarify our comparison methodology and experimental setup. See lines 314--329 where we discuss the experimental setup, and lines 330--359 where we discuss the performance metrics. In particular:
> 	- Despite their methodological differences, all considered methods can be viewed as a function which takes a dataset as input and returns a per-timestep estimated leakiness value as output. Our performance metrics compare the fidelity of these estimated leakiness values and are agnostic to how they were generated.
> 	- Because the specific numbers used to encode leakiness by the different baselines are not directly comparable, we use metrics which are sensitive only to the *relative* leakiness assigned to different timesteps.
> 	- The baselines entail supervised training of a classifier, and ALL entails adversarially training a classifier $\Phi_\theta$ vs. a noise distribution. We use the same architecture for both classifiers.
> 	- We use the same training step count and minibatch size for supervised classifier training and for pretraining + adversarial training of ALL. Additionally, we use the same hyperparameter tuning budget to tune the important hyperparameters of both.
> ## Can we add a notation table to the main paper?
> - As requested, we have added a table near the beginning of our manuscript listing the important variables and their dimensionality. See Table 1.
> ## Does the saturation effect of the sigmoid function influence our algorithm?
> - The sigmoid function is used precisely because of its saturation effect: we want to use SGD to train a vector of probabilities. There is no guarantee that the SGD updates will keep these in the interval $[0, 1]$, so we use the sigmoid function to instead train logits which can take on any value in $\mathbb{R},$ then transform them to stay within the desired interval.
> - This technically means that our algorithm can no longer learn probabilities which are exactly 0 or 1. However, the probabilities can become arbitrarily close to these, and in practice we find they become close enough after a reasonable number of training steps.
> - Note that while saturating activation functions notoriously cause vanishing gradient problems when used repeatedly as the hidden activation of a deep or recurrent network, this is not an issue here because we are using it only once at the output of the noise generator.
> ## Do we have insight into the complexity/scaling behavior of the proposed method?
> - We list and discuss the computational complexity and wall clock runtime of our method and the deep learning baselines in Appendix E.3.8.

---

> > ### Comment · Reviewer_g2z4 · 2026-01-30
> >
> > My belated thanks to the authors for the additional clarification. ：）Thanks！

---

### Review · Reviewer_ozmX · 2025-11-26

**Summary Of Contributions:**

The authors present Adversarial Leakage Localization (ALL), a deep learning algorithm designed to address a self-defined constraint optimization problem that quantifies leakage, specifically in the context of side-channel attacks against AES, ECC, and RSA implementations using power and electromagnetic measurements. They evaluate ALL against 11 methods across two second-order datasets and four first-order datasets.

**Audience:**

Yes

**Audience Explanation:**

The authors present a novel deep learning-based methodology in the domain of side-channel leakage localization. This work aligns with TMLR's scope, as it contributes to the understanding of the computational principles (specifically, leakage localization) through learning (given their deep learning-based approach). As such, it would likely be of interest to at least some individuals in the audience.

**Broader Impact Concerns:**

I find the Broader Impact Statement to be generally accurate in its intent. However, I would argue that while the work is defensive in nature, algorithms that improve the identification of any system's weaknesses (such as leakage) do ultimately formally increase capabilities of attackers, regardless of the authors' intentions or the permissibility of the threat model (profiling). Nevertheless, I concur with the authors' assertion that "there are limited avenues to directly exploit this work for malicious purposes".

**Claims And Evidence:**

Yes

**Claims Explanation:**

To the extent of my knowledge, the paper does not appear to make *bold* statements that lack empirical or rigorous evidence. It is clearly written, does not, to my knowledge, claim novelty over existing published work, and does not appear to re-implement ideas that have already been reproduced. I do share some concerns about the methodology, results and comparisons made to previous work, which I have attached in the requested changes section.

**Requested Changes:**

- **[5.Results]:** In the main section of the paper, the authors employ a scalar representation of oracle agreement as the primary method for evaluating results. This approach could raise some concerns, considering that, as the authors acknowledge, there is no established consensus on how to evaluate leakage localization algorithms and that this particular evaluation method is introduced by the authors themselves. While the method is novel, since it also appears to favor their work, I recommend incorporating the data from other methodologies (for example, Tables 11, 12, and 13, Appendix, E.3.6) to provide a more comprehensive and balanced perspective.

- **[General]:** Please address consistency in naming conventions. "input * grad" is also referred to as "input * gradient" and "Input * grad". "Saliency" is also referred to as "saliency". "Second-order" is sometimes referred to as "2nd-order".

- **[5.Results]:** It seems that ASCADv1 is the only second-order dataset used. Could you please elaborate on the decision to use only this dataset?

- **[5.2, Figure 5, E.3.8]**: It would be helpful to provide additional context regarding the conditions under which comparisons between methods are conducted. It should be intuitive to the reader that these evaluations enable a fair comparison, despite differences in the underlying methodologies. As noted in section E.3.8, all methods except for ALL (and supervised training) are used to "interpret" a pre-trained supervised classifier, while ALL is not. Because of this, Figure 5 may initially seem unintuitive. Could you please expand on and clarify these points?

- **[5.2, Figure 5]:** I would recommend using a different color for 1-occlusion and m*-occlusion.

- **[4.3]:** Consider the claims: "our method can detect the second-order leakage (...) the first-order parametric methods cannot" and similarly, “first-order parametric methods completely fail in this setting because they are not sensitive to the second-order associations". The limitations of first-order parametric methods are well-established and were clearly pointed out in the introduction. Given the known inherent differences, I believe it should be avoided to reinstate these limitations to create a favorable comparison to ALL.

- **[Suggestion]:** I suggest using a different LaTeX font style for ALL (\textsc{ALL}, \textsf{ALL}, \texttt{ALL}, \emph{ALL}). Often, "all" serves as a natural continuation within a sentence, and using all caps may even be justified occasionally for emphasis. In the first few sections, this differentiation is important, as it can be confusing for readers to discern whether references are made to "Adversarial Leakage Localization" or simply to an emphasized "all" word. I never encountered a section where the answer was not intuitive (or it was not ALL, the paper's method), but it did naturally make me do a couple "double takes" on some occasions.

---

> ### Author Response · Authors · 2026-01-17
> **Rebuttal (1/2)**
>
> Thank you for your detailed review and helpful suggestions. We have updated our manuscript to address your concerns.
> ## Should incorporate additional performance evaluation methodologies
> - As requested, we have extended our main results table (Table 2 in updated manuscript) to include performance under all 4 of the quantitative performance metrics previously in Tables 10--13 of our appendix.
> 	- We found and fixed an isolated issue in the error bar computation for the DNN occlusion metrics; thus, the error bars for these two metrics are narrower than in the initial submission.
> - As a supplement to these quantitative metrics, we have updated Fig. 5 with additional qualitative results on the ASCADv1-variable dataset from our appendix. In this setting there are multiple distinct pairs of internal AES variables which are sufficient to determine the secret data. We show that our method identifies leakage associated with all such pairs, whereas the strongest baseline primarily identifies leakage from only one such pair.
> ## Why are the ASCADv1 datasets the only second-order datasets used?
> - To our knowledge, the ASCADv1 datasets are the only public second-order datasets which have been used in prior work on deep learning-based leakage localization. We follow this common practice.
> - The ASCADv1 datasets are particularly convenient because their leakage has been studied in detail in a white-box manner by Egger et al. (2022), and we can use this analysis to validate the outputs of our method and the baselines.
> - We emphasize that ASCADv1-fixed and ASCADv1-variable are distinct datasets derived from the same AES-128 code and hardware. They differ in the following important respects, resulting in complementary evaluation settings:
> 	- They were measured with different setups: the former was sampled at 200MS/s and the latter at 500MS/s. Additionally, the leakage of the former is significantly less temporally-local than the latter, possibly due to stronger low-pass filtering effects during measurement (Egger et al. 2022).
> 	- For the former dataset the AES key was held constant across all collected datapoints, whereas for the latter the key was randomized.
> 	- The former consists of 60k power traces each with 700 measurements, whereas the latter consists of 300k power traces each with 1400 measurements.
>
> Reference: Maximilian Egger, Thomas Schamberger, Lars Tebelmann, Florian Lippert, and Georg Sigl. A second look at the ASCAD databases. In International Workshop on Constructive Side-Channel Analysis and Secure Design, pp. 75–99. Springer, 2022.
>
> ## Can we clarify how we can fairly compare the methods despite differences in underlying methodology?
> -  We have expanded our results section to clarify our comparison methodology and experimental setup. See lines 314--329 where we discuss the experimental setup, and lines 330--360 where we discuss the performance metrics.
> - Here are some salient details:
> 	- Despite their methodological differences, all considered methods can be viewed as a function which takes a dataset as input and returns a per-timestep estimated leakiness value as output. Our performance metrics compare the fidelity of these estimated leakiness values and are agnostic to how they were generated.
> 	- Because the specific numbers used to encode leakiness by the different baselines are not directly comparable, we use metrics which are sensitive only to the relative leakiness assigned to different timesteps.
> 	- The baselines entail supervised training of a classifier, and ALL entails adversarially training a classifier $\Phi_\theta$ vs. a noise distribution. We use the same architecture for both classifiers.
> 	- We use the same training step count and minibatch size for supervised classifier training and for pretraining + adversarial training of ALL. Additionally, we use the same hyperparameter tuning budget to tune the important hyperparameters of both.
> ## Can we reduce emphasis on performance vs. first-order baselines?
> - Based on this request, we have made the following changes:
> 	- We removed the first paragraph of Sec. 4.3 where we re-stated the advantages of our method over the first-order baselines.
> 	- We shortened Sec. 3.1 where we present first-order methods as related work. We also rephrased it to focus on situating ALL in the broader landscape of side-channel leakage localization approaches, in which first-order methods are useful for white-box analysis despite being a poor fit for our setting.
> 	- We re-phrased exposition around the 'Toy setting' experiments to present first-order comparisons as done for completeness, rather than as a peer baseline.

---

> ### Author Response · Authors · 2026-01-17
> **Rebuttal (2/2)**
>
> ## Broader impact concerns
> - We agree with your point here, and have updated our broader impact statement (lines 367--377).
> - We now more-narrowly state that our method does not directly improve the performance of profiling side-channel attacks, rather than 'attackers'.
> - We also note that identifying weakness of a system can potentially benefit attackers as well as defenders.
> ## Additional requested changes
> - We have updated the manuscript so that the baselines use consistent names.
> - We have updated the right subplot of Fig. 5 (now Fig. 16 in appendix) so that the 1-occlusion and m*-occlusion traces have different colors.
> - We have replaced "ALL" with "\textsf{ALL}" throughout the manuscript.

---

### Review · Reviewer_8bB8 · 2026-01-07

**Summary Of Contributions:**

This paper proposes a defense against side-channel attacks targeting physical implementations of cryptographic algorithms. The side-channel attack considered in this work aims at recovering secret variables of the scheme, such as cryptographic keys, by exploiting power and electromagnetic emission measurements from the cryptographic hardware. The proposed defense, called ALL, identifies, among a set of measurements, those that leak the most information, quantified through an arbitrary statistical association between the target information and the measurement at a given time. The leakiness of a measurement is defined via a constrained optimization problem, which is solved through the core deep learning algorithm underlying ALL, since solving it exactly would require exponential time in the number of measurements. Compared to state-of-the-art approaches, ALL is able to detect more subtle and complex leakages more accurately, while also accounting for redundancy among measurements in a sequence when computing leakiness.

The experimental evaluation is comprehensive, considering 11 baselines and 6 datasets. ALL often outperforms the baselines in terms of detecting leaky measurements. Moreover, the method is easy to tune and shows consistent performance across different parameter values.

**Audience:**

Yes

**Audience Explanation:**

The topic of this work embraces three different research areas: machine learning, cryptography, and computer security. These three components are equally represented and even though the main focus of the paper is a side-channel attack against cryptographic schemes, the authors do a very good job of providing the necessary background, allowing even an audience whose primary expertise is in AI or machine learning to fully appreciate the work. Therefore, I think that an interdisciplinary work like this may be of interest to the TMLR community because of its originality compared to contributions that are more narrowly focused on machine learning topics and it may also attract readers from other research communities to TMLR.

**Broader Impact Concerns:**

Nothing to signal.

**Claims And Evidence:**

Yes

**Claims Explanation:**

The experimental evaluation supports the effectiveness of ALL, that achieves strong results across different metrics. The experimental axes considered are numerous, including a clear example illustrating cases in which ALL succeeds while other algorithms fail, experiments demonstrating ability of ALL to detect the leakiness of specific components of the cryptographic pipeline and an important ablation study presented in the Appendix. Overall, the results are convincing with respect to the baselines and the paper does a good job of clearly stating the limitations of the approach (also discussed in the Appendix).

**Requested Changes:**

I propose the following suggestion to improve the presentation and strengthen the work:
- I would consider moving part of Section 5.2 to the Appendix (such as the center and right subfigures of Figure 5 and their related discussion) and, instead, integrating more details from the Appendix about the Simulated AES-128 datasets experiment. Indeed, as far as I understand, this experiment is the one that shows that, in practice, it is possible to understand how and why the hardware leaks, since the ALL algorithm is applied while varying some components of the cryptographic implementation. In my opinion, this section is particularly important, given the motivations behind this work, and its content should be emphasized more, while the analysis of ALL’s performance under different hyperparameter settings or training steps can be delegated to the Appendix.

---

> ### Author Response · Authors · 2026-01-17
>
> Thank you for your positive review. We are glad you appreciated our work!
> - As suggested, we have brought additional details about the synthetic AES-128 experiments from the appendix to the main paper. See lines 291--309 of our updated manuscript.

---

> > ### Comment · Reviewer_8bB8 · 2026-01-24
> > **Thank you**
> >
> > Thank you to the authors for addressing my comment.

---

### Decision · Action_Editor_SQ7U · 2026-02-15

**Recommendation:** Accept as is

**Audience:**

Yes

**Audience Explanation:**

The topic is of interest to some of the TMLR audience.

**Claims And Evidence:**

Yes

**Claims Explanation:**

After the authors responses, the reviewers agree that claims are supported by evidence.

The paper proposes a method for defending against side-channel attacks targeting physical implementations of cryptographic algorithms.  The proposed defense, called ALL, identifies, among a set of measurements, those that leak the most information, quantified through an arbitrary statistical association between the target information and the measurement at a given time.  The claims about the performance of the method are scoped well and supported with clear empirical evidence on a variety of datasets and in comparison to appropriate baselines.